# Compress then Serve: Serving Thousands of LoRA Adapters with Little Overhead

## Abstract

Fine-tuning large language models (LLMs) with low-rank adaptations (LoRAs) has become common practice, often yielding numerous copies of the same LLM differing only in their LoRA updates. This paradigm presents challenges for systems that serve real-time responses to queries that each involve a different LoRA. Prior works optimize the design of such systems but still require continuous loading and offloading of LoRAs, as it is infeasible to store thousands of LoRAs in GPU memory. To mitigate this issue, we investigate the efficacy of model compression when serving LoRAs. We propose a method for joint compression of LoRAs into a shared basis paired with LoRA-specific scaling matrices. We extend our algorithm to learn clusters of LoRAs that are more amenable to joint compression, allowing it to scale gracefully to large LoRA collections. Our experiments with up to 500 LoRAs demonstrate that compressed LoRAs preserve performance while offering major throughput gains in realistic serving scenarios with over a thousand LoRAs, maintaining 80% of the throughput of serving a *single* LoRA.

## 1 Introduction

The myriad uses for foundation models (FMs) have led to a proliferation of specialized models, each fine-tuned to perform a downstream task. To avoid fine-tuning foundation models with billions of parameters, more parameter-efficient fine-tuning (PEFT) algorithms were proposed. An especially successful PEFT method is low-rank adaptation (LoRA) (Hu et al., 2021), which learns low-rank additive changes to neural network matrices. Because of the low-rank parameterization, these matrices (called adapter weights) contain orders-of-magnitude fewer parameters than the base model. Still, LoRA can achieve performance on par with full fine-tuning (Hu et al., 2021).

LoRA's popularity has triggered a growing need to serve large collections of LoRA adapters at scale. For example, proprietary and open-source LLM providers offer fine-tuning services (OpenAI, 2024; TogetherAI, 2024; Predibase, 2024) with user bases likely in thousands or even hundreds of thousands. As each user wants to use their own fine-tuned version of the LLM, simply serving a dedicated fine-tuned LLM per user becomes infeasible. To this end, S-LoRA (Sheng et al., 2023) considers a system where only the base LLM is placed on an inference server and individual LoRA adapters are switched as needed at inference time. S-LoRA optimizes the system's inner workings via custom CUDA kernels and memory management to increase the throughput when serving multiple LoRAs. Such multi-LoRA system design has also been adopted in vLLM (Kwon et al., 2023), a state-of-the-art LLM serving engine. Despite the optimized system design, serving LoRAs still has a fundamental limitation: when the number of adapters is large, they need to be constantly loaded and offloaded from GPU memory to accommodate incoming requests, degrading throughput.

The problem of accommodating multiple LoRA adapters is also apparent when placing LLMs on edge devices, where smaller LLMs are fine-tuned for various tasks and the adapters are swapped depending on the task at hand (Gunter et al., 2024). In this case, the amount of adapters is smaller, e.g., a few dozen (Gunter et al., 2024), but the memory constraints are also tighter due to the limited capacity of edge devices.

In this work, we consider the problem of compressing a collection of LoRAs. We have two key objectives: (1) preserving the performance of the original LoRAs and (2) improving the throughput of serving many LoRAs. We formulate LoRA compression as a reconstruction problem, where the

goal is to approximate the original adapters via collections of matrices of a smaller total size. We investigate an approach based on compressing LoRAs *jointly* by finding a shared basis and LoRA-specific scaling matrices, and propose a joint diagonalization-based algorithm (JD). To improve reconstruction error for large numbers of LoRAs while keeping the number of parameters in check, we propose a clustering approach where each cluster is compressed independently using the joint diagonalization algorithm. Our clustering algorithm is based on alternating between optimizing the cluster assignments and the per-cluster reconstruction error.

We showcase the benefits of joint compression in Figure 1. We chose JD configurations that match the constraints. When serving up to 64 unique LoRAs we use JD without clustering and for 128 or more, we pick the number of clusters to match the performance of compressed and original LoRAs. In each case, the GPU memory footprint of the compressed and original LoRAs is matched for a fair comparison to the vLLM's multi-LoRA inference engine. When serving over 1000 LoRAs, compression increases throughput by a factor of 1.6 and maintains 80% of the throughput of serving the base LLM (or a single LoRA merged into the LLM). Detailed results are presented in Section 6.

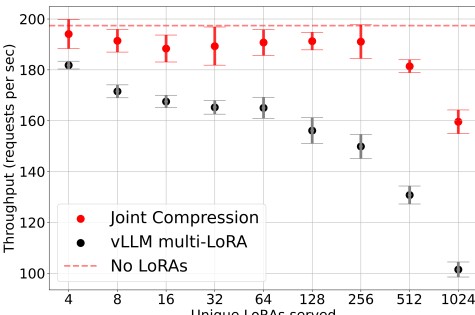

Figure 1: Throughput gains when serving 1000s of compressed LoRAs with vLLM.

We summarize our main contributions below:

- We formulate the problem of compressing a collection of LoRAs and propose a joint compression scheme based on joint diagonalization.
- For large numbers of LoRAs, we scale the joint compression scheme by proposing a clustering algorithm where each cluster is jointly compressed to minimize the overall reconstruction error.
- We establish theoretical guarantees for the reconstruction error central to our compression formulation and verify the relation between reconstruction loss and performance empirically.
- We train a collection of 500 high-quality LoRAs for `Mistral-7B-Instruct-v0.2` (Jiang et al., 2023a) on 500 natural instruction tasks (Wang et al., 2022) and demonstrate that our compression techniques preserve the performance of the original LoRAs. We will release the 500 LoRAs to facilitate future work on LoRA compression as well as the code for our method.
- We incorporate LoRA compression into a state-of-the-art LLM serving system and demonstrate that it is possible to serve over 1000 LoRAs across thousands of asynchronous requests with throughput comparable to serving a *single LoRA*.

## 2 RELATED WORK

Parameter-efficient fine-tuning (PEFT) has become prevalent for updating foundation models thanks to the need for efficiency in training and communication (Lialin et al., 2023). Many PEFT methods have been proposed, e.g. (Houlsby et al., 2019; Liu et al., 2022) and LoRA (Hu et al., 2021) became the standard, partially due to the ease of switching between LoRAs in inference time.

Several works improve LoRA (Liu et al., 2024; Wang et al., 2024), sometimes with algebraic methods like SVD (Meng et al., 2024; Zhang et al., 2023; Jiang et al., 2023b) or by leveraging its statistical properties (Zhu et al., 2024; Zeng & Lee, 2024). Relatively few, however, accelerate inference times. S-LoRA (Sheng et al., 2023) provides an efficient means of switching between LoRAs. (Wen & Chaudhuri, 2024) adapts training to reduce batch multiplications, accelerating inference. Our method achieves a similar outcome (see Appendix D) without changing the LoRA formulation or requiring that LoRAs be trained in a dedicated way; future improvements to LoRA will also benefit from this aspect of our work (e.g., Meng et al. (2024)).

Punica (Chen et al., 2023) introduces Segmented Gather Matrix-Vector Multiplication (SGMV) to optimize multi-LoRA serving by parallelizing feature-weight multiplications in batches and grouping requests that utilize the same LoRA model. Our approach, by contrast, emphasizes parameter reduction as a means to efficiently serve multiple LoRAs, providing an orthogonal strategy that can be seamlessly integrated with Punica's methods to further enhance performance. In our vLLM

experiments, we leveraged the Punica kernel for multi-LoRA implementation, demonstrating the application of our method in conjunction with Punica's optimizations.

There are many efforts to compress models (Cheng et al., 2017; Gholami et al., 2022; Sharma et al., 2024; Li et al., 2018)—including some specifically for LoRAs—to accelerate inference. Predominantly, pruning and sparsification methods delete weights (Yadav et al., 2023a), and quantization methods reduce the weights' precision (Dettmers et al., 2024). Some works compress weights to reduce model size but typically require decompression and hence do not save GPU memory (Hershcovitch et al., 2024). Similarly to our work, while most methods increase speed at the cost of performance, a few note increased performance and generalization after compression (Yadav et al., 2023a; Nadjahi et al., 2023; Hershcovitch et al., 2024; Sharma et al., 2024).

Our work also relates to model merging (Choshen et al., 2022; Wortsman et al., 2022; Matena & Raffel, 2021) and mixture of experts methods (Muqeeth et al., 2024; Yadav et al., 2024). These methods reuse models trained by others (Choshen et al., 2023; Raffel, 2023), serving them together as one compressed model. Despite this similarity, these methods create a single general model that acts on any input, while our model allows for more performant per-task solutions.

## 3 RANK-BASED LoRA COMPRESSION

LoRA updates are parameterized by pairs of matrices $A, B$, whose product $BA$ updates the fixed weight matrices $W_0 \in \mathbb{R}^{d_B \times d_A}$ of a neural network foundation model. Given an input $x$ to a layer, the output of the LoRA-updated model at this layer is $(W_0 + BA)x$.

In formulating our compression algorithms, we consider a collection of given LoRA adapters $\{(A_i, B_i)\}_{i=1}^n$ that we would like to serve. We let $r_i$ refer to the rank of the LoRA adapter-pair $(A_i, B_i)$, i.e., $B_i \in \mathbb{R}^{d_B \times r_i}$, $A_i \in \mathbb{R}^{r_i \times d_A}$.

While our compression technique has access only to a collection of $\{(A_i, B_i)\}_{i=1}^n$ pairs, in our experiments we will assess the efficacy of compression by comparing how the compressed matrices perform relative to the uncompressed LoRAs on typical data. For this reason, although in this section we optimize a Frobenius norm reconstruction error relative to the product $B_i A_i$, in reality this is a proxy for the nonlinear and complex way that compression errors in the adapters impact transformer performance. Our experimental evaluation will thus focus on the performance of the compressed LoRAs against the uncompressed versions on real data in §6.

Our compression methods significantly reduce the overall number of parameters. Reducing parameters through compression theoretically accelerates storage and serving processes for a collection of LoRAs. This reduction, however, alters the computational dynamics during inference, so parameter reduction alone does not immediately imply faster throughput. In light of the complexities of GPU optimization, we experimentally assess the throughput under realistic conditions in §6.3.

### 3.1 JOINT DIAGONALIZATION

For compression to scale to large numbers of LoRAs, the compressed number of parameters should not scale linearly with $n$. Hence seeking to compress each LoRA individually (e.g., via SVD as detailed in the experimental baselines) is inherently limited.

To address this, we suggest a Joint Diagonalization (JD) method, which optimizes a shared basis onto which we can project the set of $n$ LoRAs. This will allow structure to be shared, implicitly grouping and/or merging the collection of LoRAs.

In this model, each LoRA product $B_i A_i$ is factorized into the form $U \Sigma_i V$, where $U$ and $V$ are shared across all LoRAs and $\Sigma_i$ is specific to each LoRA. In this formulation, every $\Sigma_i$ shares the same rank $r$. This allows $U$ and $V$ to be pre-loaded onto the GPU, with $\Sigma_i$ loaded when necessary for each batch. The matrices $\Sigma_i$ can be either diagonal or small square matrices, accelerating the forward pass compared to conventional multi-LoRA serving configurations.

**Objective function.** Motivated by the relationship of singular value decomposition to minimizing the Frobenius norm of the reconstruction error, we also propose to minimize the Frobenius norm of

the adapter matrix approximation error. Specifically, we use the following objective function:

$$\min_{\{\Sigma_i\}_{i=1}^n, U, V} \sum_{i=1}^n \|B_i A_i - U \Sigma_i V^\top\|_{\text{Fro}}^2. \tag{1}$$

Note this problem is *not* solved by a single matrix SVD, since $U$ and $V$ are shared among all terms but the $\Sigma_i$'s are not. Using the Frobenius norm has the added benefit of making the objective convex in each argument separately, suggesting the possibility of efficient optimization. This objective function is underdetermined, however, so we consider two constrained regimes below.

**Full $\Sigma_i$ approximation.** The first method we call JD − Full. Without loss of generality, $U$ and $V$ can be constrained to be orthogonal, so long as $\Sigma_i$ remains an unconstrained full matrix. JD − Full adopts this restriction to make the optimization better posed, but note it does not restrict the expressiveness of the objective equation 1. This setting yields the following optimization problem:

$$\text{JD-Full}_r(\{B_i A_i\}_{i=1}^n) = \underset{\substack{\{\Sigma_i\}_{i=1}^n \\ U^\top U = V^\top V = I_r}}{\arg\min} \sum_{i=1}^n \|B_i A_i - U \Sigma_i V^\top\|_{\text{Fro}}^2 \quad (\text{JD} - \text{Full}) \tag{2}$$

An efficient alternating algorithm to solve this objective function can be found in Appendix A.

**Diagonal $\Sigma_i$ approximation.** As an alternative, we can leave $U$, $V$ unconstrained (other than to have $r$ columns) and instead constrain the matrices $\Sigma_i$ to be diagonal (but not necessarily positive). This formulation yields the following optimization problem:

$$\text{JD-Diag}_r(\{B_i A_i\}_{i=1}^n) = \underset{\{\Sigma_i\}_{i=1}^n, U, V}{\arg\min} \sum_{i=1}^n \|B_i A_i - U \text{diag}(\Sigma_i) V^\top\|_{\text{Fro}}^2 \quad (\text{JD} - \text{Diag})$$

$$\tag{3}$$

An efficient alternating least squares algorithm to optimize this objective can be found in Appendix A. This diagonal version has some per-LoRA parameter savings when compared to JD − Full, since the diagonal $\Sigma_i$ only needs $r$ parameters instead of $r^2$.

## 3.2 CLUSTERING

As the number of LoRAs $n$ grows and becomes more diverse, the rank $r$ needed for Joint Diagonalization to achieve good performance will tend to increase. This increases the size and number of parameters of each $\Sigma_i$ that needs to be stored, especially for JD-Full which will require $O(nr^2)$ storage for these matrices. If the necessary $r$ is growing proportionally to $n$, then this storage will eventually become the bottleneck.

To resolve this limitation with very large $n$, we propose to group the $n$ LoRAs into $|C_j|$ clusters $C_j$. Each cluster is given its own rank $r$ JD compression, and the clusters are chosen such that the overall reconstruction error is minimized. Specifically, the overall objective is

$$\min_{\{\{C_j\}, U_j, V_j\}, \{\Sigma_i\}} \sum_j \sum_{i \in C_j} \|B_i A_i - U_j \Sigma_i V_j\|_F^2,$$

and we optimize this by alternating between cluster assignments and the JD of each cluster. The algorithm details are in Appendix A.3. Typically, the goal with large $n$ is to have $|C_j|$ grow with $n$ as $r$ becomes fixed. Comparing $k$ rank-$r$ JD-Full clusters to a rank-$kr$ JD-Full single cluster compression, observe that the clustered approach requires $O(dkr + nr^2)$ parameters, while the single-cluster approach requires $O(dkr + nk^2r^2)$ parameters due to the increased sizes of the $\Sigma_i$s. While these two approaches have the same rank, note that they may have different reconstruction abilities. Empirically, we find that multiple clusters significantly aid performance for $n \geq 100$.

# 4 THEORETICAL ANALYSIS

In this section, we seek to better understand the role of the joint diagonalization method presented in §3.1 and how this understanding further motivates the clustering approach. We will focus on the full-$\Sigma_i$ case with orthogonal $U$, $V$ matrices. Note that, for the same $r$, the $r-\mathsf{JD}-\mathsf{Diag}$ has at least as large reconstruction error as $r-\mathsf{JD}-\mathsf{Full}$ since it imposes an additional constraint on the $\Sigma_i$.

Firstly, note that perfect reconstruction can be achieved if and only if $r$ is large enough, since there exist $U, V$ such that all the $B_i$, $A_i$ are in the spans of $U, V$ resp. if and only if $r \geq \tilde{r}$:

**Proposition 1.** *Suppose that for all $i$, $\mathrm{rank}(B_i A_i) = r_i$, and let*

$$\tilde{r} = \max \left\{ \mathrm{rank}([A_1, \ldots, A_n]), \mathrm{rank}([B_1^\top \ldots, B_n^\top]) \right\}.$$

*Note $\max_i r_i \leq \tilde{r} \leq \sum_{i=1}^n r_i$. Then $\mathsf{JD} - \mathsf{Full}$ (equation 2) with $r = \tilde{r}$ achieves lossless compression (perfect reconstruction), and using $r < \tilde{r}$ will give nonzero reconstruction error.*

Due to training noise, $\tilde{r}$ will equal $\sum_{i=1}^n r_i$ almost always. This implies that in most realistic settings, the joint diagonalization approach is a lossy reconstruction.

This reconstruction loss can be significant, as the following theorem shows (proved in Appendix B):

**Theorem 1.** *Consider $n$ LoRAs ($\{A_i, B_i\}_{i=1}^n$) with $r, n \leq d^2$, and form the matrix*

$$L = [ \ \mathrm{vec}(B_1 A_1) \ \ \cdots \ \ \mathrm{vec}(B_n A_n) \ ].$$

*Let $\sigma_j$ be the singular values of $L$, sorted from largest to smallest, and let $\bar{\sigma}_j$ be the singular values of $\sum_{i=1}^n B_i A_i$. Then, using $\mathsf{JD} - \mathsf{Full}$ (equation 2),*

$$\sum_{j=1}^r \bar{\sigma}_j^2 \leq \sum_{i=1}^n \|\Sigma_i\|_{\mathrm{Fro}}^2 = \sum_{i=1}^n \|U\Sigma_i V^\top\|_{\mathrm{Fro}}^2 \leq \sum_{j=1}^{\min(r^2, n)} \sigma_j^2,$$

*implying the sum of squared Frobenius norms of the reconstructed LoRAs satisfies*

$$\frac{\sum_{i=1}^n \|U\Sigma_i V^\top\|_{\mathrm{Fro}}^2}{\sum_{i=1}^n \|B_i A_i\|_{\mathrm{Fro}}^2} \leq \frac{\sum_{j=1}^{\min(r^2, n)} \sigma_j^2}{\sum_{j=1}^n \sigma_j^2} \leq 1, \ and \ \frac{\sum_{i=1}^n \|U\Sigma_i V^\top - B_i A_i\|_{\mathrm{Fro}}^2}{\sum_{i=1}^n \|B_i A_i\|_{\mathrm{Fro}}^2} \geq 1 - \frac{\sum_{j=1}^{\min(r^2, n)} \sigma_j^2}{\sum_{j=1}^n \sigma_j^2}.$$

In other words, if the singular values of $L$ are not concentrated in the top $r^2$ entries, significant reconstruction error is unavoidable.

**Remark 1** (Lower bound and merging). *The lower bound $\sum_{j=1}^r \bar{\sigma}_j^2$ could be achieved by setting all the $\Sigma_i$ equal, i.e., using a fully merged model instead of only merging the subspaces $U$, $V$ and allowing $\Sigma_i$ to vary with $i$.*

**Remark 2** (Upper bound and grouping). *The upper bound is smallest when the LoRAs are relatively clustered, i.e., when groups of vectors $\mathrm{vec}(B_i A_i)$ are similar. This situation raises the magnitude of the largest singular values of $L$, raising the upper bound in the proposition. As the LoRAs are $d \times d$ matrices that can be thought of as points in $d^2$ dimensional space, for typical values of $d$ well into the hundreds, it is likely that unrelated LoRAs will be unclustered, i.e., they will have relatively low inner products with each other.*

For the case of orthogonal LoRAs, the singular values of $L$ are the norms of the LoRAs, and we immediately have the following corollary:[1]

**Corollary 1.** *Suppose (e.g., due to normalization) that the inputs to the joint diagonalization algorithm all have unit Frobenius norm, i.e., $\|B_i A_i\|_{\mathrm{Fro}} = 1$. Moreover, assume that the LoRAs are all orthogonal in the sense $\mathrm{tr}((B_i A_i)(B_j A_j)^\top) = 0$ for $i \neq j$. Then, using the $\mathsf{JD} - \mathsf{Full}$ method equation 2, we have $1 \leq \sum_{i=1}^n \|\Sigma_i\|_{\mathrm{Fro}}^2 \leq \min(r^2, n)$, implying that the sum of squared Frobenius norms of the reconstructed LoRAs satisfies*

$$1 - \frac{1}{n} \geq \frac{\sum_{i=1}^n \|U\Sigma_i V^\top - B_i A_i\|_{\mathrm{Fro}}^2}{\sum_{i=1}^n \|B_i A_i\|_{\mathrm{Fro}}^2} \leq 1 - \min\left(\frac{r^2}{n}, 1\right).$$

---

[1]A result for isotropic Gaussian LoRAs could be obtained via the quantiles of the Marchenko-Pastur Law.

This implies that for the common setting where $r^2 \ll n$, the reconstructed LoRAs will be significantly smaller than the original LoRAs and necessarily have significant reconstruction error.

The results in this section illustrate the tradeoffs of using joint diagonalization. If the LoRAs are similar or well-clustered, reconstruction error will be low. On the other hand, if the LoRAs are random and orthogonal, reconstruction error will be high.

Since the loss space of transformers is highly complex, increasing weight reconstruction error does not necessarily imply degrading LLM performance. Interestingly, in Figure 3 below, we see that while large reconstruction error rapidly decreases performance, moderate (but still relatively large, at around 0.4) reconstruction error does not damage performance and may even slightly outperform the zero-error setting. This observation motivates our focus on minimizing weight reconstruction error, while also suggesting that our approach is capable of achieving something deeper than compression. Specifically, the tendency of joint diagonalization is to find subspaces that are shared among many LoRAs when $r$ is large, and to *merge* subspaces when $r$ is small. When $r$ is particularly small, this tendency towards *averaging* all or some of the LoRA signals directly connects to the concept of *merging* LoRAs, whose empirical success (Shah et al., 2023; Huang et al., 2024) could explain the success of our procedure despite the nonlinearity of transformers.

Experiments in Appendix G.9 explore this idea further, comparing reconstruction of real-world LoRAs to reconstruction of randomly sampled LoRAs. The reconstruction error is generally large, but significantly lower than the reconstruction error for random noise, indicating that there is a major shared component between the LoRAs that is being successfully retained.

That said, as the number of LoRAs grows, the shared component may not be significant enough to maintain sufficiently low reconstruction error with low rank $r$. This motivates the introduction of *clustering* in §3.2, since clustering seeks to find groups of LoRAs that are similar and better compressible by joint diagonalization. In particular, if the number of clusters $|C_j|$ grows with $n$, the reconstruction error may no longer degrade with $n$ even when $r$ is fixed.

In the extreme case where $|C_j| = n$, each LoRA is compressed independently. By the Eckart-Young Theorem, JD applied to a single LoRA reduces to an SVD, replacing each rank-$r_i$ LoRA adapter $B_i A_i$ with a reduced rank-$r$ approximation, where typically $r < \frac{1}{n} \sum_{i=1}^{n} r_i$:

$$\text{SVD}_r(B_i A_i) = U_i \Sigma_i V_i^\top, \quad \forall i = 1, \ldots, n. \tag{4}$$

As $\Sigma_i V_i^\top$ can be saved as a single matrix, this approach has $rn(d_A + d_B)$ parameters. We refer to this $|C_j| = n$ method as $\mathsf{r} - \mathsf{SVD}$ and find that it underperforms our other methods, while outperforming the baseline uncompressed LoRAs significantly. This result parallels Jiang et al. (2023b)'s observation that lowering LoRA ranks is beneficial for multi-task learning and model merging.

## 5 Training LoRAs & Evaluating Task Performance

### 5.1 Training

We trained LoRA adapters on 500 natural instruction tasks (Wang et al., 2022) using `Mistral-7B-Instruct-v0.2` (Jiang et al., 2023a) as the base model. All LoRA adapters were configured with a rank of 16, i.e., $\forall i, r_i = 16$.

We selected 10 diverse tasks (Table 2 in Appendix C) manually for consistent evaluation across experiments and randomly sampled an additional 490 tasks, resulting in a total of 500 tasks. These tasks were exclusively in English (both input and output), ensuring higher quality and thorough review (Wang et al., 2022). The tasks represent a realistic and varied set, not inherently clustered. Each task dataset was divided into training, validation, and test sets.

Hyperparameters, such as early stopping, were tuned using the validation sets. Evaluation on the test sets demonstrated that LoRA consistently outperformed the base model in terms of both Rouge scores and loss metrics, as shown in Table 1. Details are provided in Appendix C.

### 5.2 Evaluation

We evaluated multiple metrics for the natural instruction tasks, including cross-entropy loss, Rouge-1, Rouge-L (Lin, 2004), exact match, and *agreement* between uncompressed and compressed LoRA.

Table 1: Comparison of metrics before and after LoRA training across 500 tasks.

| Metric | Base Model | LoRA |
|---|---|---|
| Loss | $4.99 \pm 3.11$ | $0.43 \pm 0.57$ |
| Exact Match | $2.28 \pm 7.89$ | $66.66 \pm 34.34$ |
| Rouge-1 | $20.38 \pm 18.90$ | $76.74 \pm 24.89$ |
| Rouge-L | $19.66 \pm 18.16$ | $76.22 \pm 25.27$ |

Here, *agreement* measures the exact match in task-generations between the uncompressed LoRA model and the compressed LoRA model, rather than comparing to ground truth data. While detailed results and discussions for all metrics are provided in Appendix G, our primary focus in the main text is on Rouge-L. We find that all metrics correlate, but Rouge-L correlates most strongly with downstream utility. This finding aligns with prior work (Wang et al., 2022), which demonstrates that Rouge-L correlates well with classification accuracy.

While cross-entropy is used for optimization during training, identical generation outputs across models can yield different cross-entropy losses. Exact match is too rigid and does not account for the variability in task responses. Similarly, agreement does not capture the inexactness associated with most of our tasks, nor does it account for the performance gains or losses of the compressed LoRAs. Arguably, practitioners are primarily concerned with task performance in the settings for which the LoRA was designed, rather than exact generational agreement between models.

Joint diagonalization optimizes reconstruction error measured by the Frobenius norm, and our theoretical analysis in §4 bounds this reconstruction error. Empirically, reconstruction error and downstream Rouge-L performance correlate.

Instead of listing the absolute performance of different methods, we compute the performance difference between the base model and the LoRA model for each task. We present the ratio

$$\text{Performance relative to LoRA} := \frac{\text{method-performance}}{\text{LoRA-performance}}$$

for the specific method in question, highlighting relative improvement with respect to the uncompressed LoRA and the base model.

## 6 EXPERIMENTS

### 6.1 TASK PERFORMANCE

For each method, we vary the number of $n$ LoRAs that are compressed and the compression rank $r$. We run each experiment three times with different random seeds and report the mean and standard deviation. See Table 4 for results where we evaluate on the same ten manually selected tasks (Table 2) across settings. Every compressed collection of LoRAs contains these 10 tasks (i.e., in-distribution tasks), and each collection contains the smaller collections as subsets.

We normalize each LoRA adapter to have a Frobenius norm of one prior to running joint diagonalization. This normalization enhances performance and reduces the variance in reconstruction error. We restore the original norms of the LoRA adapters before reconstruction and testing.

Figure 2a illustrates the Total Parameter Saved Ratio versus the Number of Unique LoRAs served. We only include methods that maintain over 99% of the original LoRA's performance (as measured by RougeL). Notably, our JD methods uniquely approach the compression efficacy of a single LoRA, and with clustering, this aggressive reduction in size also maintains performance in larger LoRA collections.

Figure 2b illustrates the Rouge-L scored of the compressed LoRAs divided by the Rouge-L score of the uncompressed LoRAs. It is interesting to note that JD variants often increase generalization and outperform the original LoRA. In Appendix G, we include multiple tables of results for additional metrics, relative as well as absolute.

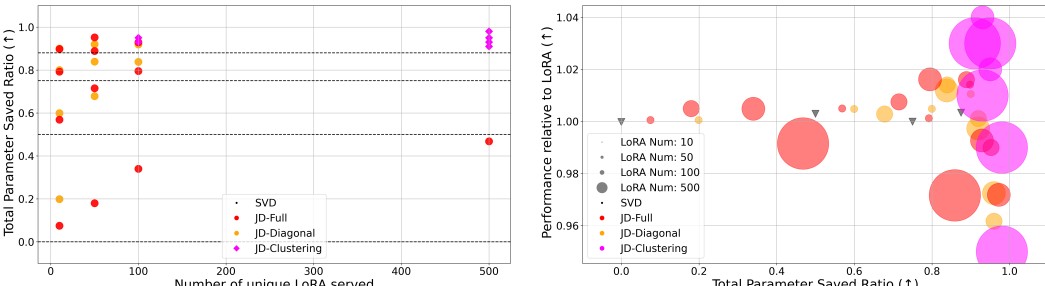

(a) Total parameter saved ratio with the number of unique LoRAs served.

(b) Performance relative to LoRA with total parameter saved ratio

Figure 2: Performance after compression. In (a), we only include methods that maintain over 99% of the original LoRA's performance (as measured by RougeL). In (b), we compare the performance of compressed LoRAs relative to uncompressed ones, with higher values on both axes reflecting better performance. The Total Parameter Saved Ratio depicts the number of parameters saved for a system with a large number $n$ of different LoRAs. It is computed as: $r_{total} := 1 - \frac{\text{num. parameters after compression}}{\text{num. parameters before compression}}$.

For efficiency, we limited the JD methods to ten iterations instead of pursuing full convergence. While the alternating algorithm quickly reaches an approximation of the minimizer, squeezing out the last few digits of precision takes many more iterations with limited to no performance gain. Appendix G.10 also evaluates an alternative eigenvalue iteration algorithm that more rapidly converges once $U, V$ are close to a minimizer, with minimal performance differences.

## 6.2 PERFORMANCE AND RECONSTRUCTION ERROR

Figure 3 relates reconstruction error and performance. The $y$-axis measures the mean performance improvement of Rouge-L relative to uncompressed LoRA, and the $x$-axis quantifies the mean reconstruction error between the compressed reconstruction of the product $BA$ and the original uncompressed product $BA$. Although the relationship between performance and reconstruction error is nonlinear, it demonstrates a generally decreasing, somewhat exponential trend. Notably, the minimal reconstruction error does not correlate with optimal performance, indicating that a degree of lossy reconstruction may be advantageous for enhancing generalization.

To select hyperparameters (compression rank and number of clusters) for the clustering experiments, we first assessed reconstruction error on a single LoRA module over a range of settings (see Appendix F). These preliminary experiments enabled efficient selection of cluster counts and rank values for compressing all LoRA modules.

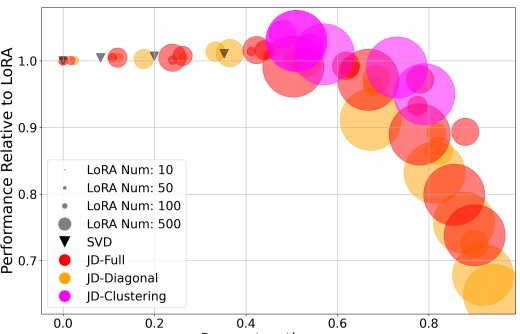

Figure 3: Reconstruction error vs. performance.

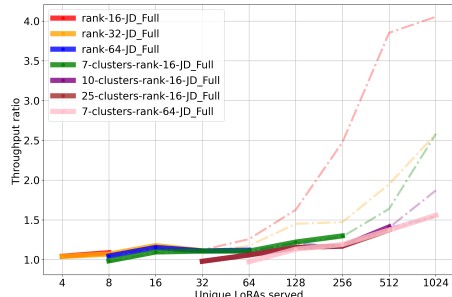

Figure 4: Throughput ratio when serving varying numbers of LoRAs with vLLM.

## 6.3 THROUGHPUT OF SERVING COMPRESSED LoRAS

Results in the previous sections demonstrate how to select an appropriate joint compression setting guided by the reconstruction error, such that the performance of the original LoRAs is preserved.

Naturally, the rank and/or the number of clusters for the compression needs to increase as we compress larger LoRA collections to match LoRA performance.

In Figure 4 we study how throughput with various compression settings compares to the vLLM multi-LoRA throughput with the matched GPU memory footprint. Specifically, for each number of unique LoRAs served and each compression setting, we compute the corresponding number of LoRAs to be placed on the GPU during serving and report the ratio of the two throughputs. For example, when serving 64 unique LoRAs and using rank 64 JD-Full compression, we report the ratio of throughputs of rank 64 JD-Full and vLLM multi-LoRA with 6 LoRAs allowed on the GPU at a time (see Appendix E for details). As the number of unique LoRAs increases, vLLM multi-LoRA throughput degrades as it needs to schedule the requests and load and offload the adapters. We note that vLLM multi-LoRA already employs many advanced system optimization techniques, such as efficient scheduling and non-blocking CPU-GPU communication when swapping LoRAs (Sheng et al., 2023; Kwon et al., 2023), but system optimization alone is not sufficient to fully mitigate throughput degradation when serving many LoRAs.

In Figure 4 we see that across all LoRA collection sizes our compression techniques improve the throughput of vLLM multi-LoRA. Additionally, we highlight regions for each compression setting where compression is sufficiently moderate to achieve 99%+ of LoRA performance, according to the results in Section 6.2. We also note that compression with a larger rank or too many clusters does not improve baseline throughput when serving a smaller number of LoRAs and should not be used in such cases. For example, rank 16 JD-Full improves baseline throughput with 4 and 8 LoRAs, but will underperform with more LoRAs, while 7 cluster rank 64 JD-Full does not improve throughput with 64 or fewer LoRAs, but when serving 1000+ LoRAs it improves the throughput significantly while maintaining the performance. To conclude, an appropriate joint compression setting improves vLLM multi-LoRA throughput and preserves performance for LoRA collections of any size between 4 and 1024, as we showed in Figure 1. Specific compression settings for each LoRA collection size are listed in Appendix E.

Finally, we note that vLLM extensively uses custom CUDA kernels. To accommodate our compression techniques, we minimally adjusted the vLLM code to generate additional kernels needed by the compressed LoRAs while we utilized Punica (Chen et al., 2023) kernel to further accelerate matrix multiplication. A pseudo code is given in E.4 to show how we utilize the batch multiplication kernel. There likely is room for improvement to optimize the newly added kernels.

**Additional details** In this experiment we considered a varying number of rank-16 LoRAs, using a dataset of Shakespeare sonnets as inputs[2] arriving *asynchronously*. We measured throughput, i.e., the number of requests served per second when generating ten tokens per request. The base model was Mistral 7B Instruct as in the other experiments; we simulated random LoRAs and assigned inputs to LoRAs at random. Experiments were conducted on H100 80GB GPU capped at 40% memory consumption. This was done to reflect cost concerns in practical situations where a service provider might want to serve many LoRAs from cheaper hardware with lower memory than higher-end GPUs. This setting also takes into account the scenario where the LLM is large compared to the size of GPU and yet a provider may want to serve many LoRAs efficiently using the same device.

## 6.4 RECOMMENDATIONS

JD-Full is generally preferred over JD-Diag, although for smaller numbers of LoRAs (less than 100), the performance difference is negligible. While JD-Full alone is effective up to 100 LoRAs, incorporating clustering at scales of 500 LoRAs significantly enhances performance.

We recommend the following procedure for hyperparameter selection. For 100 or fewer LoRAs, JD-Full can be utilized independently without substantial degradation, using a rank approximately equal to (number of LoRAs$/2$) + 7. Beyond 100 LoRAs, clustering becomes increasingly critical. A robust method for any number of LoRAs up to 500 involves employing JD-Full with clustering. Specifically, select a LoRA module from the middle of the network, apply a compression rank of 16,

---

[2]https://www.kaggle.com/datasets/shivamshinde123/william-shakespeares-sonnet/data

and experiment with an exponentially increasing number of clusters. Compute the reconstruction error for each setting on this module across all LoRAs—a computationally efficient process. Choose the minimal number of clusters that achieves a reconstruction loss below 0.5, and then use these settings to compress all LoRA modules. An example of this procedure applied to 500 LoRAs is illustrated in Figure 5 in the Appendix.

We note that tuning hyperparameters as discussed above using reconstruction loss as a validation metric is especially convenient since it can be done efficiently on CPU without having to perform expensive LLM evaluation. As our experiments demonstrate, compression settings that achieve below 0.5 reconstruction loss reliably translate into preserving 99% or more of the LoRA performance, sometimes even outperforming the original LoRAs.

For inference, this procedure is executed as a preprocessing step before deploying our inference server. As new LoRAs are submitted, they are initially served uncompressed. A background cron job re-runs the compression algorithm on the CPU every six hours, and upon completion, updates the served LoRA parameters with the compressed versions.

## 7 DISCUSSION

This study introduces approaches to LoRA compression, addressing significant challenges facing foundation models and large language models. Our contributions include theoretical formulations, empirical validation, and practical implementations that enhance the understanding and application of LLMs in scalable environments.

The implications of our findings are manifold. Our theoretical guarantees for reconstruction error not only increase confidence in the use of compressed models but also lay a groundwork for future explorations in this area. Demonstrating that our compression techniques can preserve up to 100% of the original LoRAs' performance highlights the effectiveness of our methods. Furthermore, integrating LoRA compression into state-of-the-art LLM serving systems demonstrates the potential for resource optimization, with throughput for thousands of LoRAs nearing that of a single LoRA.

The promising results of our study suggest several future research directions. First, further compression may be possible via quantization. Our joint-diagonalization compression and quantization are independent axes of approaching the problem and exploring a combined solution can be fruitful. Second, when scaling to hundreds of thousands of LoRAs, joint compression, while effective, will not be sufficient to fit all LoRAs onto the GPU, thus requiring a procedure to schedule the requests. Our clustering variant offers opportunities to develop an efficient scheduling mechanism that takes into account the cluster assignments of LoRAs corresponding to the incoming requests.

In conclusion, our research significantly advances the deployment of LLMs by providing robust, scalable, and efficient compression solutions. The ability of compressed LoRAs to maintain high performance while facilitating substantial resource savings opens new avenues for the broader application and adoption of LLMs across various industries. We encourage the community to build upon our findings and the shared LoRAs to further explore and enhance the utility of these technologies.

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

## A   Joint Diagonalization Algorithms

### A.1   Alternating Methods

Our goal is to derive algorithms that optimize equation 1. Common to both methods, we expand the objective functional:

$$\sum_i \|B_i A_i - U\Sigma_i V^\top\|_{\text{Fro}}^2 = \sum_i \text{tr}((B_i A_i - U\Sigma_i V^\top)(B_i A_i - U\Sigma_i V^\top)^\top) \text{ by definition}$$

$$= \sum_i \left[\text{tr}(B_i A_i A_i^\top B_i^\top) - 2\text{tr}(B_i A_i V\Sigma_i^\top U^\top) + \text{tr}(U\Sigma_i V^\top V\Sigma_i^\top U^\top)\right]$$

$$= \text{const.} - 2\sum_i \text{tr}(B_i A_i V\Sigma_i^\top U^\top) + \sum_i \|U\Sigma_i V^\top\|_{\text{Fro}}^2. \tag{5}$$

Using this expansion, we now consider the two settings discussed in §3.1.

**Case 1: Non-diagonal $\Sigma_i$, orthogonal $U, V$.** Setting the derivative of equation 5 with respect to $\Sigma_i$ to zero, we find

$$\Sigma_i = \Sigma_i^*(U, V) = U^\top B_i A_i V. \tag{6}$$

We simplify our objective function after plugging in this expression:

$$\sum_i \|B_i A_i - U\Sigma_i V^\top\|_{\text{Fro}}^2 + \text{const.} = \sum_i \left[\|\Sigma_i\|_{\text{Fro}}^2 - 2\text{tr}(B_i A_i V\Sigma_i^\top U^\top)\right] \text{ from equation 5}$$

$$= \sum_i \left[\text{tr}(U^\top B_i A_i VV^\top A_i^\top B_i^\top U) - 2\text{tr}(B_i A_i VV^\top A_i^\top B_i^\top UU^\top)\right] \text{ from equation 6}$$

$$= -\sum_i \text{tr}(B_i A_i VV^\top A_i^\top B_i^\top UU^\top).$$

Substituting equation 6, we find

$$U_{opt}, V_{opt} = \arg\max_{\substack{U^\top U=I \\ VV^\top=I}} \sum_{i=1}^n \|U^\top B_i A_i V\|_{\text{Fro}}^2 = \arg\max_{\substack{U^\top U=I \\ VV^\top=I}} \sum_{i=1}^n \|\Sigma_i^*(U, V)\|_{\text{Fro}}^2. \tag{7}$$

Note that

$$\sum_{i=1}^n \|U^\top B_i A_i V\|_{\text{Fro}}^2 = \text{tr}\left(\left(\sum_{i=1}^n B_i A_i VV^\top A_i^\top B_i^\top\right) UU^\top\right)$$

$$= \text{tr}\left(\left(\sum_{i=1}^n B_i^\top A_i^\top UU^\top A_i B_i\right) VV^\top\right),$$

by the identity $\|A\|_{\text{Fro}}^2 = \text{tr}(A^\top A)$. Hence, we optimize equation 7 by alternating between $U$ and $V$:

- $U$ **iteration:** Define $M := \sum_i B_i A_i VV^\top A_i^\top B_i^\top$. Parenthesizing this expression properly requires only $O((m + n)r)$ storage/computation time. With this definition, we maximize $\text{tr}(MUU^\top)$ over $U$ satisfying $U^\top U = I$. Since $M$ is positive semidefinite, the optimum is to take $U$ to be the $r$ eigenvectors of $M$ with largest eigenvalue, equivalent to an SVD problem.
- $V$ **iteration:** Define $N := \sum_i A_i^\top B_i^\top UU^\top B_i A_i$. Similarly to the previous step, we take $V$ to contain the $r$ eigenvectors of $N$ with largest eigenvalue, again solvable using an SVD.

This method decreases the objective in each step.

**Case 2: Diagonal $\Sigma_i$.** If constrain $\Sigma_i$ to be diagonal, we interpret our objective function equation 1 as a "triple least squares" problem. We compute gradients:

$$\nabla_U \sum_i \|B_i A_i - U\Sigma_i V^\top\|_{\text{Fro}}^2 = 2\sum_i (U\Sigma_i V^\top - B_i A_i)V\Sigma_i^\top$$

$$\nabla_V \sum_i \|B_i A_i - U\Sigma_i V^\top\|_{\text{Fro}}^2 = 2\sum_i (V\Sigma_i^\top U^\top - A_i^\top B_i^\top)U\Sigma_i$$

$$\nabla_{\Sigma_i} \sum_i \|B_i A_i - U\Sigma_i V^\top\|_{\text{Fro}}^2 = 2U^\top(U\Sigma_i V^\top - B_i A_i)V$$

These expressions suggest efficient $r \times r$ linear systems to solve for $U, V$:

$$U = \left(\sum_i B_i A_i V\Sigma_i^\top\right)\left(\sum_i \Sigma_i V^\top V\Sigma_i^\top\right)^{-1}$$

$$V = \left(\sum_i A_i^\top B_i^\top U\Sigma_i\right)\left(\sum_i \Sigma_i^\top U^\top U\Sigma_i\right)^{-1}.$$

For $\Sigma_i$, we extract the diagonal from our gradient above:

$$\begin{aligned}
\text{diag}(U^\top U\Sigma_i V^\top V)_j &= (U^\top U\Sigma_i V^\top V)_{jj} \\
&= \sum_m (U^\top U)_{jm}\Sigma_{imm}(V^\top V)_{mj} \\
&= (U^\top U \circ V^\top V)\text{diag}(\Sigma_i) \\
\text{diag}(U^\top B_i A_i V)_j &= \sum_m (U^\top B_i)_{jm}(A_i V)_{mj} \\
&= \sum_m (U^\top B_i)_{jm}(V^\top A_i^\top)_{jm} \\
&= (U^\top B_i \circ V^\top A_i^\top)\mathbf{1} \\
\implies \text{diag}(\Sigma_i) &= (U^\top U \circ V^\top V)^{-1}(U^\top B_i \circ V^\top A_i^\top)\mathbf{1}
\end{aligned}$$

Here $\circ$ denotes the Hadamard product.

Combining these expressions, we use a simple coordinate descent algorithm cycling between the following three steps:

1. Solve for $U$
2. Solve for $V$
3. Solve for the $\Sigma_i$'s
4. Optionally, normalize so $\sum_i \|\Sigma_i\|_{\text{Fro}}^2 = 1$

## A.2 Additional Eigenvalue Iteration Algorithm

For the first case in §A.1, we introduce an alternative algorithm that eschews the use of SVD. This alternative is optimized for GPU execution, enabling tractable runs to convergence.

To derive this algorithm, we employ Lagrange multipliers to formulate the derived objective from equation 7:

$$U_{opt}, V_{opt} = \arg\max_{\substack{U^\top U = I \\ VV^\top = I}} \sum_{i=1}^n \|U^\top B_i A_i V\|_{\text{Fro}}^2, \tag{8}$$

yielding the expression

$$\Lambda = -\frac{1}{2}\|U^\top B_i A_i V\|_{\text{Fro}}^2 - \frac{1}{2}\text{tr}(X^\top(I - U^\top U)) - \frac{1}{2}\text{tr}(Y^\top(I - V^\top V)). \tag{9}$$

Taking the derivatives gives

$$\nabla_U \Lambda = -\sum_i B_i(A_i V)(V^\top A_i^\top)(B_i^\top U) + UX \tag{10}$$

$$\nabla_V \Lambda = -\sum_i A_i^\top (B_i^\top U)(U^\top B_i)(A_i V) + VY \tag{11}$$

Setting these derivatives to zero shows

$$\sum_i B_i(A_i V)(V^\top A_i^\top)(B_i^\top U) = UX \tag{12}$$

$$\sum_i A_i^\top (B_i^\top U)(U^\top B_i)(A_i V) = VY. \tag{13}$$

Here, one can show that the Lagrange multiplier matrices $X$ and $Y$ are diagonal and nonnegative, since the problem reduces to an eigenvalue problem when either $U$ or $V$ is fixed; this is essentially the argument behind the alternating algorithm in Appendix A. Hence, taking inspiration from classical eigenvalue iteration, we use the following updates to improve our estimates of $U$ and $V$:

$$U_0^{(k+1)} \leftarrow \sum_i B_i(A_i V^{(k)})((V^{(k)})^\top A_i^\top)(B_i^\top U^{(k)}) \tag{14}$$

$$V_0^{(k+1)} \leftarrow \sum_i A_i^\top (B_i^\top U^{(k)})((U^{(k)})^\top B_i)(A_i V^{(k)}) \tag{15}$$

$$U^{(k+1)} \leftarrow \texttt{orthogonalize}(U_0^{(k+1)}) \tag{16}$$

$$V^{(k+1)} \leftarrow \texttt{orthogonalize}(V_0^{(k+1)}) \tag{17}$$

Here, the function $\texttt{orthogonalize}$ orthogonalizes the columns of a matrix, e.g. by using the $Q$ part of the reduced-size $QR$ factorization. Although we lack a formal convergence proof, in practice we find that this method reliably reaches a local optimum of our problem.

By executing matrix operations in the specified sequence, these computations can be rapidly performed on GPUs. Note the expressions above are parenthesized to avoid constructing a large matrix product as an intermediate computation.

### A.3 CLUSTERING ALGORITHM

**Initialization**: We run joint diagonalization with a single $U, V$ then perform k-means with $|C_j|$ clusters on the space of $\Sigma_i$'s. This gives us our first clusters and we can use random initialization $U_j, V_j$ for each cluster but the $\Sigma_i$ can be maintained as initialization.

**Step 1**: Using the alternating JD algorithms from earlier in this section, we optimize the problem $\min_{U_j, V_j, \Sigma_i} \sum_{i \in C_j} ||B_i A_i - U_j \Sigma_i V_j^\top||_F^2$ for each $j$ independently.

**Step 2**: New cluster assignment for $i$ : $\min_j \min_{\Sigma_i} ||B_i A_i - U_j \Sigma_i V_j^\top||_F^2$. If any assignment changes we go to Step 1, else we have converged.

## B  PROOF OF THEOREM 1

*Proof.* For the lower bound, note that by Jensen's inequality,

$$\sum_{i=1}^n \|U^\top B_i A_i V\|_{\text{Fro}}^2 \geq \left\| U^\top \sum_{i=1}^n B_i A_i V \right\|_{\text{Fro}}^2,$$

for any $U, V$. Hence,

$$\sup_{U, V \in \text{St}(k,d)} \sum_{i=1}^n \|U^\top B_i A_i V\|_{\text{Fro}}^2 \geq \sup_{U, V \in \text{St}(k,d)} \left\| U^\top \sum_{i=1}^n B_i A_i V \right\|_{\text{Fro}}^2. \tag{18}$$

By the definition of singular value decomposition, the right hand side of equation 18 is maximized with $U, V$ being the top $r$ singular vectors of $\sum_{i=1}^{n} B_i A_i$, yielding $\left\| U^\top \sum_{i=1}^{n} B_i A_i V \right\|_{\text{Fro}}^2 = \sum_{i=1}^{r} \bar{\sigma}_i^2$. Recalling that $\Sigma_i = U^\top B_i A_i V$ yields the lower bound.

For the upper bound, recall that $\Sigma_i = U^\top B_i A_i V$. Rearranging,

$$\text{vec}(\Sigma_i) = (V^\top \otimes U^\top)\text{vec}(B_i A_i).$$

Define

$$\bar{\Sigma} := [\text{vec}(\Sigma_1), \dots, \text{vec}(\Sigma_n)].$$

By our previous simplification,

$$\bar{\Sigma} = (V^\top \otimes U^\top)L.$$

Now

$$\sum_{i=1}^{n} \|\Sigma_i\|_{\text{Fro}}^2 = \|\bar{\Sigma}\|_{\text{Fro}}^2 = \text{tr}\left(((V \otimes U)(V \otimes U)^\top)(LL^\top)\right)$$

Since $U, V$ are orthogonal and size $d \times r$, the top $r^2$ eigenvalues of the symmetric matrix $(V \otimes U)(V \otimes U)^\top$ will be equal to 1, and the rest will equal 0. The eigenvalues of the symmetric matrix $LL^\top$ will be equal to the squared singular values of $L$. We can then apply the Von Neumann trace inequality to obtain the upper bound.

The last statement follows from the Pythagorean theorem and the fact that the $\Sigma_i$ is a projection of $B_i A_i$ to the $U, V$ subspace. $\square$

Note that we have only used the fact that the matrix $(V \otimes U)$ has singular values equal to 1; we have not used the fact that it has Kronecker product structure. On the other hand, each vector $\text{vec}(B_i A_i)$ is a sum of $r_i$ Kronecker products and cannot be expressed as a Kronecker product. As a result, while the upper bound in the Von Neumann trace inequality is achieved if the eigenvectors of the two matrices align, the Kronecker product structure is a severe constraint and the upper bound we have provided is generous.

## C  TRAINING LoRAs

We trained LoRA adapters on 500 natural instruction tasks (Wang et al., 2022) using `Mistral-7B-Instruct-v0.2` (Jiang et al., 2023a) as the base model. All LoRA adapters were configured with a rank of 16, i.e., $\forall i, r_i = 16$. We selected 10 diverse tasks manually for consistent evaluation across experiments and randomly sampled an additional 490 tasks, resulting in a total of 500 tasks. These tasks were exclusively in English (both input and output), ensuring higher quality and thorough review (Wang et al., 2022). Each task dataset was divided into training, validation, and test sets (80-10-10). Hyperparameters, such as early stopping, were tuned using the validation sets; that is, we train for five epochs and take the best-performing epoch-checkpoint per validation loss. Evaluation on the test sets demonstrated that LoRA consistently outperformed the base model in terms of both Rouge scores and loss metrics (see Table 1).

Table 2: Main Evaluation Tasks

| Task Number | Name | Type | Domain |
|---|---|---|---|
| task280 | stereoset_classification_stereotype_type | classification | stereoset |
| task190 | snli_classification | snli | image captions |
| task391 | causal_relationship | commonsense | cause and effect |
| task290 | tellmewhy_question_answerability | answerability | story |
| task1391 | winogrande_easy_answer_generation | commonsense | social and physical |
| task1342 | amazon_us_reviews_title | title generation | amazon reviews |
| task442 | com_qa_paraphrase_question_generation | question generation | wikipedia |
| task620 | ohsumed_medical_subject_headings_answer_generation | keyword tagging | scientific |
| task1598 | nyc_long_text_generation | data to text | restaurants |
| task039 | qasc_find_overlapping_words | overlap extraction | natural science |

In Table 3 we include all 500 tasks that were used.

Table 3: List of Tasks

| Task ID | Description | Task ID | Description | Task ID | Description |
|---|---|---|---|---|---|
| task280 | stereoset classification stereotype type | task190 | snli classification | task391 | causal relationship |
| task290 | tellmewhy question answerability | task1391 | winogrande easy answer generation | task1342 | amazon us reviews title |
| task442 | com qa paraphrase question generation | task620 | ohsumed medical subject headings answer generation | task1598 | nyc long text generation |
| task039 | qasc find overlapping words | task769 | qed summarization | task1448 | disease entity extraction ncbi dataset |
| task247 | dream answer generation | task513 | argument stance classification | task875 | emotion classification |
| task515 | senteval odd word out | task627 | xlwic same meaning sentence generation | task1534 | daily dialog question classification |
| task1551 | every ith element from kth element | task583 | udeps eng coarse pos tagging | task1431 | head qa answer generation |
| task270 | csrg counterfactual context generation | task1487 | organism substance extraction anem dataset | task679 | hope edi english text classification |
| task456 | matres intention classification | task385 | socialiqa incorrect answer generation | task1607 | ethos text classification |
| task278 | stereoset antistereotype sentence generation | task022 | cosmosqa passage inappropriate binary | task210 | logic2text structured text generation |
| task137 | detoxifying-lms classification toxicity | task574 | air dialogue sentence generation | task629 | dbpedia 14 classification |
| task1378 | quarel correct answer generation | task1194 | kth largest element | task1529 | scitail1.1 classification |
| task453 | swag answer generation | task102 | commongen sentence generation | task460 | qasper answer generation |
| task1204 | atomic classification hinderedby | task1384 | deal or no dialog classification | task1572 | samsum summary |
| task699 | mmmlu high school biology answer generation | task1631 | openpi answer generation | task1722 | civil comments threat classification |
| task580 | socialiqa answer generation | task605 | longest common subsequence in lists | task1152 | bard analogical reasoning causation |
| task1283 | hrngo quality classification | task637 | extract and sort unique digits in a list | task723 | mmmlu moral disputes answer generation |
| task084 | babi t1 supporting fact identification | task201 | mnli neutral classification | task956 | leetcode strong password check |
| task167 | strategyqa question generation | task1192 | food flavor profile | task300 | storycloze order generation |
| task1714 | convai3 sentence generation | task388 | torque token classification | task516 | senteval conjoints inversion |
| task127 | scan action command all generation | task362 | spolin yesand response classification | task1158 | bard analogical reasoning manipulating items |
| task322 | jigsaw threat classification | task697 | mmmlu formal logic answer generation | task1566 | propara structured text generation |
| task076 | splash correcting SQL mistake | task1451 | drug dose extraction | task1135 | xcsr en commonsense mc classification |
| task341 | winomt gender anti classification | task267 | concatenate and reverse elements from i to j | task1720 | civil comments toxicity classification |
| task1452 | location entity extraction btc corpus | task131 | scan action command long generation | task685 | mmmlu clinical knowledge answer generation |
| task727 | mmmlu prehistory answer generation | task1590 | diplomacy text generation | task1731 | quartz question answering |
| task047 | answering science questions | task929 | products reviews classification | task1592 | yahoo answers topics classification |
| task1326 | qa zre question generation from answer | task615 | moviesqa answer generation | task1216 | atomic classification causes |
| task689 | mmmlu college mathematics answer generation | task1156 | bard analogical reasoning tools | task1657 | gooaq question generation |
| task833 | poem sentiment classification | task1206 | atomic classification isbefore | task1151 | swap max min |
| task244 | count elements in set union | task1562 | zest text modification | task1043 | essential terms answering incomplete questions |
| task044 | essential terms identifying essential words | task722 | mmmlu random topic answer generation | task183 | rhyme generation |
| task563 | discofuse answer generation | task155 | count nouns and verbs | task353 | casino negotiation elicit preference classification |
| task616 | cola classification | task1724 | civil comments insult classification | task288 | gigaword summarization |
| task092 | check prime classification | task707 | mmmlu high school microeconomics answer generation | task577 | curiosity dialogs classification |
| task742 | lhoestq frequency answer generation | task706 | mmmlu high school mathematics answer generation | task1401 | obqa sentence generation |
| task1393 | superglue copa text completion | task1198 | atomic classification owant | task966 | ruletaker fact checking from context |
| task219 | rocstories title answer generation | task1211 | atomic classification hassubevent | task050 | multirc answerability |
| task494 | review polarity answer generation | task1379 | quarel incorrect answer generation | task176 | break decompose questions |
| task068 | abductivenli incorrect answer generation | task566 | circa classification | task333 | hateeval hate classification en |
| task593 | sciq explanation generation | task667 | mmmlu business ethics answer generation | task130 | scan action command long generation |
| task161 | count words containing letter | task507 | position of numerical elements in list | task1502 | hatexplain classification |
| task505 | count numerical elements in list | task633 | dbpedia 14 answer generation | task1645 | medical question pair classification |
| task1486 | cell extraction anem dataset | task1146 | country capital | task1380 | quarel correct option generation |
| task1088 | array of products | task033 | winogrande answer generation | task085 | unnatural addsub arithmetic |
| task1294 | wiki qa answer verification | task080 | piqa answer generation | task489 | mwsc question generation |
| task1721 | civil comments obscenity classification | task1713 | convai3 sentence generation | task721 | mmmlu medical genetics answer generation |
| task1403 | check validity date mmddyyyy | task746 | yelp restaurant review classification | task728 | mmmlu professional accounting answer generation |
| task889 | goemotions classification | task1583 | bless meronym classification | task1665 | trianglecopa question generation |
| task708 | mmmlu high school physics answer generation | task1419 | mathqa gain | task963 | librispeech asr next word prediction |
| task454 | swag incorrect answer generation | task308 | jeopardy answer generation all | task828 | copa cause effect commonsense |
| task579 | socialiqa classification | task753 | svamp addition question answering | task1404 | date conversion |
| task1201 | atomic classification xintent | task901 | freebase qa category question generation | task1567 | propara question generation |
| task1319 | country by barcode prefix | task858 | inquisitive span detection | task1200 | atomic classification xeffect |
| task492 | mwsc incorrect answer generation | task675 | google wellformed query sentence generation | task094 | conala calculate mean |
| task1506 | celebrity minimal dob span | task694 | mmmlu econometrics answer generation | task614 | glucose cause event detection |
| task1390 | wscfixed coreference | task1355 | sent comp summarization | task714 | mmmlu human sexuality answer generation |
| task457 | matres conditional classification | task1565 | triviaqa classification | task834 | mathdataset classification |
| task642 | esnli classification | task732 | mmmlu public relations answer generation | task1605 | ethos text classification |
| task326 | jigsaw obscene classification | task1292 | yelp review full text categorization | task716 | mmmlu jurisprudence answer generation |
| task1479 | organization entity extraction btc corpus | task1147 | country currency | task153 | tomqa find location hard clean |
| task1495 | adverse drug event classification | task1196 | atomic classification oeffect | task1489 | sarcasmdetection tweet classification |
| task294 | storycommonsense motiv text generation | task157 | count vowels and consonants | task147 | afs argument similarity gay marriage |
| task1197 | atomic classification oreact | task754 | svamp common-division question answering | task1599 | smcalflow classification |
| task1420 | mathqa general | task285 | kpa keypoint matching | task587 | amazonfood polarity correction classification |
| task1338 | peixian equity sentiment classifier | task116 | com2sense commonsense reasoning | task713 | mmmlu human aging answer generation |
| task431 | senteval object count | task067 | abductivenli answer generation | task934 | turk simplification |
| task617 | amazonreview category text generation | task696 | mmmlu elementary mathematics answer generation | task846 | pubmedqa classification |
| task933 | wiki auto style transfer | task865 | mawps addsub question answering | task671 | ambigqa text generation |
| task1398 | obqa question generation | task1518 | limit answer generation | task628 | xlwic different meaning sentence generation |
| task1286 | openbookqa question answering | task1596 | event2mind text generation 2 | task298 | storycloze correct end classification |
| task645 | summarization | task903 | deceptive opinion spam classification | task594 | sciq question generation |
| task413 | mickey en sentence perturbation generation | task719 | mmmlu management answer generation | task672 | nummersense |
| task1418 | bless semantic relation classification | task475 | yelp polarity classification | task357 | casino negotiation small talk classification |
| task1387 | anli r2 entailment | task1711 | poki text generation | task304 | numeric fused head resolution |
| task750 | aqua multiple choice answering | task1320 | country domain tld | task034 | winogrande question modification object |
| task692 | mmmlu computer security answer generation | task1456 | kth smallest element | task119 | semeval 2019 task10 geometric mathematical answer |
| task211 | logic2text classification | task363 | sst2 polarity classification | task1087 | two number sum |
| task083 | babi t1 answer generation | task1385 | anli r1 entailment | task1308 | amazonreview category classification |
| task1656 | gooaq answer generation | task892 | gap reverse coreference resolution | task499 | extract and add numbers from list |
| task1409 | dart text generation | task1207 | atomic classification atlocation | task564 | discofuse classification |
| task325 | jigsaw identity attack classification | task206 | collatz conjecture | task890 | gcwd classification |
| task1520 | qa srl answer generation | task703 | mmmlu high school geography answer generation | task318 | stereoset gender classification |
| task366 | synthetic return primes | task335 | hateeval aggressive classification en | task600 | longest common substring in two strings |
| task477 | cls english dvd classification | task138 | detoxifying-lms classification fluency | task291 | semeval 2020 task4 commonsense validation |
| task074 | squad1.1 question generation | task1389 | hellaswag completion | task192 | hotpotqa sentence generation |
| task316 | crows-pairs stereotype classification | task1609 | xquad en question generation | task296 | storycloze correct end classification |
| task666 | mmmlu astronomy answer generation | task1582 | bless hypernym generation | task1728 | web nlg data to text |
| task701 | mmmlu high school computer science answer generation | task275 | enhanced wsc paraphrase generation | task107 | splash question to SQL |
| task079 | conala concat strings | task1157 | bard analogical reasoning rooms for containers | task1167 | penn treebank coarse pos tagging |
| task403 | creak commonsense inference | task359 | casino negotiation vouch fair classification | task517 | emo classify emotion of dialogue |
| task351 | winomt gender identifiability anti classification | task964 | librispeech asr text auto completion | task904 | hate speech offensive classification |
| task1148 | maximum ascii value | task879 | schema guided dstc8 classification | task636 | extract and sort unique alphabets in a list |
| task1509 | evalution antonyms | task207 | max element lists | task228 | arc easy answer generation |

We use Huggingface (Wolf et al., 2020) in our implementation. For the base model, we use quantization with configuration:

```
BitsAndBytesConfig(
    load_in_4bit=True,
    bnb_4bit_use_double_quant=True,
    bnb_4bit_quant_type="nf4",
```

```
            bnb_4bit_compute_dtype=torch.bfloat16,
        )
```

and LoRA configuration:

```
    LoraConfig(
        r=16,
        lora_alpha=32,
        target_modules=["q_proj", "k_proj", "v_proj"],
        lora_dropout=0.05,
        bias="none",
        task_type="CAUSAL_LM",
        init_lora_weights=init_lora_weights,
    )
```

## D  AVOIDING BATCHED MATRIX MULTIPLICATION (BMM)

Fast LoRA (Wen & Chaudhuri, 2024) aims to alleviate the batched matrix multiplication (BMM) bottleneck when serving many LoRAs. They propose an adapter parameterization that replaces addition with elementwise multiplication, avoiding BMM and improving LoRA throughput at lower ranks. Our JD LoRA formulation also circumvents or heavily reduces the impact of BMM as discussed below, and both individual and joint compression methods can be applied to Fast LoRAs.

In the envisioned deployment scenario, a service provider hosts a large collection of LoRAs. Upon receiving a request, each user specifies both the input data and the desired LoRA identifier. The provider then processes the base model augmented with the specified LoRA for each user's data. As a provider is batching a collection of requests for GPU parallelization, they can expect to frequently have more than one unique LoRA identifier per batch.

Traditionally, a specific LoRA is integrated into the base model by transforming $W_0 \to W_0 + B_i A_i$. Serving multiple LoRAs conventionally would necessitate maintaining and executing a separate copy of the base model for each LoRA, bringing substantial computational overhead. Alternatively, the computation for $W_0 x$ and $B_i A_i x$ can be performed independently and subsequently merged. This strategy necessitates only a single instance of $W_0 x$ computation and storage of LoRA-specific parameters rather than the entire base model.

Consider the batch processing of $\mathbf{BAx}$, where boldface indicates that $B_i, A_i$ are stacked into tensors of dimensions $(b \times m \times r)$ and $(b \times r \times n)$ respectively, with batched data $\mathbf{x}$ shaped $(b \times l \times n)$:

$$\mathbf{Ax} \leftrightarrow (b \times r \times n) \times (b \times l \times n) \to (b \times l \times r) \text{ bmm}$$
$$\mathbf{B}(\mathbf{Ax}) \leftrightarrow (b \times m \times r) \times (b \times l \times r) \to (b \times l \times m) \text{ bmm}.$$

Here, "bmm" denotes batched matrix multiplication, a known bottleneck in both throughput and latency. Consider the corresponding operations for our joint compression scheme, $U\Sigma V^\top x$:

$$V^\top \mathbf{x} \leftrightarrow (\tilde{r} \times n) \times (b \times l \times n) \to (b \times l \times \tilde{r}) \text{ broadcasted}$$
$$\mathbf{\Sigma}(V^\top \mathbf{x}) \leftrightarrow (b \times \tilde{r}) \times (b \times l \times \tilde{r}) \to (b \times l \times \tilde{r}) \text{ broadcasted}$$
$$U(\mathbf{\Sigma} V^\top \mathbf{x}) \leftrightarrow (m \times \tilde{r}) \times (b \times l \times \tilde{r}) \to (b \times l \times m) \text{ broadcasted}$$

In our optimized setup, batched matrix multiplications can be completely circumvented if the $\Sigma_i$ matrices are diagonal. If not, given that $\tilde{r} \ll m, n$, any required batched matrix multiplication remains computationally inexpensive.

## E  GPU MEMORY USAGE COMPUTATION FOR JD COMPRESSION

The GPU memory consumption is primarily influenced by the number of parameters that need to be stored and processed during inference. In this section, we introduce the detail of how we compute the GPU consumption of our method, and how we find the number of vLLM multi-LoRA that share the same GPU utilization.

- $D$: Hidden dimension size (e.g., $D = 4098$).
- $r$: Rank of the shared basis matrices for compression (e.g., $r = 16, 32, 64$).
- $N$: Maximum number of LoRA modules being served simultaneously (max_lora_num).
- $c$: Number of clusters in our clustering method (e.g., $c = 7, 10, 25$).

In Figure 1, we use different JD-compression settings for serving different number of unique LoRAs. Specifically:

- **Serving 4 unique LoRAs:**
  Ours: rank 16 JD-Full.
  vLLM multiLoRA baseline: max-gpu-lora = 2.
- **Serving 8 unique LoRAs:**
  Ours: rank 16 JD-Full.
  vLLM multiLoRA baseline: max-gpu-lora = 2.
- **Serving 16 unique LoRAs:**
  Ours: rank 32 JD-Full.
  vLLM multiLoRA baseline: max-gpu-lora = 3.
- **Serving 32 unique LoRAs:**
  Ours: rank 64 JD-Full.
  vLLM multiLoRA baseline: max-gpu-lora = 5.
- **Serving 64 unique LoRAs:**
  Ours: rank 64 JD-Full.
  vLLM multiLoRA baseline: max-gpu-lora = 6.
- **Serving 128 unique LoRAs:**
  Ours: 7 clusters, rank 16 JD-Full.
  vLLM multiLoRA baseline: max-gpu-lora = 8.
- **Serving 256 unique LoRAs:**
  Ours: 10 clusters, rank 16 JD-Full.
  vLLM multiLoRA baseline: max-gpu-lora = 10.
- **Serving 512 unique LoRAs:**
  Ours: 25 clusters, rank 16 JD-Full.
  vLLM multiLoRA baseline: max-gpu-lora = 26.
- **Serving 1024 unique LoRAs:**
  Ours: 7 clusters, rank 64 JD-Full.
  vLLM multiLoRA baseline: max-gpu-lora = 60.

### E.1 BASELINE GPU MEMORY USAGE

The baseline for our comparison is the standard LoRA method with a rank of 16. The total parameter count for the baseline is given by:

$$\text{Params}_{\text{baseline}} = D \times 2 \times 16.$$

This accounts for the parameters in the LoRA-adapted layers, where the factor of 2 represents the weights and biases.

### E.2 GPU MEMORY USAGE FOR JD FULL METHOD

For the Joint Decomposition (JD) Full method without clustering, the total parameter count is:

$$\text{Params}_{\text{JD\_Full}} = D \times 2 \times r + N \times r^2.$$

- $D \times 2 \times r$: Parameters for the base model adapted with rank-$r$ LoRA.
- $N \times r^2$: Additional parameters introduced by each of the $N$ LoRA modules, each of size $r \times r$.

The GPU memory usage ratio relative to the baseline is:

$$\text{GPU Usage Ratio}_{\text{JD\_Full}} = \frac{\text{Params}_{\text{JD\_Full}}}{\text{Params}_{\text{baseline}}} = \frac{D \times 2 \times r + N \times r^2}{D \times 2 \times 16}.$$

### E.3 GPU MEMORY USAGE FOR CLUSTERING METHOD

When employing clustering, the parameter count changes due to the addition of cluster-specific parameters:

$$\text{Params}_{\text{Clustering}} = D \times 2 \times r \times c + N \times (r^2 + 1).$$

- $D \times 2 \times r \times c$: Parameters for the base model adapted with rank-$r$ LoRA across $c$ clusters.
- $N \times (r^2 + 1)$: Additional parameters for each LoRA module and cluster assignments.

The GPU memory usage ratio is:

$$\text{GPU Usage Ratio}_{\text{Clustering}} = \frac{\text{Params}_{\text{Clustering}}}{\text{Params}_{\text{baseline}}} = \frac{D \times 2 \times r \times c + N \times (r^2 + 1)}{D \times 2 \times 16}.$$

### E.4 PUNICA

In our `vLLM` experiments, we specifically utilized the Punica kernel for implementing multi-LoRA, applying our approach in conjunction with Punica's capabilities. Our custom function, `add_lora_slice_with_sigma`, implements the following key steps:

1. **Initialize Buffers**: Creates temporary storage for intermediate calculations if not already provided.
2. **Apply Matrix A**: Transforms `x` using matrix `A`, storing the result in `buffer`.
3. **Apply Matrix Sigma**: Further transforms `buffer` using `Sigma`, storing the result in `buffer_sigma`.
4. **Apply Matrix B and Update y**: Finally, transforms `buffer_sigma` using `B`, applies scaling, and updates a slice of `y` in place.

Below is the pseudocode for `add_lora_slice_with_sigma`, illustrating the integration:

Listing 1: Pseudocode for 'add_lora_slice_with_sigma'

```
Function add_lora_slice_with_sigma(y, x, wa_t_all, wb_t_all, wsigma_t_all
    , indices, layer_idx, scale, y_offset, y_slice_size, buffer=None):
    # Initialize buffers if not provided
    if buffer is None:
        buffer = create_tensor(shape=(x.size(0), R), dtype=float32)
        buffer_sigma = create_tensor(shape=(buffer.size(0), R), dtype=
            float32)
    # Step 1: Apply matrix A
    dispatch_bgmv_low_level(buffer, x, wa_t_all, indices, layer_idx,
        scale=1.0)
    # Step 2: Apply matrix Sigma
    dispatch_bgmv_low_level(buffer_sigma, buffer, wsigma_t_all, indices,
        layer_idx, scale=1.0)
    # Step 3: Apply matrix B and update y slice
    dispatch_bgmv_low_level(y, buffer_sigma, wb_t_all, indices, layer_idx
        , scale, y_offset, y_slice_size)
End Function
```

## F SELECTING NUMBER OF CLUSTERS

To identify optimal hyperparameters for the clusters compression method, we analyzed the relationship between reconstruction error and the parameter saved ratio for a single LoRA module, as shown in Figure 5. By comparing the results across different numbers of Low-Rank Adaptation (LoRA) configurations (100 and 500, depicted in subfigures 5a and 5b), we were able to observe the trade-off between model size reduction and reconstruction accuracy. Based on these findings, we selected the rank and number-of-clusters hyperparameters that effectively balance these two objectives. The chosen settings were then used to conduct full-scale experiments.

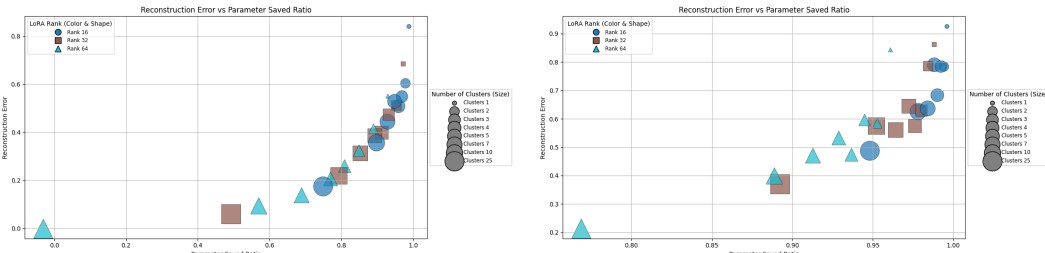

(a) Recon. Error vs Parameter Saved Ratio for 100 LoRAs

(b) Recon. Error vs Parameter Saved Ratio for 500 LoRAs

Figure 5: Comparison of reconstruction error against the parameter saved ratio for different numbers of LoRA configurations for a single LoRA module. The left subplot shows results for 100 LoRAs, while the right subplot displays results for 500 LoRAs. These plots illustrate the trade-off between reconstruction accuracy and compression efficiency, providing insights into optimal parameter settings for compression.

## G  ADDITIONAL RESULTS

This section elaborates on the results that underpin the figures presented in the main text and showcases a consistent correlation across various evaluation metrics. Additionally, we assess the significance of achieving convergence and the performance of compression on new unseen LoRA models.

### G.1  RELATIVE ROUGE-L PERFORMANCE AND COMPRESSION RATE

Table 4 presents comprehensive results from the experiments underlying Figure 2a for each evaluation task. Additionally, we incorporate results using the Ties-merging benchmark (Yadav et al., 2023b), which consolidates all LoRA-adapters into a single adapter of identical configuration and parameter count; this integration significantly compromises performance.

### G.2  ABSOLUTE ROUGE-L PERFORMANCE AND COMPRESSION RATE

Table 5 provides the full results behind Table 4, but with Rouge-L scores instead of relative performance compared to LoRA.

### G.3  RELATIVE ROUGE-1 PERFORMANCE AND COMPRESSION RATE

Table 6 provides full results for relative performance of Rouge-1, which shows the same trends as the results for relative performance of Rouge-L (Table 4).

### G.4  ABSOLUTE ROUGE-1 PERFORMANCE AND COMPRESSION RATE

Table 7 provides full results for absolute performance of Rouge-1, which shows the same trends as the results for absolute performance of Rouge-L (Table 5).

### G.5  RELATIVE EXACT-MATCH PERFORMANCE AND COMPRESSION RATE

Table 8 provides full results for relative performance of exact-match, which shows the same trends as the results for relative performance of Rouge-L (Table 4).

### G.6  LOSS AND COMPRESSION RATE

Table 9 provides full results for test loss (cross-entropy), which shows the same trends as the results for relative performance of Rouge-L (Table 4).

| Model Type | Method Type | task039 | task190 | task280 | task290 | task391 | task442 | task620 | task1342 | task1391 | task1598 | Average | Para. Saved |
|---|---|---|---|---|---|---|---|---|---|---|---|---|---|
|  | base | 0.26 ± 0.00 | 0.02 ± 0.00 | 0.19 ± 0.00 | 0.42 ± 0.00 | 0.11 ± 0.00 | 0.47 ± 0.00 | 0.11 ± 0.00 | 0.23 ± 0.00 | 0.19 ± 0.00 | 0.77 ± 0.00 | 0.28 ± 0.21 | 1.00/1.00 |
|  | lora | 1.00 ± 0.00 | 1.00 ± 0.00 | 1.00 ± 0.00 | 1.00 ± 0.00 | 1.00 ± 0.00 | 1.00 ± 0.00 | 1.00 ± 0.00 | 1.00 ± 0.00 | 1.00 ± 0.00 | 1.00 ± 0.00 | 1.00 ± 0.00 | 0.00/0.00 |
| TIES | 10 | 0.81 ± 0.00 | 0.57 ± 0.02 | 0.45 ± 0.04 | 0.10 ± 0.01 | 0.83 ± 0.01 | 0.47 ± 0.00 | 0.69 ± 0.01 | 0.57 ± 0.00 | 0.82 ± 0.01 | 0.85 ± 0.00 | 0.62 ± 0.23 | 1.00 / 1.00 |
|  | 50 | 0.59 ± 0.00 | 0.41 ± 0.00 | 0.18 ± 0.05 | 0.03 ± 0.01 | 0.91 ± 0.01 | 0.31 ± 0.00 | 0.65 ± 0.00 | 0.62 ± 0.00 | 0.32 ± 0.04 | 0.84 ± 0.00 | 0.48 ± 0.28 | 1.00 / 1.00 |
|  | 100 | 0.55 ± 0.00 | 0.40 ± 0.00 | 0.20 ± 0.05 | 0.01 ± 0.02 | 0.88 ± 0.00 | 0.33 ± 0.00 | 0.64 ± 0.00 | 0.57 ± 0.02 | 0.01 ± 0.00 | 0.82 ± 0.00 | 0.44 ± 0.30 | 1.00 / 1.00 |
|  | 500 | 0.37 ± 0.00 | 0.26 ± 0.00 | 0.01 ± 0.00 | 0.00 ± 0.00 | 0.83 ± 0.00 | 0.29 ± 0.00 | 0.57 ± 0.00 | 0.37 ± 0.00 | 0.01 ± 0.00 | 0.43 ± 0.00 | 0.31 ± 0.26 | 1.00 / 1.00 |
| SVD | SVD 2 | 0.98 ± 0.03 | 1.07 ± 0.02 | 1.00 ± 0.00 | 1.00 ± 0.00 | 1.00 ± 0.00 | 0.98 ± 0.01 | 1.00 ± 0.01 | 1.00 ± 0.10 | 1.00 ± 0.01 | 1.00 ± 0.01 | 1.00 ± 0.04 | 0.88 / 0.88 |
|  | SVD 4 | 0.99 ± 0.04 | 1.04 ± 0.01 | 1.00 ± 0.00 | 1.00 ± 0.00 | 1.00 ± 0.01 | 1.00 ± 0.01 | 1.00 ± 0.00 | 0.99 ± 0.02 | 0.99 ± 0.01 | 0.99 ± 0.01 | 1.00 ± 0.03 | 0.75 / 0.75 |
|  | SVD 8 | 1.00 ± 0.00 | 1.02 ± 0.01 | 1.00 ± 0.00 | 1.00 ± 0.00 | 1.00 ± 0.00 | 1.00 ± 0.00 | 1.00 ± 0.00 | 1.01 ± 0.00 | 1.00 ± 0.01 | 1.01 ± 0.01 | 1.00 ± 0.01 | 0.50 / 0.50 |
|  | SVD 16 | 1.00 ± 0.00 | 1.00 ± 0.00 | 1.00 ± 0.00 | 1.00 ± 0.00 | 1.00 ± 0.00 | 1.00 ± 0.00 | 1.00 ± 0.00 | 1.00 ± 0.00 | 1.00 ± 0.00 | 1.00 ± 0.00 | 1.00 ± 0.00 | 0.00 / 0.00 |
| 10 diagonal (D) | 16 D | 1.02 ± 0.01 | 1.01 ± 0.01 | 1.00 ± 0.00 | 1.00 ± 0.01 | 0.99 ± 0.00 | 0.96 ± 0.00 | 1.02 ± 0.02 | 1.13 ± 0.03 | 0.99 ± 0.02 | 0.98 ± 0.01 | 1.01 ± 0.05 | 1.00 / 0.90 |
|  | 32 D | 1.01 ± 0.01 | 1.05 ± 0.01 | 1.00 ± 0.00 | 0.99 ± 0.00 | 1.01 ± 0.01 | 0.99 ± 0.00 | 0.97 ± 0.01 | 1.05 ± 0.03 | 1.00 ± 0.01 | 1.00 ± 0.01 | 1.00 ± 0.03 | 1.00 / 0.80 |
|  | 64 D | 1.00 ± 0.00 | 1.03 ± 0.01 | 1.00 ± 0.00 | 1.00 ± 0.00 | 1.00 ± 0.00 | 1.00 ± 0.00 | 1.00 ± 0.00 | 0.99 ± 0.01 | 1.01 ± 0.00 | 1.00 ± 0.01 | 1.00 ± 0.01 | 1.00 / 0.60 |
|  | 128 D | 1.00 ± 0.00 | 1.01 ± 0.00 | 1.00 ± 0.00 | 1.00 ± 0.00 | 1.00 ± 0.00 | 1.00 ± 0.00 | 1.00 ± 0.00 | 1.01 ± 0.01 | 0.99 ± 0.01 | 1.00 ± 0.00 | 1.00 ± 0.01 | 1.00 / 0.20 |
|  | 256 D | 1.00 ± 0.00 | 1.00 ± 0.00 | 1.00 ± 0.00 | 1.00 ± 0.00 | 1.00 ± 0.00 | 1.00 ± 0.00 | 1.00 ± 0.00 | 1.00 ± 0.00 | 1.00 ± 0.00 | 1.00 ± 0.00 | 1.00 ± 0.00 | 1.00 / -0.60 |
| 10 full (F) | 16 F | 1.02 ± 0.00 | 1.06 ± 0.01 | 1.00 ± 0.00 | 1.00 ± 0.00 | 0.99 ± 0.01 | 0.98 ± 0.00 | 1.01 ± 0.02 | 1.07 ± 0.00 | 1.01 ± 0.01 | 1.00 ± 0.00 | 1.01 ± 0.03 | 1.00/0.90 |
|  | 32 F | 1.02 ± 0.01 | 1.04 ± 0.01 | 1.00 ± 0.00 | 1.00 ± 0.00 | 1.00 ± 0.00 | 0.99 ± 0.00 | 0.96 ± 0.01 | 1.00 ± 0.02 | 1.00 ± 0.01 | 1.01 ± 0.00 | 1.00 ± 0.02 | 0.99/0.79 |
|  | 64 F | 1.00 ± 0.00 | 1.03 ± 0.00 | 1.00 ± 0.00 | 1.00 ± 0.00 | 1.00 ± 0.00 | 1.00 ± 0.00 | 1.01 ± 0.00 | 0.98 ± 0.01 | 1.01 ± 0.00 | 1.01 ± 0.00 | 1.00 ± 0.01 | 0.97/0.57 |
|  | 128 F | 1.00 ± 0.00 | 1.01 ± 0.01 | 1.00 ± 0.00 | 1.00 ± 0.00 | 1.00 ± 0.00 | 1.00 ± 0.00 | 1.00 ± 0.00 | 0.99 ± 0.00 | 1.00 ± 0.00 | 1.00 ± 0.00 | 1.00 ± 0.00 | 0.88/0.07 |
|  | 256 F | 1.00 ± 0.00 | 1.00 ± 0.00 | 1.00 ± 0.00 | 1.00 ± 0.00 | 1.00 ± 0.00 | 1.00 ± 0.00 | 1.00 ± 0.00 | 1.00 ± 0.00 | 1.00 ± 0.00 | 1.00 ± 0.00 | 1.00 ± 0.00 | 0.50/-1.10 |
| 50 diagonal (D) | 16 D | 0.98 ± 0.04 | 0.98 ± 0.01 | 1.00 ± 0.00 | 0.92 ± 0.06 | 0.84 ± 0.07 | 0.92 ± 0.02 | 0.68 ± 0.05 | 0.87 ± 0.10 | 0.88 ± 0.07 | 0.83 ± 0.02 | 0.89 ± 0.10 | 1.00 / 0.98 |
|  | 32 D | 1.00 ± 0.02 | 1.02 ± 0.02 | 1.00 ± 0.00 | 0.99 ± 0.00 | 0.96 ± 0.01 | 0.95 ± 0.02 | 0.84 ± 0.02 | 1.00 ± 0.13 | 0.88 ± 0.01 | 0.88 ± 0.01 | 0.96 ± 0.07 | 1.00 / 0.96 |
|  | 64 D | 1.02 ± 0.00 | 1.05 ± 0.02 | 1.00 ± 0.00 | 1.00 ± 0.00 | 0.99 ± 0.01 | 0.97 ± 0.00 | 0.99 ± 0.01 | 1.09 ± 0.03 | 1.01 ± 0.01 | 0.90 ± 0.01 | 1.00 ± 0.05 | 1.00 / 0.92 |
|  | 128 D | 1.01 ± 0.00 | 1.08 ± 0.01 | 1.00 ± 0.00 | 1.00 ± 0.00 | 0.99 ± 0.01 | 0.98 ± 0.00 | 0.98 ± 0.01 | 1.11 ± 0.03 | 1.00 ± 0.00 | 1.00 ± 0.01 | 1.01 ± 0.04 | 1.00 / 0.84 |
|  | 256 D | 1.01 ± 0.01 | 1.03 ± 0.01 | 1.00 ± 0.00 | 1.00 ± 0.00 | 1.00 ± 0.01 | 1.00 ± 0.00 | 0.97 ± 0.03 | 1.01 ± 0.03 | 1.00 ± 0.01 | 1.01 ± 0.00 | 1.00 ± 0.02 | 1.00 / 0.68 |
| 50 full (F) | 16 F | 0.99 ± 0.04 | 1.00 ± 0.01 | 1.00 ± 0.01 | 0.96 ± 0.01 | 0.95 ± 0.02 | 0.94 ± 0.01 | 0.64 ± 0.10 | 1.01 ± 0.15 | 0.97 ± 0.02 | 0.87 ± 0.00 | 0.93 ± 0.12 | 1.00/0.98 |
|  | 32 F | 1.02 ± 0.00 | 1.00 ± 0.02 | 1.00 ± 0.00 | 1.00 ± 0.00 | 0.98 ± 0.01 | 0.96 ± 0.00 | 0.95 ± 0.01 | 1.09 ± 0.02 | 1.01 ± 0.02 | 0.89 ± 0.01 | 0.99 ± 0.05 | 0.99/0.95 |
|  | 64 F | 1.02 ± 0.01 | 1.06 ± 0.02 | 1.00 ± 0.00 | 1.00 ± 0.00 | 0.99 ± 0.01 | 0.98 ± 0.01 | 1.03 ± 0.01 | 1.11 ± 0.04 | 1.00 ± 0.01 | 0.98 ± 0.02 | 1.02 ± 0.04 | 0.97/0.89 |
|  | 128 F | 1.02 ± 0.00 | 1.06 ± 0.01 | 1.00 ± 0.00 | 1.00 ± 0.00 | 1.00 ± 0.01 | 0.98 ± 0.00 | 0.98 ± 0.01 | 1.03 ± 0.04 | 1.00 ± 0.01 | 1.00 ± 0.00 | 1.00 ± 0.03 | 0.88/0.72 |
|  | 256 F | 1.00 ± 0.00 | 1.02 ± 0.00 | 1.00 ± 0.00 | 1.00 ± 0.00 | 1.00 ± 0.00 | 0.99 ± 0.00 | 0.99 ± 0.00 | 1.01 ± 0.01 | 1.00 ± 0.00 | 1.01 ± 0.00 | 1.00 ± 0.01 | 0.50/0.18 |
| 100 diagonal (D) | 16 D | 0.80 ± 0.07 | 0.89 ± 0.06 | 0.93 ± 0.03 | 0.96 ± 0.01 | 0.50 ± 0.09 | 0.78 ± 0.01 | 0.28 ± 0.07 | 0.52 ± 0.10 | 0.78 ± 0.03 | 0.81 ± 0.02 | 0.72 ± 0.22 | 1.00 / 0.99 |
|  | 32 D | 0.95 ± 0.06 | 0.98 ± 0.01 | 1.00 ± 0.00 | 0.91 ± 0.06 | 0.80 ± 0.14 | 0.89 ± 0.06 | 0.60 ± 0.10 | 0.77 ± 0.26 | 0.91 ± 0.02 | 0.83 ± 0.02 | 0.86 ± 0.14 | 1.00 / 0.98 |
|  | 64 D | 1.01 ± 0.03 | 1.01 ± 0.01 | 1.00 ± 0.00 | 0.98 ± 0.02 | 0.96 ± 0.01 | 0.94 ± 0.01 | 0.88 ± 0.05 | 1.11 ± 0.08 | 0.96 ± 0.02 | 0.87 ± 0.03 | 0.97 ± 0.07 | 1.00 / 0.96 |
|  | 128 D | 1.01 ± 0.00 | 1.02 ± 0.01 | 1.00 ± 0.00 | 1.00 ± 0.00 | 0.99 ± 0.01 | 0.97 ± 0.00 | 1.00 ± 0.03 | 1.11 ± 0.02 | 1.00 ± 0.00 | 0.89 ± 0.02 | 1.00 ± 0.05 | 1.00 / 0.92 |
|  | 256 D | 1.00 ± 0.00 | 1.06 ± 0.00 | 1.00 ± 0.00 | 1.00 ± 0.00 | 0.99 ± 0.00 | 0.98 ± 0.00 | 1.00 ± 0.01 | 1.11 ± 0.03 | 1.00 ± 0.01 | 0.98 ± 0.01 | 1.01 ± 0.04 | 1.00 / 0.84 |
| 100 full (F) | 16 F | 0.95 ± 0.01 | 0.97 ± 0.03 | 0.97 ± 0.03 | 0.97 ± 0.03 | 0.93 ± 0.01 | 0.92 ± 0.01 | 0.64 ± 0.03 | 0.89 ± 0.16 | 0.87 ± 0.02 | 0.83 ± 0.01 | 0.89 ± 0.11 | 1.00/0.99 |
|  | 32 F | 1.00 ± 0.02 | 0.99 ± 0.01 | 1.00 ± 0.00 | 1.00 ± 0.00 | 0.97 ± 0.01 | 0.95 ± 0.00 | 0.86 ± 0.03 | 1.12 ± 0.03 | 0.96 ± 0.01 | 0.87 ± 0.00 | 0.97 ± 0.07 | 0.99/0.97 |
|  | 64 F | 1.02 ± 0.00 | 1.00 ± 0.02 | 1.00 ± 0.00 | 1.00 ± 0.00 | 0.98 ± 0.00 | 0.96 ± 0.00 | 0.99 ± 0.01 | 1.09 ± 0.01 | 0.99 ± 0.02 | 0.89 ± 0.00 | 0.99 ± 0.05 | 0.97/0.93 |
|  | 128 F | 1.01 ± 0.01 | 1.05 ± 0.01 | 1.00 ± 0.00 | 0.99 ± 0.00 | 1.00 ± 0.00 | 0.98 ± 0.00 | 1.03 ± 0.01 | 1.10 ± 0.01 | 1.01 ± 0.00 | 0.99 ± 0.01 | 1.02 ± 0.04 | 0.88/0.80 |
|  | 256 F | 1.01 ± 0.01 | 1.03 ± 0.01 | 1.00 ± 0.00 | 1.00 ± 0.00 | 1.01 ± 0.00 | 0.99 ± 0.00 | 0.98 ± 0.00 | 1.00 ± 0.03 | 1.01 ± 0.00 | 1.01 ± 0.00 | 1.00 ± 0.01 | 0.50/0.34 |
| 100 w/clusters (C) | 16 C 5 | 1.12 | 1.02 | 1.00 | 1.00 | 0.98 | 0.96 | 1.00 | 1.20 | 1.04 | 0.90 | 1.02 | 1.00/0.95 |
|  | 16 C 7 | 1.12 | 1.02 | 1.00 | 1.00 | 1.00 | 0.97 | 1.01 | 1.30 | 1.03 | 0.92 | 1.04 | 1.00/0.93 |
| 500 diagonal (D) | 16 D | 0.57 ± 0.07 | 0.55 ± 0.03 | 0.83 ± 0.04 | 0.78 ± 0.16 | 0.85 ± 0.04 | 0.68 ± 0.07 | 0.24 ± 0.01 | 0.43 ± 0.01 | 0.76 ± 0.06 | 0.79 ± 0.01 | 0.65 ± 0.20 | 1.00 / 1.00 |
|  | 32 D | 0.61 ± 0.12 | 0.55 ± 0.08 | 0.83 ± 0.02 | 0.84 ± 0.12 | 0.91 ± 0.02 | 0.71 ± 0.05 | 0.29 ± 0.05 | 0.47 ± 0.08 | 0.79 ± 0.04 | 0.79 ± 0.01 | 0.68 ± 0.20 | 1.00 / 1.00 |
|  | 64 D | 0.73 ± 0.02 | 0.63 ± 0.11 | 0.89 ± 0.04 | 0.97 ± 0.00 | 0.94 ± 0.00 | 0.83 ± 0.05 | 0.45 ± 0.09 | 0.50 ± 0.07 | 0.82 ± 0.02 | 0.80 ± 0.02 | 0.76 ± 0.18 | 1.00 / 0.99 |
|  | 128 D | 0.84 ± 0.08 | 0.92 ± 0.02 | 0.97 ± 0.03 | 0.98 ± 0.01 | 0.94 ± 0.00 | 0.88 ± 0.02 | 0.60 ± 0.15 | 0.53 ± 0.01 | 0.85 ± 0.05 | 0.80 ± 0.02 | 0.83 ± 0.15 | 1.00 / 0.98 |
|  | 256 D | 0.99 ± 0.03 | 0.99 ± 0.00 | 1.00 ± 0.00 | 1.00 ± 0.00 | 0.96 ± 0.00 | 0.92 ± 0.03 | 0.66 ± 0.06 | 0.84 ± 0.14 | 0.92 ± 0.02 | 0.84 ± 0.01 | 0.91 ± 0.11 | 1.00 / 0.97 |
| 500 full (F) | 16 F | 0.57 ± 0.01 | 0.43 ± 0.07 | 0.78 ± 0.01 | 0.97 ± 0.00 | 0.96 ± 0.00 | 0.83 ± 0.01 | 0.64 ± 0.00 | 0.53 ± 0.03 | 0.83 ± 0.01 | 0.83 ± 0.00 | 0.75 ± 0.17 | 1.00/1.00 |
|  | 32 F | 0.79 ± 0.05 | 0.54 ± 0.04 | 0.93 ± 0.02 | 0.98 ± 0.00 | 0.97 ± 0.00 | 0.90 ± 0.01 | 0.69 ± 0.01 | 0.50 ± 0.00 | 0.86 ± 0.02 | 0.83 ± 0.01 | 0.81 ± 0.16 | 0.99/0.99 |
|  | 64 F | 1.02 ± 0.00 | 0.96 ± 0.01 | 0.94 ± 0.01 | 1.00 ± 0.00 | 0.96 ± 0.00 | 0.97 ± 0.01 | 0.73 ± 0.01 | 0.54 ± 0.01 | 0.91 ± 0.01 | 0.86 ± 0.00 | 0.89 ± 0.14 | 0.97/0.96 |
|  | 128 F | 1.03 ± 0.01 | 0.97 ± 0.02 | 0.99 ± 0.00 | 1.00 ± 0.00 | 0.98 ± 0.00 | 0.96 ± 0.00 | 0.87 ± 0.01 | 1.07 ± 0.02 | 0.98 ± 0.00 | 0.87 ± 0.00 | 0.97 ± 0.06 | 0.88/0.86 |
|  | 256 F | 1.03 ± 0.00 | 1.03 ± 0.01 | 1.00 ± 0.00 | 1.00 ± 0.00 | 0.99 ± 0.01 | 0.97 ± 0.01 | 0.99 ± 0.02 | 1.03 ± 0.01 | 1.00 ± 0.01 | 0.87 ± 0.00 | 0.99 ± 0.05 | 0.50/0.47 |
| 500 w/clusters (C) | 16 C 7 | 1.09 | 1.00 | 0.99 | 1.00 | 0.98 | 0.95 | 0.72 | 0.87 | 0.98 | 0.90 | 0.95 | 1.00/0.98 |
|  | 16 C 10 | 1.10 | 1.01 | 1.00 | 0.99 | 0.97 | 0.93 | 0.70 | 1.30 | 1.02 | 0.88 | 0.99 | 1.00/0.98 |
|  | 16 C 25 | 1.10 | 1.00 | 1.00 | 0.99 | 0.99 | 0.96 | 0.98 | 1.31 | 1.03 | 0.91 | 1.03 | 1.00/0.95 |
|  | 64 C 5 | 1.09 | 0.98 | 1.00 | 1.00 | 0.99 | 0.96 | 0.99 | 1.18 | 1.04 | 0.87 | 1.01 | 0.97/0.93 |
|  | 64 C 7 | 1.12 | 1.02 | 1.00 | 1.00 | 1.00 | 0.96 | 0.99 | 1.22 | 1.04 | 0.93 | 1.03 | 0.97/0.91 |

Table 4: **Relative** In-Distribution *ROUGE-L* scores for various tasks and methods

## G.7 AGREEMENT AND COMPRESSION RATE

Table 10 provides full results for *agreement*, which shows the same trends as the results for relative performance of Rouge-L (Table 4). Note that *agreement* measures the exact match in task generations between the uncompressed LoRA model and the compressed LoRA model, rather than comparing to the task's ground truth data. The comparison is very strict and requires an exact match between the generations of the two models (LoRA and the compressed LoRA), comparing each sample one at a time.

## G.8 RECONSTRUCTION ERROR AND COMPRESSION RATE

Table 11 provides the full results of the experiments behind Figure 3 for every evaluation task.

## G.9 RECONSTRUCTION ERROR: TRAINED VS. RANDOM

Table 12 provides the reconstruction error on random (untrained) LoRA matrices. Comparing with Table 11, we find that reconstruction error is consistently higher on random (untrained LoRA) matrices than on trained LoRA matrices. This demonstrates that after training, LoRAs have a shared structure that JD exploits.

| Model Type | Method Type | Tasks | | | | | | | | | | Average | Para. Saved |
|---|---|---|---|---|---|---|---|---|---|---|---|---|---|
| | | task039 | task190 | task280 | task290 | task391 | task442 | task620 | task1342 | task1391 | task1598 | | |
| | base | 24.44 ±.00 | 1.60 ±.00 | 19.13 ±.00 | 39.22 ±.00 | 10.27 ±.00 | 35.46 ±.00 | 7.85 ±.00 | 6.22 ±.00 | 17.82 ±.00 | 38.87 ±.00 | 20.24 ±13.27 | 1.00/1.00 |
| | lora | 95.00 ±.00 | 86.00 ±.00 | 99.00 ±.00 | 93.67 ±.00 | 94.33 ±.00 | 74.88 ±.00 | 74.40 ±.00 | 26.68 ±.00 | 95.00 ±.00 | 50.32 ±.00 | 78.87 ±22.56 | 0.00/0.00 |
| TIES | 10 | 76.50 ±1.00 | 49.00 ±1.73 | 44.33 ±4.04 | 9.80 ±0.58 | 78.56 ±0.96 | 35.24 ±0.00 | 51.37 ±0.67 | 15.26 ±0.12 | 77.67 ±1.15 | 42.72 ±0.01 | 48.05 ±23.61 | 1.00 / 1.00 |
| | 50 | 55.80 ±0.00 | 35.00 ±0.00 | 18.00 ±5.20 | 2.42 ±0.50 | 85.78 ±0.96 | 23.03 ±0.00 | 48.03 ±0.00 | 16.50 ±0.00 | 30.00 ±3.46 | 42.47 ±0.02 | 35.70 ±23.01 | 1.00 / 1.00 |
| | 100 | 52.43 ±0.00 | 34.00 ±0.00 | 19.67 ±4.62 | 1.09 ±1.66 | 83.33 ±0.00 | 24.89 ±0.00 | 47.52 ±0.00 | 15.18 ±0.42 | 1.00 ±0.00 | 41.19 ±0.03 | 32.03 ±24.50 | 1.00 / 1.00 |
| | 500 | 35.18 ±0.00 | 22.00 ±0.00 | 1.00 ±0.00 | 0.00 ±0.00 | 78.00 ±0.00 | 21.46 ±0.00 | 42.22 ±0.04 | 9.93 ±0.13 | 1.00 ±0.00 | 21.50 ±0.03 | 23.27 ±23.64 | 1.00 / 1.00 |
| SVD | SVD 2 | 93.15 ±2.77 | 92.24 ±1.85 | 99.09 ±0.18 | 93.44 ±0.14 | 93.89 ±0.35 | 73.74 ±0.51 | 74.55 ±0.98 | 26.80 ±2.79 | 95.06 ±1.35 | 50.21 ±0.44 | 79.11 ±22.72 | 0.88 / 0.88 |
| | SVD 4 | 94.01 ±3.60 | 89.21 ±0.71 | 99.05 ±0.09 | 93.65 ±0.03 | 94.66 ±0.63 | 74.89 ±0.33 | 73.61 ±1.15 | 26.34 ±2.13 | 93.98 ±0.77 | 50.47 ±0.54 | 78.90 ±22.68 | 0.75 / 0.75 |
| | SVD 8 | 95.00 ±0.00 | 87.40 ±0.59 | 99.05 ±0.03 | 93.65 ±0.03 | 94.36 ±0.38 | 74.58 ±0.12 | 75.07 ±0.00 | 26.71 ±0.27 | 95.51 ±1.09 | 50.89 ±0.07 | 81.01 ±21.74 | 0.50 / 0.50 |
| | SVD 16 | 95.00 ±0.00 | 86.00 ±0.00 | 99.00 ±0.00 | 93.67 ±0.00 | 94.33 ±0.00 | 74.90 ±0.03 | 74.23 ±0.18 | 26.68 ±0.00 | 95.00 ±0.00 | 50.30 ±0.02 | 78.36 ±22.97 | 0.00 / 0.00 |
| 10 diagonal (D) | 16 D | 96.67 ±0.58 | 87.00 ±1.00 | 99.00 ±0.00 | 94.00 ±0.67 | 93.11 ±0.38 | 72.08 ±0.06 | 76.26 ±1.19 | 30.11 ±0.79 | 94.00 ±1.73 | 49.30 ±0.46 | 79.15 ±22.18 | 1.00 / 0.90 |
| | 32 D | 95.67 ±0.58 | 90.00 ±1.00 | 99.00 ±0.00 | 93.00 ±0.33 | 94.89 ±0.51 | 73.86 ±0.31 | 71.92 ±0.84 | 27.89 ±0.70 | 94.67 ±0.58 | 50.36 ±0.26 | 79.13 ±22.75 | 1.00 / 0.80 |
| | 64 D | 95.00 ±0.00 | 88.33 ±0.58 | 99.00 ±0.00 | 93.67 ±0.00 | 94.78 ±0.38 | 74.61 ±0.13 | 74.97 ±0.58 | 26.35 ±0.25 | 96.00 ±0.00 | 50.99 ±0.06 | 79.37 ±22.94 | 1.00 / 0.60 |
| | 128 D | 95.00 ±0.00 | 86.67 ±0.58 | 99.00 ±0.00 | 93.67 ±0.00 | 94.33 ±0.00 | 74.92 ±0.13 | 74.96 ±0.51 | 26.45 ±0.23 | 95.00 ±0.00 | 50.21 ±0.12 | 79.02 ±22.84 | 1.00 / 0.20 |
| | 256 D | 95.00 ±0.00 | 86.00 ±0.00 | 99.00 ±0.00 | 93.67 ±0.00 | 94.33 ±0.00 | 74.88 ±0.00 | 74.40 ±0.00 | 26.68 ±0.00 | 95.00 ±0.00 | 50.27 ±0.02 | 78.92 ±22.77 | 1.00 / -0.60 |
| 10 full (F) | 16 F | 97.00 ±0.00 | 91.00 ±1.00 | 99.00 ±0.00 | 93.56 ±0.19 | 93.56 ±0.69 | 73.60 ±0.36 | 74.94 ±1.25 | 28.66 ±0.03 | 96.00 ±1.00 | 50.15 ±0.20 | 79.75 ±22.72 | 1.00/0.90 |
| | 32 F | 96.67 ±0.58 | 89.33 ±0.58 | 99.00 ±0.00 | 93.22 ±0.19 | 94.44 ±0.19 | 74.11 ±0.19 | 71.74 ±0.59 | 26.74 ±0.50 | 94.67 ±0.58 | 50.63 ±0.24 | 79.06 ±23.01 | 0.99/0.79 |
| | 64 F | 95.00 ±0.00 | 88.67 ±0.58 | 99.00 ±0.00 | 93.67 ±0.00 | 94.56 ±0.38 | 74.56 ±0.13 | 75.47 ±0.58 | 26.26 ±0.34 | 96.00 ±0.00 | 50.89 ±0.17 | 79.41 ±22.97 | 0.97/0.57 |
| | 128 F | 95.00 ±0.00 | 86.67 ±0.58 | 99.00 ±0.00 | 93.67 ±0.00 | 94.33 ±0.00 | 75.04 ±0.03 | 74.40 ±0.00 | 26.53 ±0.13 | 95.00 ±0.00 | 50.36 ±0.03 | 79.00 ±22.81 | 0.88/0.07 |
| | 256 F | 95.00 ±0.00 | 86.00 ±0.00 | 99.00 ±0.00 | 93.67 ±0.00 | 94.33 ±0.00 | 74.90 ±0.03 | 74.29 ±0.19 | 26.68 ±0.00 | 95.00 ±0.00 | 50.30 ±0.03 | 78.92 ±22.77 | 0.50/-1.10 |
| 50 diagonal (D) | 16 D | 92.76 ±3.53 | 84.67 ±1.15 | 99.00 ±0.00 | 86.17 ±5.81 | 79.68 ±6.21 | 69.07 ±1.54 | 50.65 ±3.97 | 23.27 ±2.60 | 83.90 ±6.43 | 41.86 ±0.96 | 71.10 ±23.99 | 1.00 / 0.98 |
| | 32 D | 95.33 ±2.08 | 87.33 ±2.08 | 99.00 ±0.00 | 92.60 ±0.29 | 90.32 ±1.04 | 71.16 ±1.47 | 62.51 ±1.64 | 26.60 ±3.54 | 93.33 ±1.15 | 44.35 ±0.41 | 76.25 ±23.81 | 1.00 / 0.96 |
| | 64 D | 97.00 ±0.00 | 90.33 ±1.53 | 99.00 ±0.00 | 93.78 ±0.19 | 93.00 ±0.58 | 72.37 ±0.35 | 73.39 ±0.93 | 29.06 ±0.80 | 95.67 ±0.58 | 45.43 ±0.34 | 78.90 ±23.29 | 1.00 / 0.92 |
| | 128 D | 96.33 ±0.58 | 92.67 ±0.58 | 99.00 ±0.00 | 93.56 ±0.19 | 93.00 ±0.58 | 73.32 ±0.24 | 73.03 ±1.09 | 29.51 ±0.93 | 95.00 ±0.00 | 50.16 ±0.74 | 79.56 ±22.51 | 1.00 / 0.84 |
| | 256 D | 95.67 ±0.58 | 88.33 ±0.58 | 99.00 ±0.00 | 93.56 ±0.19 | 94.67 ±0.67 | 74.82 ±0.24 | 72.36 ±2.07 | 26.90 ±0.75 | 95.33 ±0.58 | 50.73 ±0.46 | 79.14 ±22.90 | 1.00 / 0.68 |
| 50 full (F) | 16 F | 94.06 ±3.54 | 85.67 ±1.15 | 98.67 ±0.58 | 90.35 ±1.37 | 89.90 ±1.91 | 70.32 ±0.66 | 47.62 ±7.28 | 26.88 ±3.96 | 92.33 ±1.53 | 43.68 ±0.24 | 73.95 ±24.73 | 1.00/0.98 |
| | 32 F | 97.00 ±0.00 | 85.67 ±1.53 | 99.00 ±0.00 | 93.67 ±0.00 | 92.22 ±0.69 | 71.88 ±0.30 | 71.01 ±1.02 | 29.07 ±0.65 | 95.67 ±1.53 | 44.97 ±0.41 | 78.02 ±23.18 | 0.99/0.95 |
| | 64 F | 96.67 ±0.58 | 91.00 ±2.00 | 99.00 ±0.00 | 93.56 ±0.19 | 93.22 ±0.51 | 73.16 ±0.41 | 76.28 ±0.51 | 29.67 ±0.12 | 95.33 ±0.58 | 49.31 ±1.00 | 79.72 ±22.50 | 0.97/0.89 |
| | 128 F | 97.00 ±0.00 | 91.00 ±1.00 | 99.00 ±0.00 | 93.33 ±0.00 | 94.11 ±0.51 | 73.51 ±0.23 | 73.17 ±0.58 | 27.53 ±1.12 | 95.00 ±1.00 | 50.56 ±0.06 | 79.42 ±22.93 | 0.88/0.72 |
| | 256 F | 95.00 ±0.00 | 88.00 ±0.00 | 99.00 ±0.00 | 93.67 ±0.00 | 94.44 ±0.19 | 74.25 ±0.21 | 74.97 ±0.58 | 26.79 ±0.09 | 96.00 ±0.00 | 50.36 ±0.19 | 79.30 ±22.82 | 0.50/0.18 |
| 100 diagonal (D) | 16 D | 76.43 ±7.07 | 76.67 ±4.93 | 91.61 ±2.75 | 89.99 ±1.07 | 47.55 ±8.56 | 58.08 ±0.72 | 20.77 ±5.50 | 13.90 ±2.79 | 73.93 ±3.13 | 40.74 ±0.85 | 58.97 ±26.83 | 1.00 / 0.99 |
| | 32 D | 90.10 ±5.85 | 84.00 ±1.00 | 99.00 ±0.00 | 85.52 ±5.34 | 75.69 ±12.75 | 66.62 ±4.18 | 44.66 ±7.26 | 20.49 ±7.07 | 86.67 ±1.86 | 42.01 ±0.94 | 69.48 ±25.14 | 1.00 / 0.98 |
| | 64 D | 95.56 ±2.49 | 86.67 ±0.58 | 99.00 ±0.00 | 92.24 ±1.68 | 90.89 ±1.17 | 70.35 ±0.45 | 65.62 ±4.03 | 29.58 ±2.02 | 91.67 ±2.31 | 43.64 ±1.36 | 76.52 ±23.02 | 1.00 / 0.96 |
| | 128 D | 96.00 ±0.00 | 87.33 ±1.15 | 99.00 ±0.00 | 93.89 ±0.19 | 93.00 ±0.58 | 72.70 ±0.30 | 74.34 ±2.07 | 29.66 ±0.54 | 93.67 ±0.58 | 44.82 ±0.89 | 78.44 ±22.87 | 1.00 / 0.92 |
| | 256 D | 95.00 ±0.00 | 91.00 ±0.00 | 99.00 ±0.00 | 93.56 ±0.19 | 93.11 ±0.19 | 73.05 ±0.20 | 74.52 ±0.95 | 29.67 ±0.67 | 95.33 ±0.58 | 49.42 ±0.65 | 79.37 ±22.38 | 1.00 / 0.84 |
| 100 full (F) | 16 F | 90.70 ±1.07 | 83.00 ±2.65 | 96.00 ±3.00 | 91.22 ±2.94 | 87.94 ±0.54 | 68.72 ±1.05 | 47.57 ±2.54 | 23.75 ±4.33 | 82.33 ±2.08 | 41.51 ±0.67 | 71.27 ±24.23 | 1.00/0.99 |
| | 32 F | 95.33 ±1.53 | 85.00 ±1.00 | 99.00 ±0.00 | 93.50 ±0.22 | 91.44 ±0.84 | 70.94 ±0.02 | 63.64 ±1.98 | 29.82 ±0.81 | 91.67 ±0.58 | 43.94 ±0.18 | 76.43 ±23.01 | 0.99/0.97 |
| | 64 F | 97.00 ±0.00 | 85.67 ±1.53 | 99.00 ±0.00 | 93.78 ±0.19 | 92.56 ±0.19 | 72.11 ±0.08 | 73.29 ±0.64 | 29.15 ±0.24 | 94.33 ±1.53 | 44.97 ±0.05 | 78.18 ±23.03 | 0.97/0.93 |
| | 128 F | 96.33 ±0.58 | 90.33 ±0.58 | 99.00 ±0.00 | 93.00 ±0.00 | 93.89 ±0.19 | 73.11 ±0.36 | 76.50 ±1.01 | 29.45 ±0.35 | 96.00 ±0.00 | 49.81 ±0.34 | 79.74 ±22.47 | 0.88/0.80 |
| | 256 F | 96.33 ±0.58 | 88.67 ±0.58 | 99.00 ±0.00 | 93.67 ±0.00 | 94.89 ±0.19 | 74.40 ±0.16 | 72.90 ±0.12 | 26.77 ±0.68 | 96.00 ±0.00 | 50.83 ±0.09 | 79.35 ±23.04 | 0.50/0.34 |
| 100 w/clusters (C) | 16 C 5 | 98.00 | 88.00 | 99.00 | 93.38 | 91.67 | 72.02 | 76.80 | 27.74 | 96.00 | 46.06 | 78.87 | 1.00/0.95 |
| | 16 C 7 | 98.00 | 88.00 | 99.00 | 93.67 | 94.00 | 72.97 | 76.83 | 29.91 | 95.00 | 47.33 | 79.47 | 1.00/0.93 |
| 500 diagonal (D) | 16 D | 54.44 ±6.87 | 47.00 ±2.83 | 82.21 ±3.59 | 73.38 ±14.97 | 80.08 ±3.71 | 51.02 ±5.31 | 17.49 ±1.10 | 11.58 ±0.21 | 72.67 ±6.03 | 39.65 ±0.28 | 53.16 ±24.97 | 1.00 / 1.00 |
| | 32 D | 58.08 ±11.52 | 47.00 ±7.07 | 82.06 ±1.69 | 78.62 ±11.23 | 85.57 ±1.48 | 52.98 ±3.81 | 21.73 ±3.95 | 12.53 ±2.26 | 75.33 ±4.04 | 39.78 ±0.42 | 55.66 ±25.48 | 1.00 / 1.00 |
| | 64 D | 69.21 ±2.03 | 54.50 ±9.19 | 88.33 ±4.04 | 91.11 ±0.38 | 88.78 ±0.38 | 62.36 ±3.52 | 33.36 ±6.69 | 13.34 ±1.86 | 77.67 ±2.31 | 40.42 ±0.98 | 62.16 ±26.05 | 1.00 / 0.99 |
| | 128 D | 79.77 ±0.37 | 79.50 ±2.12 | 95.89 ±2.83 | 91.89 ±1.39 | 88.67 ±0.00 | 65.92 ±1.79 | 44.98 ±10.98 | 14.14 ±0.19 | 81.00 ±5.00 | 40.34 ±0.80 | 67.82 ±26.35 | 1.00 / 0.98 |
| | 256 D | 93.83 ±2.52 | 85.00 ±0.00 | 99.00 ±0.00 | 93.78 ±0.19 | 90.56 ±0.38 | 68.95 ±1.92 | 49.39 ±4.36 | 22.33 ±3.78 | 87.33 ±2.31 | 42.15 ±0.73 | 72.83 ±25.93 | 1.00 / 0.97 |
| 500 full (F) | 16 F | 54.30 ±1.13 | 37.00 ±5.66 | 77.67 ±0.58 | 91.00 ±0.00 | 90.56 ±0.19 | 62.47 ±0.79 | 47.56 ±0.29 | 14.18 ±0.67 | 79.00 ±1.00 | 41.58 ±0.23 | 60.31 ±24.42 | 1.00/1.00 |
| | 32 F | 75.10 ±4.92 | 46.50 ±3.54 | 91.67 ±1.53 | 91.56 ±0.19 | 91.56 ±0.38 | 67.37 ±0.83 | 51.17 ±0.81 | 13.44 ±0.02 | 81.67 ±1.53 | 41.92 ±0.42 | 65.84 ±25.64 | 0.99/0.99 |
| | 64 F | 96.94 ±0.42 | 82.50 ±0.71 | 93.33 ±0.58 | 93.89 ±0.69 | 90.67 ±0.00 | 72.30 ±0.71 | 54.63 ±0.79 | 14.49 ±0.27 | 86.33 ±0.58 | 43.16 ±0.08 | 72.49 ±26.64 | 0.97/0.96 |
| | 128 F | 97.67 ±0.58 | 83.50 ±2.12 | 98.00 ±0.00 | 93.56 ±0.19 | 92.00 ±0.00 | 71.92 ±0.19 | 65.02 ±0.81 | 28.49 ±0.55 | 93.00 ±0.00 | 43.85 ±0.12 | 76.47 ±23.77 | 0.88/0.86 |
| | 256 F | 95.00 ±0.00 | 88.50 ±0.71 | 99.00 ±0.00 | 93.78 ±0.19 | 93.00 ±0.88 | 72.45 ±0.38 | 73.77 ±1.21 | 27.59 ±0.39 | 95.33 ±0.58 | 43.81 ±0.17 | 78.18 ±24.16 | 0.50/0.47 |
| 500 w/clusters (C) | 16 C 7 | 95.00 | 86.00 | 98.00 | 93.67 | 91.67 | 71.19 | 54.69 | 20.03 | 90.00 | 46.34 | 74.66 | 1.00/0.98 |
| | 16 C 10 | 96.00 | 87.00 | 99.00 | 93.00 | 91.33 | 69.93 | 53.48 | 30.09 | 94.00 | 44.89 | 75.87 | 1.00/0.98 |
| | 16 C 25 | 96.00 | 86.00 | 99.00 | 92.71 | 93.00 | 72.13 | 74.59 | 30.21 | 95.00 | 46.66 | 78.53 | 1.00/0.95 |
| | 64 C 5 | 95.00 | 84.00 | 99.00 | 93.67 | 92.67 | 72.32 | 75.60 | 27.17 | 96.00 | 44.43 | 77.99 | 0.97/0.93 |
| | 64 C 7 | 98.00 | 88.00 | 99.00 | 94.00 | 93.33 | 72.18 | 75.83 | 28.14 | 96.00 | 47.68 | 79.22 | 0.97/0.91 |

Table 5: **Absolute** In-Distribution ROUGE-L scores for various tasks and methods

## G.10 CONVERGENCE

Table 13 presents outcomes where the JD-Full algorithm is executed until convergence. Our convergence criterion is defined as follows:

$$\max\left(\|U_{t+1} - U_t U_t^\top U_{t+1}\|_{\mathrm{Fro}}/\|U_{t+1}\|_{\mathrm{Fro}}, \|V_{t+1} - V_t V_t^\top V_{t+1}\|_{\mathrm{Fro}}/\|V_{t+1}\|_{\mathrm{Fro}}\right) < \tau \qquad (19)$$

where the tolerance threshold $\tau$ is set to 0.001. Due to the slow per-iteration computation times of the primary JD-Full algorithm, which quickly reaches an approximate optimum but then has a long tail of convergence for final digits of precision, we devised an alternative eigenvalue iteration algorithm (Appendix A.2) optimized for GPU acceleration. Our analysis indicates that adherence to this convergence criterion does not significantly alter the results.

## G.11 OUT-OF-DISTRIBUTION PERFORMANCE (LORA-HUB)

For completeness, we incorporate results using the protocol of LoRA-hub (Huang et al., 2024). That is, 100 LoRA-adapters are sampled, independent of the evaluation task, representing a measure of out-of-distribution performance. This also means that each result on a task is averaged across all 100 LoRA-adapters (as there is no *a priori* LoRA-to-task mapping). These results were obtained without normalizing the LoRA-adapters before applying the JD algorithms, a step we later identified as beneficial. We present performance comparison in Table 15. Table 14 presents the average agreement between uncompressed and compressed LoRA across 10 evaluation tasks. Results per task for JD-diagonal and JD-full are shown in Table 16 and Table 17, respectively.

From these tables, we find that the JD algorithms successfully maintain performance in this out-of-distribution context.

| Model Type | Method Type | Tasks | | | | | | | | | | Average | Para. Saved |
|---|---|---|---|---|---|---|---|---|---|---|---|---|---|
| | | task039 | task190 | task280 | task290 | task391 | task442 | task620 | task1342 | task1391 | task1598 | | |
| | base | 0.26 ±0.00 | 0.02 ±0.00 | 0.19 ±0.00 | 0.42 ±0.00 | 0.11 ±0.00 | 0.51 ±0.00 | 0.11 ±0.00 | 0.26 ±0.00 | 0.19 ±0.00 | 0.80 ±0.00 | 0.29 ±0.22 | 1.00/1.00 |
| | lora | 1.00 ±0.00 | 1.00 ±0.00 | 1.00 ±0.00 | 1.00 ±0.00 | 1.00 ±0.00 | 1.00 ±0.00 | 1.00 ±0.00 | 1.00 ±0.00 | 1.00 ±0.00 | 1.00 ±0.00 | 1.00 ±0.00 | 0.00/0.00 |
| TIES | 10 | 0.81 ±0.00 | 0.57 ±0.02 | 0.45 ±0.04 | 0.10 ±0.01 | 0.83 ±0.01 | 0.52 ±0.00 | 0.71 ±0.01 | 0.58 ±0.00 | 0.82 ±0.01 | 0.80 ±0.00 | 0.62 ±0.22 | 1.00 / 1.00 |
| | 50 | 0.59 ±0.00 | 0.41 ±0.00 | 0.18 ±0.05 | 0.03 ±0.01 | 0.91 ±0.01 | 0.34 ±0.00 | 0.67 ±0.00 | 0.62 ±0.00 | 0.32 ±0.04 | 0.78 ±0.00 | 0.48 ±0.27 | 1.00 / 1.00 |
| | 100 | 0.55 ±0.00 | 0.40 ±0.00 | 0.20 ±0.05 | 0.01 ±0.02 | 0.88 ±0.00 | 0.36 ±0.00 | 0.65 ±0.00 | 0.57 ±0.02 | 0.01 ±0.00 | 0.78 ±0.00 | 0.44 ±0.29 | 1.00 / 1.00 |
| | 500 | 0.37 ±0.00 | 0.26 ±0.00 | 0.01 ±0.00 | 0.00 ±0.00 | 0.83 ±0.00 | 0.31 ±0.00 | 0.58 ±0.00 | 0.37 ±0.00 | 0.01 ±0.00 | 0.41 ±0.00 | 0.32 ±0.26 | 1.00 / 1.00 |
| SVD | SVD 2 | 0.98 ±0.03 | 1.07 ±0.02 | 1.00 ±0.00 | 1.00 ±0.00 | 1.00 ±0.00 | 0.99 ±0.00 | 1.01 ±0.01 | 1.00 ±0.10 | 1.00 ±0.01 | 0.99 ±0.01 | 1.00 ±0.04 | 0.88 / 0.88 |
| | SVD 4 | 0.99 ±0.04 | 1.04 ±0.01 | 1.00 ±0.00 | 1.00 ±0.00 | 1.00 ±0.01 | 1.00 ±0.00 | 0.99 ±0.01 | 0.99 ±0.08 | 0.99 ±0.01 | 1.01 ±0.00 | 1.00 ±0.03 | 0.75 / 0.75 |
| | SVD 8 | 1.00 ±0.00 | 1.02 ±0.01 | 1.00 ±0.00 | 1.00 ±0.00 | 1.00 ±0.00 | 1.00 ±0.00 | 1.01 ±0.00 | 1.00 ±0.01 | 1.01 ±0.01 | 1.01 ±0.00 | 1.00 ±0.01 | 0.50 / 0.50 |
| | SVD 16 | 1.00 ±0.00 | 1.00 ±0.00 | 1.00 ±0.00 | 1.00 ±0.00 | 1.00 ±0.00 | 1.00 ±0.00 | 1.00 ±0.00 | 1.00 ±0.00 | 1.00 ±0.00 | 1.00 ±0.00 | 1.00 ±0.00 | 0.00 / 0.00 |
| 10 diagonal (D) | 16 D | 1.02 ±0.01 | 1.01 ±0.01 | 1.00 ±0.00 | 1.00 ±0.01 | 0.99 ±0.00 | 0.97 ±0.00 | 1.03 ±0.02 | 1.12 ±0.03 | 0.99 ±0.02 | 0.99 ±0.00 | 1.01 ±0.04 | 1.00 / 0.90 |
| | 32 D | 1.01 ±0.01 | 1.05 ±0.01 | 1.00 ±0.00 | 0.99 ±0.00 | 1.01 ±0.01 | 0.99 ±0.00 | 0.97 ±0.01 | 1.04 ±0.03 | 1.00 ±0.01 | 1.01 ±0.01 | 1.01 ±0.02 | 1.00 / 0.80 |
| | 64 D | 1.00 ±0.00 | 1.03 ±0.01 | 1.00 ±0.00 | 1.00 ±0.00 | 1.00 ±0.00 | 1.00 ±0.00 | 1.01 ±0.01 | 0.99 ±0.01 | 1.01 ±0.00 | 1.00 ±0.01 | 1.00 ±0.01 | 1.00 / 0.60 |
| | 128 D | 1.00 ±0.00 | 1.01 ±0.00 | 1.00 ±0.00 | 1.00 ±0.00 | 1.00 ±0.00 | 1.00 ±0.00 | 1.01 ±0.01 | 0.99 ±0.01 | 1.00 ±0.00 | 1.00 ±0.00 | 1.00 ±0.01 | 1.00 / 0.20 |
| | 256 D | 1.00 ±0.00 | 1.00 ±0.00 | 1.00 ±0.00 | 1.00 ±0.00 | 1.00 ±0.00 | 1.00 ±0.00 | 1.00 ±0.00 | 1.00 ±0.00 | 1.00 ±0.00 | 1.00 ±0.00 | 1.00 ±0.00 | 1.00 / -0.60 |
| 10 full (F) | 16 F | 1.02 ±0.00 | 1.06 ±0.00 | 1.00 ±0.00 | 1.00 ±0.00 | 0.99 ±0.01 | 0.99 ±0.00 | 1.01 ±0.02 | 1.07 ±0.00 | 1.01 ±0.01 | 1.00 ±0.00 | 1.02 ±0.03 | 1.00/0.90 |
| | 32 F | 1.02 ±0.01 | 1.04 ±0.01 | 1.00 ±0.00 | 1.00 ±0.00 | 1.00 ±0.00 | 0.99 ±0.00 | 0.96 ±0.01 | 1.00 ±0.02 | 1.00 ±0.01 | 1.01 ±0.00 | 1.00 ±0.02 | 0.99/0.79 |
| | 64 F | 1.00 ±0.00 | 1.03 ±0.01 | 1.00 ±0.00 | 1.00 ±0.00 | 1.00 ±0.00 | 1.00 ±0.00 | 1.01 ±0.01 | 0.98 ±0.01 | 1.01 ±0.00 | 1.00 ±0.01 | 1.00 ±0.01 | 0.97/0.57 |
| | 128 F | 1.00 ±0.00 | 1.01 ±0.01 | 1.00 ±0.00 | 1.00 ±0.00 | 1.00 ±0.00 | 1.00 ±0.00 | 1.00 ±0.00 | 0.99 ±0.00 | 1.00 ±0.00 | 1.00 ±0.00 | 1.00 ±0.00 | 0.88/0.07 |
| | 256 F | 1.00 ±0.00 | 1.00 ±0.00 | 1.00 ±0.00 | 1.00 ±0.00 | 1.00 ±0.00 | 1.00 ±0.00 | 1.00 ±0.00 | 1.00 ±0.00 | 1.00 ±0.00 | 1.00 ±0.00 | 1.00 ±0.00 | 0.50/-1.10 |
| 50 diagonal (D) | 16 D | 0.98 ±0.04 | 0.98 ±0.01 | 1.00 ±0.00 | 0.92 ±0.06 | 0.85 ±0.06 | 0.94 ±0.02 | 0.69 ±0.05 | 0.88 ±0.10 | 0.88 ±0.07 | 0.86 ±0.01 | 0.90 ±0.10 | 1.00 / 0.98 |
| | 32 D | 1.00 ±0.02 | 1.02 ±0.02 | 1.00 ±0.00 | 0.99 ±0.00 | 0.96 ±0.01 | 0.96 ±0.02 | 0.85 ±0.02 | 1.00 ±0.12 | 0.98 ±0.01 | 0.90 ±0.04 | 0.97 ±0.06 | 1.00 / 0.96 |
| | 64 D | 1.02 ±0.00 | 1.05 ±0.02 | 1.00 ±0.00 | 1.00 ±0.00 | 0.99 ±0.01 | 0.97 ±0.01 | 0.99 ±0.01 | 1.09 ±0.03 | 0.94 ±0.00 | 0.94 ±0.00 | 1.01 ±0.04 | 1.00 / 0.92 |
| | 128 D | 1.01 ±0.01 | 1.08 ±0.01 | 1.00 ±0.00 | 1.00 ±0.00 | 0.99 ±0.01 | 0.98 ±0.00 | 0.98 ±0.02 | 1.10 ±0.03 | 1.00 ±0.00 | 1.01 ±0.00 | 1.02 ±0.04 | 1.00 / 0.84 |
| | 256 D | 1.01 ±0.01 | 1.03 ±0.01 | 1.00 ±0.00 | 1.00 ±0.00 | 1.00 ±0.01 | 1.00 ±0.00 | 0.97 ±0.03 | 1.00 ±0.01 | 1.00 ±0.00 | 1.01 ±0.00 | 1.00 ±0.02 | 1.00 / 0.68 |
| 50 full (F) | 16 F | 0.99 ±0.04 | 1.00 ±0.01 | 1.00 ±0.01 | 0.96 ±0.01 | 0.95 ±0.02 | 0.95 ±0.01 | 0.65 ±0.09 | 1.01 ±0.15 | 0.97 ±0.02 | 0.88 ±0.01 | 0.94 ±0.11 | 1.00/0.98 |
| | 32 F | 1.02 ±0.00 | 1.00 ±0.00 | 1.00 ±0.00 | 1.00 ±0.00 | 0.98 ±0.01 | 0.97 ±0.00 | 0.96 ±0.01 | 1.09 ±0.03 | 1.01 ±0.02 | 0.93 ±0.00 | 0.99 ±0.04 | 0.99/0.95 |
| | 64 F | 1.02 ±0.01 | 1.06 ±0.02 | 1.00 ±0.00 | 1.00 ±0.00 | 0.99 ±0.01 | 0.98 ±0.00 | 1.03 ±0.01 | 1.11 ±0.00 | 1.00 ±0.00 | 0.99 ±0.01 | 1.02 ±0.04 | 0.97/0.89 |
| | 128 F | 1.02 ±0.00 | 1.06 ±0.01 | 1.00 ±0.00 | 1.00 ±0.00 | 1.00 ±0.00 | 1.00 ±0.00 | 0.98 ±0.01 | 1.03 ±0.04 | 1.00 ±0.01 | 1.01 ±0.00 | 1.01 ±0.02 | 0.88/0.72 |
| | 256 F | 1.00 ±0.00 | 1.02 ±0.00 | 1.00 ±0.00 | 1.00 ±0.00 | 1.00 ±0.00 | 1.00 ±0.00 | 0.99 ±0.00 | 1.01 ±0.01 | 1.00 ±0.00 | 1.01 ±0.00 | 1.00 ±0.01 | 0.50/0.18 |
| 100 diagonal (D) | 16 D | 0.80 ±0.07 | 0.89 ±0.06 | 0.93 ±0.03 | 0.96 ±0.01 | 0.51 ±0.09 | 0.81 ±0.02 | 0.30 ±0.07 | 0.54 ±0.11 | 0.78 ±0.03 | 0.83 ±0.02 | 0.73 ±0.21 | 1.00 / 0.99 |
| | 32 D | 0.95 ±0.06 | 0.98 ±0.01 | 1.00 ±0.00 | 0.91 ±0.06 | 0.80 ±0.13 | 0.91 ±0.05 | 0.62 ±0.10 | 0.78 ±0.25 | 0.91 ±0.02 | 0.85 ±0.01 | 0.87 ±0.14 | 1.00 / 0.98 |
| | 64 D | 1.01 ±0.03 | 1.01 ±0.01 | 1.00 ±0.00 | 0.98 ±0.02 | 0.96 ±0.01 | 0.95 ±0.01 | 0.90 ±0.05 | 1.11 ±0.07 | 0.96 ±0.02 | 0.88 ±0.02 | 0.98 ±0.07 | 1.00 / 0.96 |
| | 128 D | 1.01 ±0.05 | 1.02 ±0.01 | 1.00 ±0.00 | 1.00 ±0.00 | 0.99 ±0.01 | 0.98 ±0.00 | 1.00 ±0.03 | 1.11 ±0.02 | 1.00 ±0.00 | 0.92 ±0.00 | 1.00 ±0.04 | 1.00 / 0.92 |
| | 256 D | 1.00 ±0.00 | 1.06 ±0.00 | 1.00 ±0.00 | 1.00 ±0.00 | 0.99 ±0.00 | 0.98 ±0.00 | 1.00 ±0.01 | 1.11 ±0.03 | 1.00 ±0.01 | 0.99 ±0.02 | 1.01 ±0.04 | 1.00 / 0.84 |
| 100 full (F) | 16 F | 0.95 ±0.01 | 0.97 ±0.03 | 0.97 ±0.03 | 0.97 ±0.03 | 0.93 ±0.01 | 0.93 ±0.01 | 0.66 ±0.03 | 0.90 ±0.16 | 0.87 ±0.02 | 0.85 ±0.01 | 0.90 ±0.10 | 1.00/0.99 |
| | 32 F | 1.00 ±0.02 | 0.99 ±0.01 | 1.00 ±0.00 | 1.00 ±0.00 | 0.97 ±0.01 | 0.96 ±0.00 | 0.87 ±0.03 | 1.12 ±0.03 | 0.89 ±0.00 | 0.98 ±0.07 | 0.98 ±0.07 | 0.99/0.99 |
| | 64 F | 1.02 ±0.00 | 1.00 ±0.02 | 1.00 ±0.00 | 1.00 ±0.00 | 0.98 ±0.00 | 0.97 ±0.00 | 0.99 ±0.01 | 1.10 ±0.01 | 0.99 ±0.02 | 0.93 ±0.01 | 1.00 ±0.04 | 0.97/0.93 |
| | 128 F | 1.01 ±0.01 | 1.05 ±0.01 | 1.00 ±0.00 | 1.00 ±0.00 | 1.00 ±0.00 | 1.00 ±0.00 | 0.98 ±0.00 | 1.03 ±0.01 | 1.01 ±0.00 | 1.00 ±0.02 | 1.00 ±0.03 | 0.88/0.80 |
| | 256 F | 1.01 ±0.01 | 1.03 ±0.01 | 1.00 ±0.00 | 1.00 ±0.00 | 1.01 ±0.00 | 1.00 ±0.00 | 0.98 ±0.00 | 1.00 ±0.03 | 1.01 ±0.00 | 1.01 ±0.00 | 1.00 ±0.01 | 0.50/0.34 |
| 100 w/clusters (C) | 16 C 5 | 1.12 | 1.02 | 1.00 | 1.00 | 0.98 | 0.97 | 1.01 | 1.19 | 1.04 | 0.94 | 1.03 | 1.00/0.95 |
| | 16 C 7 | 1.12 | 1.02 | 1.00 | 1.00 | 1.01 | 0.98 | 1.01 | 1.29 | 1.03 | 0.97 | 1.04 | 1.00/0.93 |
| 500 diagonal (D) | 16 D | 0.57 ±0.07 | 0.55 ±0.03 | 0.83 ±0.04 | 0.78 ±0.16 | 0.85 ±0.04 | 0.73 ±0.07 | 0.24 ±0.02 | 0.45 ±0.01 | 0.76 ±0.06 | 0.81 ±0.00 | 0.66 ±0.20 | 1.00 / 1.00 |
| | 32 D | 0.61 ±0.12 | 0.55 ±0.08 | 0.83 ±0.02 | 0.84 ±0.12 | 0.91 ±0.02 | 0.75 ±0.05 | 0.30 ±0.05 | 0.49 ±0.07 | 0.79 ±0.04 | 0.82 ±0.01 | 0.69 ±0.20 | 1.00 / 1.00 |
| | 64 D | 0.73 ±0.02 | 0.63 ±0.11 | 0.89 ±0.04 | 0.97 ±0.02 | 0.94 ±0.03 | 0.86 ±0.03 | 0.46 ±0.09 | 0.51 ±0.07 | 0.82 ±0.02 | 0.83 ±0.01 | 0.77 ±0.18 | 1.00 / 0.99 |
| | 128 D | 0.84 ±0.00 | 0.92 ±0.02 | 0.97 ±0.03 | 0.98 ±0.01 | 0.94 ±0.00 | 0.90 ±0.02 | 0.62 ±0.14 | 0.54 ±0.01 | 0.85 ±0.05 | 0.83 ±0.01 | 0.84 ±0.15 | 1.00 / 0.98 |
| | 256 D | 0.99 ±0.03 | 0.99 ±0.00 | 1.00 ±0.00 | 1.00 ±0.00 | 0.96 ±0.00 | 0.96 ±0.00 | 0.68 ±0.05 | 0.85 ±0.14 | 0.92 ±0.02 | 0.85 ±0.00 | 0.92 ±0.11 | 1.00 / 0.97 |
| 500 full (F) | 16 F | 0.57 ±0.01 | 0.43 ±0.07 | 0.78 ±0.01 | 0.97 ±0.00 | 0.96 ±0.00 | 0.86 ±0.01 | 0.65 ±0.00 | 0.55 ±0.02 | 0.83 ±0.01 | 0.84 ±0.00 | 0.76 ±0.17 | 1.00/1.00 |
| | 32 F | 0.79 ±0.05 | 0.54 ±0.04 | 0.93 ±0.02 | 0.98 ±0.00 | 0.97 ±0.00 | 0.92 ±0.00 | 0.70 ±0.01 | 0.52 ±0.00 | 0.86 ±0.02 | 0.85 ±0.00 | 0.81 ±0.16 | 0.99/0.99 |
| | 64 F | 1.02 ±0.00 | 0.96 ±0.01 | 0.94 ±0.01 | 1.00 ±0.00 | 0.96 ±0.00 | 0.97 ±0.01 | 0.74 ±0.01 | 0.55 ±0.01 | 0.91 ±0.01 | 0.87 ±0.00 | 0.89 ±0.14 | 0.97/0.96 |
| | 128 F | 1.03 ±0.02 | 0.97 ±0.02 | 0.99 ±0.00 | 1.00 ±0.00 | 0.98 ±0.00 | 0.98 ±0.00 | 0.88 ±0.01 | 1.07 ±0.02 | 0.90 ±0.00 | 0.90 ±0.00 | 0.98 ±0.05 | 0.88/0.86 |
| | 256 F | 1.03 ±0.00 | 1.03 ±0.01 | 1.00 ±0.00 | 1.00 ±0.00 | 0.99 ±0.01 | 0.97 ±0.00 | 1.00 ±0.02 | 1.04 ±0.02 | 1.00 ±0.01 | 0.93 ±0.00 | 1.00 ±0.03 | 0.50/0.47 |
| 500 w/clusters (C) | 16 C 7 | 1.09 | 1.00 | 0.99 | 1.00 | 0.98 | 0.96 | 0.72 | 0.88 | 0.98 | 0.93 | 0.95 | 1.00/0.98 |
| | 16 C 10 | 1.10 | 1.01 | 1.00 | 0.99 | 0.97 | 0.94 | 0.72 | 1.29 | 1.02 | 0.92 | 1.00 | 1.00/0.98 |
| | 16 C 25 | 1.10 | 1.00 | 1.00 | 0.99 | 0.99 | 0.97 | 0.98 | 1.30 | 1.03 | 0.96 | 1.03 | 1.00/0.95 |
| | 64 C 5 | 1.09 | 0.98 | 1.00 | 1.00 | 0.99 | 0.97 | 0.99 | 1.17 | 1.04 | 0.93 | 1.02 | 0.97/0.93 |
| | 64 C 7 | 1.12 | 1.02 | 1.00 | 1.00 | 1.00 | 0.97 | 1.00 | 1.22 | 1.04 | 0.99 | 1.04 | 0.97/0.91 |

Table 6: **Relative** In-Distribution ROUGE-1 scores for various tasks and methods

| Model Type | Method Type | Tasks | | | | | | | | | | Average | Para. Saved |
|---|---|---|---|---|---|---|---|---|---|---|---|---|---|
| | | task039 | task190 | task280 | task290 | task391 | task442 | task620 | task1342 | task1391 | task1598 | | |
| | base | 24.44 ± 0.00 | 1.60 ± 0.00 | 19.13 ± 0.00 | 39.22 ± 0.00 | 10.42 ± 0.00 | 39.88 ± 0.00 | 8.05 ± 0.00 | 6.96 ± 0.00 | 17.82 ± 0.00 | 55.03 ± 0.00 | 22.43 ± 16.49 | 1.00 / 1.00 |
| | lora | 95.00 ± 0.00 | 86.00 ± 0.00 | 99.00 ± 0.00 | 93.67 ± 0.00 | 94.33 ± 0.00 | 78.43 ± 0.00 | 74.90 ± 0.00 | 26.87 ± 0.00 | 95.00 ± 0.00 | 68.66 ± 0.00 | 81.14 ± 20.67 | 0.00/0.00 |
| TIES | 10 | 76.50 ± 0.00 | 49.00 ± 1.73 | 44.33 ± 4.04 | 9.80 ± 0.58 | 78.56 ± 0.96 | 40.44 ± 0.00 | 53.10 ± 0.67 | 15.48 ± 0.12 | 77.67 ± 1.15 | 54.89 ± 0.06 | 49.98 ± 23.33 | 1.00 / 1.00 |
| | 50 | 55.80 ± 0.00 | 35.00 ± 0.00 | 18.00 ± 5.20 | 2.42 ± 0.50 | 85.78 ± 0.96 | 26.75 ± 0.00 | 49.96 ± 0.00 | 16.73 ± 0.00 | 30.00 ± 3.46 | 53.87 ± 0.02 | 37.43 ± 23.49 | 1.00 / 1.00 |
| | 100 | 52.43 ± 0.00 | 34.00 ± 0.00 | 19.67 ± 4.62 | 1.09 ± 1.66 | 83.33 ± 0.00 | 28.57 ± 0.00 | 48.89 ± 0.00 | 15.18 ± 0.42 | 1.00 ± 0.00 | 53.44 ± 0.02 | 33.76 ± 25.22 | 1.00 / 1.00 |
| | 500 | 35.18 ± 0.00 | 22.00 ± 0.00 | 1.00 ± 0.00 | 0.00 ± 0.00 | 78.00 ± 0.00 | 24.32 ± 0.00 | 43.80 ± 0.04 | 9.96 ± 0.13 | 1.00 ± 0.00 | 27.90 ± 0.03 | 24.40 ± 23.79 | 1.00 / 1.00 |
| SVD | SVD 2 | 93.15 ± 2.77 | 92.24 ± 1.85 | 99.09 ± 0.18 | 93.44 ± 0.14 | 93.89 ± 0.35 | 77.33 ± 0.29 | 75.40 ± 1.01 | 26.90 ± 2.68 | 95.06 ± 1.35 | 67.71 ± 0.49 | 81.33 ± 20.85 | 0.88 / 0.88 |
| | SVD 4 | 94.01 ± 3.60 | 89.21 ± 0.71 | 99.05 ± 0.09 | 93.65 ± 0.03 | 94.66 ± 0.63 | 78.42 ± 0.23 | 74.09 ± 1.12 | 26.47 ± 2.06 | 93.98 ± 0.77 | 69.37 ± 0.21 | 81.22 ± 20.80 | 0.75 / 0.75 |
| | SVD 8 | 95.00 ± 0.00 | 87.40 ± 0.59 | 99.05 ± 0.09 | 93.65 ± 0.03 | 94.36 ± 0.38 | 78.21 ± 0.03 | 75.57 ± 0.00 | 26.88 ± 0.27 | 95.51 ± 1.09 | 69.33 ± 0.08 | 83.02 ± 19.87 | 0.50 / 0.50 |
| | SVD 16 | 95.00 ± 0.00 | 86.00 ± 0.00 | 99.00 ± 0.00 | 93.67 ± 0.00 | 94.33 ± 0.00 | 78.44 ± 0.03 | 74.73 ± 0.18 | 26.87 ± 0.00 | 95.00 ± 0.00 | 68.62 ± 0.04 | 80.76 ± 21.05 | 0.00 / 0.00 |
| 10 diagonal (D) | 16 D | 96.67 ± 0.58 | 87.00 ± 1.00 | 99.00 ± 0.00 | 94.00 ± 0.67 | 93.11 ± 0.38 | 76.08 ± 0.17 | 77.26 ± 1.47 | 30.15 ± 0.72 | 94.00 ± 1.73 | 68.25 ± 0.18 | 81.55 ± 20.03 | 1.00 / 0.90 |
| | 32 D | 95.67 ± 0.58 | 90.00 ± 1.00 | 99.00 ± 0.00 | 93.00 ± 0.33 | 94.89 ± 0.51 | 77.46 ± 0.24 | 72.53 ± 1.00 | 27.98 ± 0.71 | 94.67 ± 0.58 | 69.16 ± 0.41 | 81.44 ± 20.80 | 1.00 / 0.80 |
| | 64 D | 95.00 ± 0.00 | 88.33 ± 0.58 | 99.00 ± 0.00 | 93.67 ± 0.00 | 94.78 ± 0.38 | 78.28 ± 0.07 | 75.47 ± 0.58 | 26.53 ± 0.25 | 96.00 ± 0.00 | 69.36 ± 0.05 | 81.64 ± 21.06 | 1.00 / 0.60 |
| | 128 D | 95.00 ± 0.00 | 86.67 ± 0.58 | 99.00 ± 0.00 | 93.67 ± 0.00 | 94.33 ± 0.00 | 78.45 ± 0.16 | 75.46 ± 0.51 | 26.64 ± 0.23 | 95.00 ± 0.00 | 68.70 ± 0.14 | 81.29 ± 20.92 | 1.00 / 0.20 |
| | 256 D | 95.00 ± 0.00 | 86.00 ± 0.00 | 99.00 ± 0.00 | 93.67 ± 0.00 | 94.33 ± 0.00 | 78.43 ± 0.00 | 74.90 ± 0.00 | 26.87 ± 0.00 | 95.00 ± 0.00 | 68.59 ± 0.03 | 81.18 ± 20.86 | 1.00 / -0.60 |
| 10 full (F) | 16 F | 97.00 ± 0.00 | 91.00 ± 1.00 | 99.00 ± 0.00 | 93.56 ± 0.19 | 93.56 ± 0.69 | 77.64 ± 0.25 | 75.78 ± 1.25 | 28.71 ± 0.09 | 96.00 ± 1.00 | 68.69 ± 0.08 | 82.09 ± 20.68 | 1.00/0.90 |
| | 32 F | 96.67 ± 0.58 | 89.33 ± 0.58 | 99.00 ± 0.00 | 93.22 ± 0.19 | 94.44 ± 0.19 | 77.84 ± 0.21 | 72.24 ± 0.59 | 26.84 ± 0.50 | 94.67 ± 0.58 | 69.55 ± 0.08 | 81.38 ± 21.11 | 0.99/0.79 |
| | 64 F | 95.00 ± 0.00 | 88.67 ± 0.58 | 99.00 ± 0.00 | 93.67 ± 0.00 | 94.56 ± 0.38 | 78.19 ± 0.08 | 75.97 ± 0.58 | 26.43 ± 0.34 | 96.00 ± 0.00 | 69.38 ± 0.11 | 81.69 ± 21.07 | 0.97/0.57 |
| | 128 F | 95.00 ± 0.00 | 86.67 ± 0.58 | 99.00 ± 0.00 | 93.67 ± 0.00 | 94.33 ± 0.00 | 78.46 ± 0.00 | 74.90 ± 0.00 | 26.72 ± 0.13 | 95.00 ± 0.00 | 68.65 ± 0.03 | 81.24 ± 20.91 | 0.88/0.07 |
| | 256 F | 95.00 ± 0.00 | 86.00 ± 0.00 | 99.00 ± 0.00 | 93.67 ± 0.00 | 94.33 ± 0.00 | 78.44 ± 0.03 | 74.79 ± 0.19 | 26.87 ± 0.00 | 95.00 ± 0.00 | 68.64 ± 0.03 | 81.17 ± 20.86 | 0.50/-1.10 |
| 50 diagonal (D) | 16 D | 92.76 ± 3.53 | 84.67 ± 1.15 | 99.00 ± 0.00 | 86.17 ± 5.81 | 79.83 ± 6.08 | 73.55 ± 1.39 | 51.72 ± 3.78 | 23.75 ± 2.66 | 83.90 ± 6.43 | 59.05 ± 0.94 | 73.44 ± 22.08 | 1.00 / 0.98 |
| | 32 D | 95.33 ± 2.08 | 87.33 ± 2.08 | 99.00 ± 0.00 | 92.60 ± 0.29 | 90.35 ± 1.00 | 75.43 ± 1.33 | 63.84 ± 1.64 | 26.97 ± 3.21 | 93.33 ± 1.15 | 61.94 ± 0.32 | 78.61 ± 21.60 | 1.00 / 0.96 |
| | 64 D | 97.00 ± 0.00 | 90.33 ± 1.53 | 99.00 ± 0.00 | 93.78 ± 0.19 | 93.00 ± 0.58 | 76.27 ± 0.49 | 74.39 ± 0.90 | 29.28 ± 0.81 | 95.67 ± 0.58 | 64.84 ± 0.27 | 81.36 ± 20.83 | 1.00 / 0.92 |
| | 128 D | 96.33 ± 0.58 | 92.67 ± 0.58 | 99.00 ± 0.00 | 93.56 ± 0.19 | 93.00 ± 0.58 | 77.24 ± 0.19 | 73.76 ± 1.25 | 29.58 ± 0.93 | 95.00 ± 0.00 | 69.04 ± 0.54 | 81.92 ± 20.44 | 1.00 / 0.84 |
| | 256 D | 95.67 ± 0.58 | 88.33 ± 0.58 | 99.00 ± 0.00 | 93.56 ± 0.19 | 94.67 ± 0.67 | 78.45 ± 0.14 | 72.86 ± 2.07 | 27.00 ± 0.77 | 95.33 ± 0.58 | 69.61 ± 0.18 | 81.45 ± 21.00 | 1.00 / 0.68 |
| 50 full (F) | 16 F | 94.06 ± 3.54 | 85.67 ± 1.15 | 98.67 ± 0.58 | 90.35 ± 1.37 | 89.97 ± 1.78 | 74.46 ± 0.58 | 49.03 ± 7.07 | 27.14 ± 3.94 | 92.33 ± 1.53 | 60.26 ± 1.03 | 76.19 ± 22.80 | 1.00/0.98 |
| | 32 F | 97.00 ± 0.00 | 85.67 ± 1.53 | 99.00 ± 0.00 | 93.67 ± 0.00 | 92.22 ± 0.69 | 75.86 ± 0.22 | 71.68 ± 0.65 | 29.26 ± 0.70 | 95.67 ± 1.53 | 63.88 ± 0.10 | 80.39 ± 20.81 | 0.99/0.95 |
| | 64 F | 96.67 ± 0.58 | 91.00 ± 2.00 | 99.00 ± 0.00 | 93.56 ± 0.19 | 93.22 ± 0.51 | 77.17 ± 0.38 | 77.11 ± 0.51 | 29.75 ± 0.03 | 95.33 ± 0.58 | 68.13 ± 0.75 | 82.09 ± 20.33 | 0.97/0.89 |
| | 128 F | 97.00 ± 0.00 | 91.00 ± 1.00 | 99.00 ± 0.00 | 93.33 ± 0.00 | 94.11 ± 0.51 | 77.23 ± 0.17 | 73.67 ± 0.58 | 27.62 ± 1.12 | 95.00 ± 1.00 | 69.40 ± 0.16 | 81.74 ± 20.97 | 0.88/0.72 |
| | 256 F | 95.00 ± 0.00 | 88.00 ± 0.00 | 99.00 ± 0.00 | 93.67 ± 0.00 | 94.44 ± 0.19 | 77.97 ± 0.24 | 75.47 ± 0.58 | 26.96 ± 0.09 | 96.00 ± 0.00 | 69.28 ± 0.05 | 81.58 ± 20.92 | 0.50/0.18 |
| 100 diagonal (D) | 16 D | 76.43 ± 7.07 | 76.67 ± 4.93 | 91.61 ± 2.75 | 89.99 ± 1.07 | 47.89 ± 8.62 | 63.17 ± 1.31 | 22.23 ± 5.27 | 14.46 ± 2.89 | 73.93 ± 3.13 | 57.17 ± 1.05 | 61.35 ± 25.78 | 1.00 / 0.99 |
| | 32 D | 90.10 ± 5.85 | 84.00 ± 1.00 | 99.00 ± 0.00 | 85.52 ± 5.34 | 75.88 ± 12.57 | 71.15 ± 3.61 | 46.10 ± 7.39 | 21.04 ± 6.76 | 86.67 ± 1.86 | 58.64 ± 1.02 | 71.81 ± 23.39 | 1.00 / 0.98 |
| | 64 D | 95.56 ± 2.49 | 86.67 ± 0.58 | 99.00 ± 0.00 | 92.24 ± 1.68 | 90.89 ± 1.17 | 74.57 ± 0.50 | 67.07 ± 3.81 | 29.78 ± 1.92 | 91.67 ± 2.31 | 60.28 ± 1.51 | 78.77 ± 20.77 | 1.00 / 0.96 |
| | 128 D | 96.00 ± 0.00 | 87.33 ± 1.15 | 99.00 ± 0.00 | 93.89 ± 0.19 | 93.00 ± 0.58 | 76.68 ± 0.18 | 74.84 ± 2.23 | 29.79 ± 0.50 | 93.67 ± 0.58 | 63.49 ± 0.34 | 81.67 ± 20.28 | 1.00 / 0.92 |
| | 256 D | 95.00 ± 0.00 | 91.00 ± 0.00 | 99.00 ± 0.00 | 93.56 ± 0.19 | 93.11 ± 0.19 | 76.93 ± 0.23 | 75.13 ± 0.84 | 29.75 ± 0.73 | 95.33 ± 0.58 | 67.89 ± 1.34 | 81.67 ± 20.28 | 1.00 / 0.84 |
| 100 full (F) | 16 F | 90.70 ± 1.07 | 83.00 ± 2.65 | 96.00 ± 3.00 | 91.22 ± 2.94 | 87.94 ± 0.54 | 73.07 ± 0.93 | 49.41 ± 2.04 | 24.17 ± 4.22 | 82.33 ± 2.08 | 58.18 ± 0.44 | 73.60 ± 22.23 | 1.00/0.99 |
| | 32 F | 95.33 ± 1.53 | 85.00 ± 1.00 | 99.00 ± 0.00 | 93.50 ± 0.22 | 91.44 ± 0.84 | 75.00 ± 0.19 | 65.09 ± 2.23 | 30.20 ± 0.81 | 91.67 ± 0.58 | 60.92 ± 0.26 | 78.72 ± 20.72 | 0.99/0.97 |
| | 64 F | 97.00 ± 0.00 | 85.67 ± 1.53 | 99.00 ± 0.00 | 93.78 ± 0.19 | 92.56 ± 0.19 | 76.01 ± 0.13 | 73.96 ± 0.89 | 29.46 ± 0.21 | 94.33 ± 1.53 | 64.07 ± 0.37 | 80.58 ± 20.59 | 0.97/0.93 |
| | 128 F | 96.33 ± 0.58 | 90.33 ± 0.58 | 99.00 ± 0.00 | 93.00 ± 0.00 | 93.89 ± 0.19 | 77.04 ± 0.30 | 77.33 ± 1.01 | 29.49 ± 0.35 | 96.00 ± 0.00 | 68.76 ± 0.25 | 82.12 ± 20.35 | 0.88/0.80 |
| | 256 F | 96.33 ± 0.58 | 88.67 ± 0.58 | 99.00 ± 0.00 | 93.67 ± 0.00 | 94.89 ± 0.19 | 78.16 ± 0.18 | 73.40 ± 0.12 | 26.86 ± 0.68 | 95.00 ± 0.00 | 69.47 ± 0.23 | 81.64 ± 21.15 | 0.50/0.34 |
| 100 w/clusters (C) | 16 C 5 | 98.00 | 88.00 | 99.00 | 93.38 | 91.67 | 75.97 | 77.63 | 27.91 | 96.00 | 64.18 | 81.17 | 1.00/0.95 |
| | 16 C 7 | 98.00 | 88.00 | 99.00 | 93.67 | 94.00 | 77.67 | 77.67 | 30.12 | 95.00 | 66.78 | 81.91 | 1.00/0.93 |
| 500 diagonal (D) | 16 D | 54.44 ± 6.87 | 47.00 ± 2.83 | 82.21 ± 3.59 | 73.38 ± 14.97 | 80.13 ± 3.68 | 57.42 ± 5.29 | 18.33 ± 1.33 | 12.19 ± 0.30 | 72.67 ± 6.03 | 55.79 ± 0.20 | 55.64 ± 24.25 | 1.00 / 1.00 |
| | 32 D | 58.08 ± 11.52 | 47.00 ± 7.07 | 82.06 ± 1.69 | 78.62 ± 11.23 | 85.57 ± 1.48 | 59.19 ± 3.70 | 22.76 ± 3.95 | 13.15 ± 1.94 | 75.33 ± 4.04 | 56.07 ± 0.52 | 58.16 ± 24.56 | 1.00 / 1.00 |
| | 64 D | 69.21 ± 2.03 | 54.50 ± 9.19 | 88.33 ± 4.04 | 91.11 ± 0.38 | 88.78 ± 0.38 | 67.71 ± 2.59 | 34.79 ± 6.86 | 13.80 ± 1.95 | 77.67 ± 2.31 | 56.78 ± 0.73 | 64.61 ± 24.79 | 1.00 / 0.99 |
| | 128 D | 79.77 ± 0.37 | 79.50 ± 2.12 | 95.89 ± 2.83 | 91.89 ± 1.39 | 88.67 ± 0.00 | 70.27 ± 1.73 | 46.64 ± 10.58 | 14.63 ± 0.25 | 81.00 ± 5.00 | 56.88 ± 0.55 | 70.20 ± 24.63 | 1.00 / 0.98 |
| | 256 D | 93.83 ± 2.52 | 85.00 ± 0.00 | 99.00 ± 0.00 | 93.78 ± 0.19 | 90.56 ± 0.38 | 73.25 ± 1.86 | 51.14 ± 3.86 | 22.93 ± 3.86 | 87.33 ± 2.31 | 58.48 ± 0.20 | 75.20 ± 23.90 | 1.00 / 0.97 |
| 500 full (F) | 16 F | 54.30 ± 1.13 | 37.00 ± 5.66 | 77.67 ± 0.58 | 91.00 ± 0.00 | 90.56 ± 0.19 | 67.63 ± 0.45 | 48.81 ± 0.35 | 14.70 ± 0.65 | 79.00 ± 1.00 | 57.66 ± 0.19 | 62.69 ± 23.46 | 1.00/1.00 |
| | 32 F | 75.10 ± 4.92 | 46.50 ± 3.54 | 91.67 ± 1.53 | 91.56 ± 0.19 | 91.56 ± 0.38 | 72.03 ± 0.15 | 52.63 ± 0.86 | 13.93 ± 0.02 | 81.67 ± 1.53 | 58.50 ± 0.20 | 68.24 ± 24.29 | 0.99/0.99 |
| | 64 F | 96.94 ± 0.42 | 82.50 ± 0.71 | 93.33 ± 0.58 | 93.89 ± 0.69 | 90.67 ± 0.00 | 75.99 ± 0.64 | 55.63 ± 1.07 | 14.74 ± 0.27 | 86.33 ± 0.58 | 59.43 ± 0.05 | 74.69 ± 25.01 | 0.97/0.96 |
| | 128 F | 97.67 ± 0.58 | 83.50 ± 2.12 | 98.00 ± 0.00 | 93.56 ± 0.19 | 92.00 ± 0.00 | 75.80 ± 0.16 | 66.19 ± 0.81 | 28.67 ± 0.49 | 93.00 ± 0.00 | 61.53 ± 0.13 | 78.84 ± 21.50 | 0.88/0.86 |
| | 256 F | 98.00 ± 0.00 | 88.50 ± 0.71 | 99.00 ± 0.00 | 93.78 ± 0.19 | 93.00 ± 0.88 | 76.33 ± 0.29 | 74.60 ± 1.21 | 27.82 ± 0.42 | 95.33 ± 0.58 | 63.70 ± 0.14 | 80.75 ± 21.60 | 0.50/0.47 |
| 500 w/clusters (C) | 16 C 7 | 95.00 | 86.00 | 98.00 | 93.67 | 91.67 | 75.10 | 55.52 | 20.50 | 90.00 | 63.57 | 76.90 | 1.00/0.98 |
| | 16 C 10 | 96.00 | 87.00 | 99.00 | 93.00 | 91.33 | 74.17 | 55.14 | 30.29 | 94.00 | 63.09 | 78.30 | 1.00/0.98 |
| | 16 C 25 | 96.00 | 86.00 | 99.00 | 92.71 | 93.00 | 76.42 | 75.42 | 30.40 | 95.00 | 66.07 | 81.00 | 1.00/0.95 |
| | 64 C 5 | 95.00 | 84.00 | 99.00 | 93.67 | 92.67 | 76.45 | 76.43 | 27.49 | 96.00 | 64.10 | 80.48 | 0.97/0.93 |
| | 64 C 7 | 98.00 | 88.00 | 99.00 | 94.00 | 93.33 | 76.42 | 76.67 | 28.48 | 96.00 | 68.00 | 81.79 | 0.97/0.91 |

Table 7: **Absolute** In-Distribution ROUGE-1 scores for various tasks and methods

| Model Type | Method Type | task039 | task190 | task280 | task290 | task391 | task442 | task620 | task1342 | task1391 | task1598 | Average | Para. Saved |
|---|---|---|---|---|---|---|---|---|---|---|---|---|---|
| | base | 0.00 ±0.00 | 0.00 ±0.00 | 0.02 ±0.00 | 0.00 ±0.00 | 0.00 ±0.00 | 0.00 ±0.00 | 0.00 ±0.00 | 0.00 ±0.00 | 0.00 ±0.00 | 0.00 ±0.00 | 0.00 ±0.01 | 1.00 / 1.00 |
| | lora | 1.00 ±0.00 | 1.00 ±0.00 | 1.00 ±0.00 | 1.00 ±0.00 | 1.00 ±0.00 | 1.00 ±0.00 | 1.00 ±0.00 | 1.00 ±0.00 | 1.00 ±0.00 | 1.00 ±0.00 | 1.00 ±0.00 | 0.00 / 0.00 |
| TIES | 10 | 0.69 ±0.00 | 0.57 ±0.02 | 0.45 ±0.04 | 0.10 ±0.01 | 0.57 ±0.03 | 0.00 ±0.00 | 0.39 ±0.01 | 0.21 ±0.00 | 0.82 ±0.01 | 0.00 ±0.00 | 0.38 ±0.28 | 1.00 / 1.00 |
| | 50 | 0.45 ±0.00 | 0.41 ±0.00 | 0.18 ±0.05 | 0.03 ±0.01 | 0.70 ±0.02 | 0.00 ±0.00 | 0.36 ±0.00 | 0.21 ±0.00 | 0.32 ±0.04 | 0.00 ±0.00 | 0.27 ±0.22 | 1.00 / 1.00 |
| | 100 | 0.41 ±0.00 | 0.40 ±0.00 | 0.20 ±0.05 | 0.01 ±0.02 | 0.65 ±0.00 | 0.00 ±0.00 | 0.36 ±0.00 | 0.21 ±0.00 | 0.01 ±0.00 | 0.00 ±0.00 | 0.23 ±0.22 | 1.00 / 1.00 |
| | 500 | 0.22 ±0.00 | 0.26 ±0.00 | 0.01 ±0.00 | 0.00 ±0.00 | 0.60 ±0.00 | 0.00 ±0.00 | 0.32 ±0.00 | 0.07 ±0.00 | 0.01 ±0.00 | 0.00 ±0.00 | 0.15 ±0.20 | 1.00 / 1.00 |
| SVD | SVD 2 | 0.98 ±0.03 | 1.07 ±0.02 | 1.00 ±0.00 | 0.99 ±0.01 | 0.98 ±0.01 | 0.98 ±0.03 | 0.94 ±0.01 | 1.03 ±0.17 | 1.00 ±0.01 | 0.15 ±0.29 | 0.91 ±0.28 | 0.88 / 0.88 |
| | SVD 4 | 0.99 ±0.04 | 1.04 ±0.01 | 1.00 ±0.00 | 1.00 ±0.00 | 1.01 ±0.02 | 1.11 ±0.00 | 0.97 ±0.02 | 0.99 ±0.13 | 0.99 ±0.01 | 0.90 ±0.17 | 1.00 ±0.08 | 0.75 / 0.75 |
| | SVD 8 | 1.00 ±0.00 | 1.02 ±0.01 | 1.00 ±0.00 | 1.00 ±0.00 | 1.00 ±0.01 | 1.02 ±0.05 | 1.00 ±0.00 | 1.00 ±0.00 | 1.01 ±0.01 | 1.00 ±0.00 | 1.00 ±0.02 | 0.50 / 0.50 |
| | SVD 16 | 1.00 ±0.00 | 1.00 ±0.00 | 1.00 ±0.00 | 1.00 ±0.00 | 1.00 ±0.00 | 1.00 ±0.00 | 0.99 ±0.01 | 1.00 ±0.00 | 1.00 ±0.00 | 1.00 ±0.00 | 1.00 ±0.00 | 0.00 / 0.00 |
| 10 diagonal (D) | 16 D | 1.02 ±0.01 | 1.01 ±0.01 | 1.00 ±0.00 | 1.01 ±0.02 | 0.96 ±0.01 | 1.11 ±0.11 | 0.89 ±0.03 | 1.19 ±0.04 | 0.99 ±0.02 | 0.33 ±0.58 | 0.95 ±0.27 | 1.00 / 0.90 |
| | 32 D | 1.01 ±0.01 | 1.05 ±0.01 | 1.00 ±0.00 | 0.98 ±0.01 | 1.02 ±0.02 | 1.11 ±0.00 | 0.93 ±0.01 | 1.10 ±0.04 | 1.00 ±0.01 | 0.67 ±0.58 | 0.98 ±0.19 | 1.00 / 0.80 |
| | 64 D | 1.00 ±0.00 | 1.03 ±0.01 | 1.00 ±0.00 | 1.00 ±0.00 | 1.02 ±0.01 | 1.11 ±0.00 | 0.99 ±0.01 | 1.00 ±0.00 | 1.01 ±0.00 | 0.67 ±0.58 | 0.98 ±0.19 | 1.00 / 0.60 |
| | 128 D | 1.00 ±0.00 | 1.01 ±0.01 | 1.00 ±0.00 | 1.00 ±0.00 | 1.00 ±0.00 | 1.00 ±0.00 | 1.00 ±0.01 | 1.00 ±0.00 | 1.00 ±0.00 | 1.00 ±0.00 | 1.00 ±0.00 | 1.00 / 0.20 |
| | 256 D | 1.00 ±0.00 | 1.00 ±0.00 | 1.00 ±0.00 | 1.00 ±0.00 | 1.00 ±0.00 | 1.00 ±0.00 | 1.00 ±0.00 | 1.00 ±0.00 | 1.00 ±0.00 | 1.00 ±0.00 | 1.00 ±0.00 | 1.00 / -0.60 |
| 10 full (F) | 16 F | 1.02 ±0.00 | 1.06 ±0.01 | 1.00 ±0.00 | 1.00 ±0.00 | 0.97 ±0.03 | 1.15 ±0.06 | 0.92 ±0.02 | 1.17 ±0.04 | 1.01 ±0.01 | 0.67 ±0.58 | 1.00 ±0.20 | 1.00/0.90 |
| | 32 F | 1.02 ±0.01 | 1.04 ±0.01 | 1.00 ±0.00 | 0.98 ±0.01 | 1.00 ±0.01 | 1.11 ±0.00 | 0.92 ±0.01 | 1.02 ±0.04 | 1.00 ±0.01 | 1.00 ±0.00 | 1.01 ±0.05 | 0.99/0.79 |
| | 64 F | 1.00 ±0.00 | 1.03 ±0.01 | 1.00 ±0.00 | 1.00 ±0.00 | 1.01 ±0.01 | 1.07 ±0.06 | 1.01 ±0.01 | 1.00 ±0.00 | 1.01 ±0.00 | 1.01 ±0.03 | 1.01 ±0.03 | 0.97/0.57 |
| | 128 F | 1.00 ±0.00 | 1.01 ±0.01 | 1.00 ±0.00 | 1.00 ±0.00 | 1.00 ±0.00 | 1.00 ±0.00 | 1.00 ±0.00 | 1.00 ±0.00 | 1.00 ±0.00 | 1.00 ±0.00 | 1.00 ±0.00 | 0.88/0.07 |
| | 256 F | 1.00 ±0.00 | 1.00 ±0.00 | 1.00 ±0.00 | 1.00 ±0.00 | 1.00 ±0.00 | 1.00 ±0.00 | 1.00 ±0.00 | 1.00 ±0.00 | 1.00 ±0.00 | 1.00 ±0.00 | 1.00 ±0.00 | 0.50/-1.10 |
| 50 diagonal (D) | 16 D | 0.91 ±0.06 | 0.98 ±0.01 | 1.00 ±0.00 | 0.91 ±0.09 | 0.78 ±0.05 | 0.89 ±0.29 | 0.34 ±0.06 | 0.50 ±0.45 | 0.86 ±0.07 | 0.00 ±0.00 | 0.72 ±0.35 | 1.00 / 0.98 |
| | 32 D | 1.00 ±0.02 | 1.02 ±0.02 | 1.00 ±0.00 | 1.00 ±0.01 | 0.90 ±0.03 | 0.85 ±0.42 | 0.56 ±0.04 | 0.98 ±0.23 | 0.98 ±0.01 | 0.00 ±0.00 | 0.83 ±0.34 | 1.00 / 0.96 |
| | 64 D | 1.02 ±0.00 | 1.05 ±0.02 | 1.00 ±0.00 | 1.00 ±0.00 | 0.95 ±0.02 | 1.15 ±0.17 | 0.81 ±0.03 | 1.14 ±0.00 | 1.00 ±0.00 | 0.00 ±0.00 | 0.91 ±0.33 | 1.00 / 0.92 |
| | 128 D | 1.01 ±0.01 | 1.08 ±0.01 | 1.00 ±0.00 | 1.00 ±0.01 | 0.95 ±0.02 | 1.04 ±0.06 | 0.92 ±0.03 | 1.21 ±0.07 | 1.00 ±0.00 | 0.67 ±0.58 | 0.99 ±0.20 | 1.00 / 0.84 |
| | 256 D | 1.01 ±0.01 | 1.03 ±0.01 | 1.00 ±0.00 | 1.00 ±0.00 | 1.01 ±0.02 | 1.11 ±0.06 | 0.95 ±0.04 | 1.02 ±0.04 | 1.00 ±0.01 | 1.00 ±0.00 | 1.01 ±0.04 | 1.00 / 0.68 |
| 50 full (F) | 16 F | 0.96 ±0.05 | 1.00 ±0.01 | 1.00 ±0.01 | 0.95 ±0.04 | 0.87 ±0.01 | 1.04 ±0.06 | 0.31 ±0.08 | 0.98 ±0.23 | 0.97 ±0.02 | 0.00 ±0.00 | 0.81 ±0.35 | 1.00/0.98 |
| | 32 F | 1.02 ±0.00 | 1.00 ±0.00 | 1.00 ±0.00 | 1.00 ±0.00 | 0.92 ±0.01 | 1.15 ±0.06 | 0.73 ±0.04 | 1.17 ±0.04 | 1.01 ±0.02 | 0.00 ±0.00 | 0.90 ±0.33 | 0.99/0.95 |
| | 64 F | 1.02 ±0.01 | 1.06 ±0.02 | 1.00 ±0.00 | 1.00 ±0.01 | 0.96 ±0.02 | 1.22 ±0.06 | 0.94 ±0.01 | 1.17 ±0.04 | 1.00 ±0.00 | 0.00 ±0.00 | 0.94 ±0.33 | 0.97/0.89 |
| | 128 F | 1.02 ±0.00 | 1.06 ±0.01 | 1.00 ±0.00 | 0.99 ±0.00 | 0.99 ±0.02 | 1.15 ±0.06 | 0.92 ±0.01 | 1.10 ±0.08 | 1.00 ±0.00 | 1.00 ±0.00 | 1.02 ±0.07 | 0.88/0.72 |
| | 256 F | 1.00 ±0.00 | 1.02 ±0.00 | 1.00 ±0.00 | 1.00 ±0.00 | 1.00 ±0.01 | 1.04 ±0.06 | 0.99 ±0.00 | 1.00 ±0.00 | 1.01 ±0.00 | 1.00 ±0.00 | 1.01 ±0.02 | 0.50/0.18 |
| 100 diagonal (D) | 16 D | 0.54 ±0.16 | 0.89 ±0.06 | 0.90 ±0.04 | 0.89 ±0.05 | 0.42 ±0.08 | 0.44 ±0.00 | 0.08 ±0.02 | 0.00 ±0.00 | 0.76 ±0.05 | 0.00 ±0.00 | 0.49 ±0.36 | 1.00 / 0.99 |
| | 32 D | 0.85 ±0.15 | 0.98 ±0.01 | 1.00 ±0.00 | 0.86 ±0.13 | 0.70 ±0.14 | 0.74 ±0.28 | 0.28 ±0.07 | 0.48 ±0.55 | 0.91 ±0.02 | 0.00 ±0.00 | 0.68 ±0.36 | 1.00 / 0.98 |
| | 64 D | 1.00 ±0.04 | 1.01 ±0.01 | 1.00 ±0.00 | 0.98 ±0.02 | 0.88 ±0.04 | 1.07 ±0.06 | 0.58 ±0.09 | 1.10 ±0.04 | 0.96 ±0.02 | 0.00 ±0.00 | 0.86 ±0.32 | 1.00 / 0.96 |
| | 128 D | 1.01 ±0.00 | 1.02 ±0.01 | 1.00 ±0.00 | 1.00 ±0.01 | 0.95 ±0.02 | 1.11 ±0.00 | 0.81 ±0.06 | 1.21 ±0.00 | 1.00 ±0.00 | 0.00 ±0.00 | 0.91 ±0.33 | 1.00 / 0.92 |
| | 256 D | 1.00 ±0.00 | 1.06 ±0.00 | 1.00 ±0.00 | 1.00 ±0.00 | 0.96 ±0.01 | 1.11 ±0.11 | 0.92 ±0.02 | 1.21 ±0.07 | 1.00 ±0.01 | 0.00 ±0.00 | 0.93 ±0.33 | 1.00 / 0.84 |
| 100 full (F) | 16 F | 0.85 ±0.03 | 0.97 ±0.03 | 0.97 ±0.03 | 0.95 ±0.06 | 0.80 ±0.02 | 0.81 ±0.17 | 0.29 ±0.04 | 0.60 ±0.34 | 0.87 ±0.02 | 0.00 ±0.00 | 0.71 ±0.33 | 1.00/0.99 |
| | 32 F | 0.99 ±0.02 | 0.99 ±0.01 | 1.00 ±0.00 | 1.00 ±0.00 | 0.90 ±0.03 | 1.04 ±0.06 | 0.55 ±0.04 | 1.07 ±0.07 | 0.96 ±0.01 | 0.00 ±0.00 | 0.85 ±0.32 | 0.99/0.97 |
| | 64 F | 1.02 ±0.00 | 1.00 ±0.02 | 1.00 ±0.00 | 1.00 ±0.00 | 0.94 ±0.01 | 1.04 ±0.06 | 0.78 ±0.01 | 1.14 ±0.00 | 0.99 ±0.02 | 0.00 ±0.00 | 0.89 ±0.31 | 0.97/0.93 |
| | 128 F | 1.01 ±0.01 | 1.05 ±0.01 | 1.00 ±0.00 | 0.98 ±0.00 | 0.98 ±0.01 | 1.15 ±0.06 | 0.94 ±0.01 | 1.21 ±0.00 | 1.01 ±0.00 | 0.33 ±0.58 | 0.97 ±0.28 | 0.88/0.80 |
| | 256 F | 1.01 ±0.01 | 1.03 ±0.01 | 1.00 ±0.00 | 1.00 ±0.00 | 1.02 ±0.01 | 1.19 ±0.06 | 0.93 ±0.01 | 1.02 ±0.04 | 1.01 ±0.00 | 1.00 ±0.00 | 1.02 ±0.06 | 0.50/0.34 |
| 100 w/clusters (C) | 16 C 5 | 1.13 | 1.02 | 1.00 | 1.01 | 0.92 | 1.24 | 0.86 | 1.33 | 1.04 | 0.00 | 0.96 | 1.00/0.95 |
| | 16 C 7 | 1.13 | 1.02 | 1.00 | 1.00 | 1.01 | 1.24 | 0.93 | 1.51 | 1.03 | 0.00 | 0.99 | 1.00/0.93 |
| 500 diagonal (D) | 16 D | 0.22 ±0.10 | 0.55 ±0.03 | 0.81 ±0.05 | 0.33 ±0.49 | 0.70 ±0.03 | 0.15 ±0.17 | 0.03 ±0.01 | 0.00 ±0.00 | 0.76 ±0.00 | 0.00 ±0.00 | 0.35 ±0.35 | 1.00 / 1.00 |
| | 32 D | 0.27 ±0.18 | 0.55 ±0.08 | 0.82 ±0.02 | 0.49 ±0.37 | 0.75 ±0.01 | 0.22 ±0.11 | 0.05 ±0.02 | 0.02 ±0.04 | 0.79 ±0.04 | 0.00 ±0.00 | 0.39 ±0.34 | 1.00 / 1.00 |
| | 64 D | 0.40 ±0.04 | 0.63 ±0.11 | 0.89 ±0.04 | 0.91 ±0.01 | 0.80 ±0.01 | 0.48 ±0.06 | 0.13 ±0.04 | 0.05 ±0.08 | 0.82 ±0.02 | 0.00 ±0.00 | 0.51 ±0.35 | 1.00 / 0.99 |
| | 128 D | 0.61 ±0.04 | 0.92 ±0.02 | 0.97 ±0.03 | 0.93 ±0.05 | 0.80 ±0.00 | 0.74 ±0.17 | 0.22 ±0.11 | 0.12 ±0.08 | 0.85 ±0.05 | 0.00 ±0.00 | 0.61 ±0.36 | 1.00 / 0.98 |
| | 256 D | 0.95 ±0.02 | 0.99 ±0.00 | 1.00 ±0.00 | 1.00 ±0.00 | 0.86 ±0.01 | 0.85 ±0.02 | 0.28 ±0.06 | 0.55 ±0.39 | 0.92 ±0.02 | 0.00 ±0.00 | 0.73 ±0.34 | 1.00 / 0.97 |
| 500 full (F) | 16 F | 0.21 ±0.02 | 0.43 ±0.07 | 0.78 ±0.01 | 0.90 ±0.01 | 0.86 ±0.01 | 0.59 ±0.06 | 0.21 ±0.01 | 0.12 ±0.04 | 0.83 ±0.01 | 0.00 ±0.00 | 0.50 ±0.34 | 1.00/1.00 |
| | 32 F | 0.54 ±0.08 | 0.54 ±0.04 | 0.93 ±0.02 | 0.92 ±0.00 | 0.90 ±0.01 | 0.63 ±0.13 | 0.26 ±0.02 | 0.14 ±0.00 | 0.86 ±0.02 | 0.00 ±0.00 | 0.57 ±0.34 | 0.99/0.99 |
| | 64 F | 0.99 ±0.03 | 0.96 ±0.01 | 0.94 ±0.01 | 1.01 ±0.03 | 0.87 ±0.00 | 1.04 ±0.17 | 0.36 ±0.00 | 0.14 ±0.00 | 0.91 ±0.01 | 0.00 ±0.00 | 0.71 ±0.39 | 0.97/0.96 |
| | 128 F | 1.02 ±0.00 | 0.97 ±0.02 | 0.99 ±0.00 | 1.00 ±0.00 | 0.92 ±0.00 | 1.15 ±0.06 | 0.61 ±0.01 | 1.07 ±0.00 | 0.98 ±0.00 | 0.00 ±0.00 | 0.87 ±0.33 | 0.88/0.86 |
| | 256 F | 1.03 ±0.00 | 1.03 ±0.01 | 1.00 ±0.00 | 1.00 ±0.01 | 0.95 ±0.03 | 1.00 ±0.00 | 0.78 ±0.01 | 1.07 ±0.00 | 1.00 ±0.01 | 0.00 ±0.00 | 0.88 ±0.31 | 0.50/0.47 |
| 500 w/clusters (C) | 16 C 7 | 1.08 | 1.00 | 0.99 | 1.00 | 0.92 | 1.01 | 0.39 | 0.62 | 0.98 | 0.00 | 0.80 | 1.00/0.98 |
| | 16 C 10 | 1.10 | 1.01 | 1.00 | 0.98 | 0.91 | 1.01 | 0.37 | 1.51 | 1.02 | 0.00 | 0.89 | 1.00/0.98 |
| | 16 C 25 | 1.10 | 1.00 | 1.00 | 0.99 | 0.97 | 1.12 | 0.81 | 1.42 | 1.03 | 0.00 | 0.95 | 1.00/0.95 |
| | 64 C 5 | 1.09 | 0.98 | 1.00 | 1.00 | 0.96 | 1.12 | 0.83 | 1.33 | 1.04 | 0.00 | 0.94 | 0.97/0.93 |
| | 64 C 7 | 1.13 | 1.02 | 1.00 | 1.01 | 0.98 | 1.12 | 0.90 | 1.42 | 1.04 | 0.00 | 0.96 | 0.97/0.91 |

Table 8: **Relative** In-Distribution exact match scores for various tasks and methods

| Model Type | Method Type | task039 | task190 | task280 | task290 | task391 | task442 | task620 | task1342 | task1391 | task1598 | Average | Para. Saved |
|---|---|---|---|---|---|---|---|---|---|---|---|---|---|
| | base | 8.59 ± 0.08 | 9.15 ± 0.00 | 2.55 ± 0.00 | 2.88 ± 0.00 | 2.34 ± 0.00 | 3.46 ± 0.04 | 6.40 ± 0.18 | 5.55 ± 0.00 | 8.60 ± 0.00 | 2.67 ± 0.00 | 5.19 ± 2.65 | 1.00 / 1.00 |
| | lora | 0.36 ± 0.01 | 0.17 ± 0.00 | 0.01 ± 0.00 | 0.12 ± 0.00 | 0.11 ± 0.00 | 0.76 ± 0.02 | 1.17 ± 0.07 | 1.94 ± 0.00 | 0.16 ± 0.00 | 0.85 ± 0.00 | 0.57 ± 0.59 | 0.00 / 0.00 |
| SVD | SVD 2 | 0.32 ± 0.01 | 0.15 ± 0.00 | 0.01 ± 0.00 | 0.12 ± 0.00 | 0.10 ± 0.00 | 0.76 ± 0.02 | 1.13 ± 0.08 | 1.94 ± 0.00 | 0.13 ± 0.00 | 0.97 ± 0.00 | 0.57 ± 0.60 | 0.88 / 0.88 |
| | SVD 4 | 0.33 ± 0.01 | 0.16 ± 0.00 | 0.01 ± 0.00 | 0.12 ± 0.00 | 0.11 ± 0.00 | 0.76 ± 0.02 | 1.14 ± 0.08 | 1.94 ± 0.00 | 0.14 ± 0.00 | 0.86 ± 0.00 | 0.56 ± 0.59 | 0.75 / 0.75 |
| | SVD 8 | 0.35 ± 0.01 | 0.17 ± 0.00 | 0.01 ± 0.00 | 0.12 ± 0.00 | 0.11 ± 0.00 | 0.77 ± 0.02 | 1.16 ± 0.07 | 1.94 ± 0.00 | 0.15 ± 0.00 | 0.84 ± 0.00 | 0.51 ± 0.58 | 0.50 / 0.50 |
| | SVD 16 | 0.36 ± 0.01 | 0.17 ± 0.00 | 0.01 ± 0.00 | 0.12 ± 0.00 | 0.11 ± 0.00 | 0.76 ± 0.02 | 1.14 ± 0.06 | 1.94 ± 0.00 | 0.16 ± 0.00 | 0.85 ± 0.00 | 0.56 ± 0.59 | 0.00 / 0.00 |
| 10 diagonal (D) | 16 D | 0.33 ± 0.01 | 0.15 ± 0.01 | 0.01 ± 0.00 | 0.12 ± 0.00 | 0.10 ± 0.00 | 0.76 ± 0.03 | 1.13 ± 0.08 | 1.95 ± 0.01 | 0.14 ± 0.00 | 1.00 ± 0.02 | 0.57 ± 0.61 | 1.00 / 0.90 |
| | 32 D | 0.33 ± 0.01 | 0.16 ± 0.00 | 0.01 ± 0.00 | 0.12 ± 0.00 | 0.10 ± 0.00 | 0.75 ± 0.02 | 1.11 ± 0.07 | 1.93 ± 0.00 | 0.14 ± 0.01 | 0.88 ± 0.00 | 0.55 ± 0.60 | 1.00 / 0.80 |
| | 64 D | 0.35 ± 0.01 | 0.17 ± 0.00 | 0.01 ± 0.00 | 0.12 ± 0.00 | 0.11 ± 0.00 | 0.75 ± 0.02 | 1.11 ± 0.07 | 1.94 ± 0.00 | 0.15 ± 0.00 | 0.84 ± 0.00 | 0.55 ± 0.59 | 1.00 / 0.60 |
| | 128 D | 0.35 ± 0.01 | 0.17 ± 0.00 | 0.01 ± 0.00 | 0.12 ± 0.00 | 0.11 ± 0.00 | 0.75 ± 0.02 | 1.11 ± 0.07 | 1.94 ± 0.00 | 0.16 ± 0.00 | 0.84 ± 0.00 | 0.56 ± 0.59 | 1.00 / 0.20 |
| | 256 D | 0.36 ± 0.01 | 0.17 ± 0.00 | 0.01 ± 0.00 | 0.12 ± 0.00 | 0.11 ± 0.00 | 0.75 ± 0.02 | 1.12 ± 0.07 | 1.94 ± 0.00 | 0.16 ± 0.00 | 0.85 ± 0.00 | 0.56 ± 0.59 | 1.00 / -0.60 |
| 10 full (F) | 16 F | 0.33 ± 0.00 | 0.15 ± 0.00 | 0.01 ± 0.00 | 0.12 ± 0.00 | 0.10 ± 0.00 | 0.76 ± 0.02 | 1.20 ± 0.02 | 1.95 ± 0.00 | 0.13 ± 0.00 | 0.97 ± 0.00 | 0.57 ± 0.61 | 1.00 / 0.90 |
| | 32 F | 0.33 ± 0.01 | 0.16 ± 0.00 | 0.01 ± 0.00 | 0.12 ± 0.00 | 0.10 ± 0.00 | 0.75 ± 0.02 | 1.11 ± 0.07 | 1.94 ± 0.00 | 0.14 ± 0.00 | 0.86 ± 0.00 | 0.55 ± 0.60 | 0.99 / 0.79 |
| | 64 F | 0.34 ± 0.01 | 0.16 ± 0.00 | 0.01 ± 0.00 | 0.12 ± 0.00 | 0.11 ± 0.00 | 0.75 ± 0.02 | 1.11 ± 0.07 | 1.94 ± 0.00 | 0.15 ± 0.00 | 0.84 ± 0.00 | 0.55 ± 0.59 | 0.97 / 0.57 |
| | 128 F | 0.35 ± 0.01 | 0.17 ± 0.00 | 0.01 ± 0.00 | 0.12 ± 0.00 | 0.11 ± 0.00 | 0.75 ± 0.02 | 1.12 ± 0.07 | 1.94 ± 0.00 | 0.16 ± 0.00 | 0.84 ± 0.00 | 0.56 ± 0.59 | 0.88 / 0.07 |
| | 256 F | 0.36 ± 0.01 | 0.17 ± 0.00 | 0.01 ± 0.00 | 0.12 ± 0.00 | 0.11 ± 0.00 | 0.75 ± 0.02 | 1.12 ± 0.07 | 1.94 ± 0.00 | 0.16 ± 0.00 | 0.85 ± 0.00 | 0.56 ± 0.59 | 0.50 / -1.10 |
| 50 diagonal (D) | 16 D | 0.61 ± 0.06 | 0.19 ± 0.02 | 0.03 ± 0.01 | 0.29 ± 0.04 | 0.36 ± 0.04 | 0.95 ± 0.05 | 1.73 ± 0.21 | 2.66 ± 0.22 | 0.32 ± 0.11 | 1.98 ± 0.01 | 0.91 ± 0.88 | 1.00 / 0.98 |
| | 32 D | 0.37 ± 0.02 | 0.16 ± 0.00 | 0.01 ± 0.00 | 0.19 ± 0.03 | 0.18 ± 0.01 | 0.85 ± 0.05 | 1.37 ± 0.14 | 2.12 ± 0.05 | 0.16 ± 0.00 | 1.65 ± 0.03 | 0.71 ± 0.73 | 1.00 / 0.96 |
| | 64 D | 0.33 ± 0.02 | 0.15 ± 0.00 | 0.01 ± 0.00 | 0.12 ± 0.00 | 0.10 ± 0.00 | 0.79 ± 0.02 | 1.12 ± 0.08 | 1.97 ± 0.01 | 0.13 ± 0.01 | 1.13 ± 0.03 | 0.59 ± 0.63 | 1.00 / 0.92 |
| | 128 D | 0.33 ± 0.01 | 0.15 ± 0.00 | 0.01 ± 0.00 | 0.12 ± 0.00 | 0.10 ± 0.00 | 0.76 ± 0.03 | 1.10 ± 0.05 | 1.93 ± 0.00 | 0.14 ± 0.00 | 0.93 ± 0.01 | 0.56 ± 0.60 | 1.00 / 0.84 |
| | 256 D | 0.34 ± 0.01 | 0.16 ± 0.00 | 0.01 ± 0.00 | 0.12 ± 0.00 | 0.10 ± 0.00 | 0.76 ± 0.03 | 1.11 ± 0.05 | 1.93 ± 0.00 | 0.15 ± 0.00 | 0.85 ± 0.00 | 0.55 ± 0.59 | 1.00 / 0.68 |
| 50 full (F) | 16 F | 0.47 ± 0.06 | 0.17 ± 0.00 | 0.02 ± 0.00 | 0.20 ± 0.02 | 0.19 ± 0.04 | 0.86 ± 0.03 | 1.71 ± 0.10 | 2.20 ± 0.04 | 0.17 ± 0.01 | 1.84 ± 0.07 | 0.78 ± 0.80 | 1.00 / 0.98 |
| | 32 F | 0.36 ± 0.01 | 0.16 ± 0.00 | 0.01 ± 0.00 | 0.14 ± 0.00 | 0.11 ± 0.00 | 0.80 ± 0.03 | 1.14 ± 0.08 | 2.00 ± 0.01 | 0.14 ± 0.00 | 1.32 ± 0.02 | 0.62 ± 0.65 | 0.99 / 0.95 |
| | 64 F | 0.33 ± 0.01 | 0.15 ± 0.00 | 0.01 ± 0.00 | 0.12 ± 0.00 | 0.10 ± 0.00 | 0.77 ± 0.03 | 1.10 ± 0.06 | 1.94 ± 0.00 | 0.13 ± 0.00 | 1.02 ± 0.00 | 0.57 ± 0.61 | 0.97 / 0.89 |
| | 128 F | 0.33 ± 0.01 | 0.16 ± 0.00 | 0.00 ± 0.00 | 0.12 ± 0.00 | 0.10 ± 0.00 | 0.76 ± 0.03 | 1.11 ± 0.05 | 1.93 ± 0.00 | 0.14 ± 0.00 | 0.87 ± 0.00 | 0.55 ± 0.60 | 0.88 / 0.72 |
| | 256 F | 0.35 ± 0.01 | 0.16 ± 0.00 | 0.01 ± 0.00 | 0.12 ± 0.00 | 0.11 ± 0.00 | 0.76 ± 0.02 | 1.11 ± 0.05 | 1.94 ± 0.00 | 0.15 ± 0.00 | 0.84 ± 0.00 | 0.55 ± 0.59 | 0.50 / 0.18 |
| 100 diagonal (D) | 16 D | 1.69 ± 0.49 | 0.26 ± 0.04 | 0.18 ± 0.07 | 0.34 ± 0.02 | 1.01 ± 0.20 | 1.45 ± 0.10 | 3.59 ± 0.25 | 3.72 ± 0.72 | 0.44 ± 0.20 | 2.37 ± 0.09 | 1.51 ± 1.32 | 1.00 / 0.99 |
| | 32 D | 0.67 ± 0.24 | 0.18 ± 0.01 | 0.06 ± 0.05 | 0.31 ± 0.06 | 0.35 ± 0.08 | 1.04 ± 0.15 | 1.97 ± 0.13 | 2.88 ± 0.70 | 0.22 ± 0.01 | 2.12 ± 0.07 | 0.98 ± 0.98 | 1.00 / 0.98 |
| | 64 D | 0.39 ± 0.06 | 0.16 ± 0.00 | 0.01 ± 0.00 | 0.18 ± 0.02 | 0.14 ± 0.01 | 0.86 ± 0.02 | 1.39 ± 0.07 | 2.18 ± 0.04 | 0.17 ± 0.00 | 1.79 ± 0.02 | 0.73 ± 0.76 | 1.00 / 0.96 |
| | 128 D | 0.32 ± 0.00 | 0.15 ± 0.00 | 0.01 ± 0.00 | 0.12 ± 0.00 | 0.10 ± 0.00 | 0.79 ± 0.02 | 1.19 ± 0.02 | 2.00 ± 0.01 | 0.14 ± 0.01 | 1.24 ± 0.04 | 0.61 ± 0.65 | 1.00 / 0.92 |
| | 256 D | 0.32 ± 0.00 | 0.15 ± 0.00 | 0.01 ± 0.00 | 0.12 ± 0.00 | 0.10 ± 0.00 | 0.77 ± 0.02 | 1.16 ± 0.00 | 1.94 ± 0.00 | 0.13 ± 0.00 | 0.96 ± 0.01 | 0.56 ± 0.61 | 1.00 / 0.84 |
| 100 full (F) | 16 F | 0.66 ± 0.07 | 0.19 ± 0.01 | 0.03 ± 0.01 | 0.25 ± 0.02 | 0.29 ± 0.02 | 0.99 ± 0.07 | 2.50 ± 0.51 | 2.63 ± 0.03 | 0.24 ± 0.02 | 2.21 ± 0.08 | 1.00 ± 1.01 | 1.00 / 0.99 |
| | 32 F | 0.40 ± 0.00 | 0.17 ± 0.00 | 0.01 ± 0.00 | 0.15 ± 0.01 | 0.13 ± 0.01 | 0.85 ± 0.02 | 1.53 ± 0.12 | 2.17 ± 0.06 | 0.15 ± 0.01 | 1.93 ± 0.04 | 0.75 ± 0.80 | 0.99 / 0.97 |
| | 64 F | 0.34 ± 0.01 | 0.15 ± 0.00 | 0.01 ± 0.00 | 0.12 ± 0.00 | 0.11 ± 0.00 | 0.79 ± 0.01 | 1.23 ± 0.07 | 1.98 ± 0.01 | 0.15 ± 0.00 | 1.26 ± 0.01 | 0.61 ± 0.65 | 0.97 / 0.93 |
| | 128 F | 0.32 ± 0.00 | 0.15 ± 0.00 | 0.01 ± 0.00 | 0.12 ± 0.00 | 0.10 ± 0.00 | 0.77 ± 0.02 | 1.16 ± 0.01 | 1.94 ± 0.00 | 0.13 ± 0.00 | 0.99 ± 0.01 | 0.57 ± 0.61 | 0.88 / 0.80 |
| | 256 F | 0.33 ± 0.00 | 0.16 ± 0.00 | 0.00 ± 0.00 | 0.12 ± 0.00 | 0.10 ± 0.00 | 0.76 ± 0.02 | 1.15 ± 0.01 | 1.93 ± 0.00 | 0.14 ± 0.00 | 0.86 ± 0.00 | 0.56 ± 0.60 | 0.50 / 0.34 |
| 100 w/clusters (C) | 16 C 5 | 0.33 | 0.15 | 0.01 | 0.14 | 0.11 | 0.76 | 1.16 | 1.97 | 0.13 | 1.12 | 0.59 | 1.00/0.95 |
| | 16 C 7 | 0.34 | 0.15 | 0.01 | 0.13 | 0.10 | 0.75 | 1.15 | 1.96 | 0.14 | 1.06 | 0.58 | 1.00/0.93 |
| 500 diagonal (D) | 16 D | 2.95 ± 0.28 | 0.73 ± 0.29 | 0.27 ± 0.09 | 0.67 ± 0.28 | 0.52 ± 0.07 | 2.06 ± 0.30 | 4.85 ± 0.31 | 3.94 ± 0.42 | 0.50 ± 0.05 | 2.50 ± 0.03 | 1.94 ± 1.59 | 1.00 / 1.00 |
| | 32 D | 2.33 ± 0.30 | 0.62 ± 0.17 | 0.24 ± 0.05 | 0.50 ± 0.16 | 0.37 ± 0.07 | 1.86 ± 0.25 | 4.73 ± 0.35 | 3.81 ± 0.59 | 0.39 ± 0.04 | 2.46 ± 0.05 | 1.77 ± 1.57 | 1.00 / 1.00 |
| | 64 D | 1.67 ± 0.18 | 0.43 ± 0.16 | 0.13 ± 0.04 | 0.29 ± 0.02 | 0.23 ± 0.02 | 1.32 ± 0.18 | 3.99 ± 0.36 | 3.41 ± 0.28 | 0.32 ± 0.05 | 2.35 ± 0.11 | 1.45 ± 1.39 | 1.00 / 0.99 |
| | 128 D | 1.12 ± 0.02 | 0.23 ± 0.00 | 0.04 ± 0.03 | 0.21 ± 0.04 | 0.22 ± 0.03 | 1.08 ± 0.06 | 3.05 ± 0.87 | 3.09 ± 0.37 | 0.26 ± 0.03 | 2.31 ± 0.04 | 1.19 ± 1.21 | 1.00 / 0.98 |
| | 256 D | 0.54 ± 0.03 | 0.18 ± 0.01 | 0.01 ± 0.00 | 0.16 ± 0.01 | 0.15 ± 0.01 | 0.92 ± 0.08 | 2.42 ± 0.14 | 2.51 ± 0.13 | 0.19 ± 0.01 | 2.09 ± 0.02 | 0.94 ± 0.99 | 1.00 / 0.97 |
| 500 full (F) | 16 F | 2.14 ± 0.06 | 0.70 ± 0.04 | 0.28 ± 0.00 | 0.27 ± 0.01 | 0.21 ± 0.00 | 1.14 ± 0.04 | 3.06 ± 0.27 | 2.71 ± 0.01 | 0.34 ± 0.01 | 2.21 ± 0.01 | 1.33 ± 1.09 | 1.00 / 1.00 |
| | 32 F | 1.17 ± 0.07 | 0.48 ± 0.03 | 0.08 ± 0.04 | 0.21 ± 0.01 | 0.17 ± 0.00 | 0.99 ± 0.04 | 2.69 ± 0.01 | 2.47 ± 0.02 | 0.25 ± 0.02 | 2.11 ± 0.04 | 1.08 ± 0.99 | 0.99 / 0.99 |
| | 64 F | 0.51 ± 0.03 | 0.21 ± 0.04 | 0.02 ± 0.00 | 0.17 ± 0.01 | 0.14 ± 0.00 | 0.88 ± 0.04 | 2.19 ± 0.14 | 2.34 ± 0.03 | 0.20 ± 0.00 | 1.97 ± 0.02 | 0.89 ± 0.91 | 0.97 / 0.96 |
| | 128 F | 0.39 ± 0.01 | 0.16 ± 0.00 | 0.01 ± 0.00 | 0.12 ± 0.00 | 0.11 ± 0.00 | 0.81 ± 0.03 | 1.42 ± 0.07 | 2.03 ± 0.01 | 0.16 ± 0.00 | 1.71 ± 0.01 | 0.71 ± 0.74 | 0.88 / 0.86 |
| | 256 F | 0.32 ± 0.01 | 0.15 ± 0.00 | 0.01 ± 0.00 | 0.12 ± 0.00 | 0.10 ± 0.00 | 0.77 ± 0.02 | 1.18 ± 0.04 | 1.96 ± 0.00 | 0.14 ± 0.01 | 1.25 ± 0.00 | 0.61 ± 0.65 | 0.50 / 0.47 |
| 500 w/clusters (C) | 16 C 7 | 0.40 | 0.18 | 0.01 | 0.15 | 0.13 | 0.90 | 2.03 | 2.21 | 0.16 | 1.50 | 0.77 | 1.00/0.98 |
| | 16 C 10 | 0.36 | 0.16 | 0.01 | 0.14 | 0.13 | 0.87 | 2.19 | 2.04 | 0.15 | 1.38 | 0.74 | 1.00/0.98 |
| | 16 C 25 | 0.32 | 0.16 | 0.01 | 0.13 | 0.10 | 0.81 | 1.28 | 1.96 | 0.12 | 1.07 | 0.60 | 1.00/0.95 |
| | 64 C 5 | 0.36 | 0.16 | 0.01 | 0.12 | 0.10 | 0.80 | 1.17 | 1.98 | 0.14 | 1.17 | 0.60 | 0.97/0.93 |
| | 64 C 7 | 0.34 | 0.15 | 0.01 | 0.12 | 0.10 | 0.79 | 1.14 | 1.96 | 0.13 | 1.08 | 0.58 | 0.97/0.91 |

Table 9: **Absolute** In-Distribution test loss for various tasks and methods

| Model Type | Method Type | task039 | task190 | task280 | task290 | task391 | task442 | task620 | task1342 | task1391 | task1598 | Average | Para. Saved |
|---|---|---|---|---|---|---|---|---|---|---|---|---|---|
| | base | 0.00 ± 0.00 | 0.00 ± 0.00 | 1.00 ± 0.00 | 0.00 ± 0.00 | 0.00 ± 0.00 | 0.00 ± 0.00 | 0.00 ± 0.00 | 0.00 ± 0.00 | 0.00 ± 0.00 | 0.00 ± 0.00 | 0.10 ± 0.30 | 1.00 / 1.00 |
| | lora | 100.00 ± 0.00 | 100.00 ± 0.00 | 100.00 ± 0.00 | 100.00 ± 0.00 | 100.00 ± 0.00 | 100.00 ± 0.00 | 100.00 ± 0.00 | 100.00 ± 0.00 | 100.00 ± 0.00 | 100.00 ± 0.00 | 100.00 ± 0.00 | 0.00 / 0.00 |
| TIES | 10 | 41.00 ± 0.00 | 53.67 ± 0.58 | 44.33 ± 4.04 | 10.33 ± 0.58 | 46.33 ± 4.04 | 1.00 ± 0.00 | 8.00 ± 0.00 | 8.00 ± 0.00 | 76.67 ± 1.15 | 1.00 ± 0.00 | 29.03 ± 25.69 | 1.00 / 1.00 |
| | 50 | 24.00 ± 0.00 | 38.67 ± 0.58 | 17.67 ± 4.62 | 2.00 ± 0.00 | 56.33 ± 0.58 | 1.00 ± 0.00 | 8.00 ± 0.00 | 8.00 ± 0.00 | 29.67 ± 2.89 | 0.00 ± 0.00 | 18.53 ± 18.07 | 1.00 / 1.00 |
| | 100 | 22.00 ± 0.00 | 38.00 ± 0.00 | 18.67 ± 4.62 | 1.00 ± 1.73 | 51.67 ± 4.62 | 1.00 ± 0.00 | 8.00 ± 0.00 | 7.33 ± 0.58 | 2.00 ± 0.00 | 0.00 ± 0.00 | 14.97 ± 17.20 | 1.00 / 1.00 |
| | 500 | 8.00 ± 0.00 | 25.00 ± 0.00 | 1.00 ± 0.00 | 0.00 ± 0.00 | 59.00 ± 0.00 | 0.00 ± 0.00 | 0.00 ± 0.00 | 3.00 ± 0.00 | 2.00 ± 0.00 | 0.00 ± 0.00 | 9.90 ± 18.12 | 1.00 / 1.00 |
| SVD | SVD 2 | 88.33 ± 0.65 | 91.91 ± 0.94 | 100.00 ± 0.00 | 97.25 ± 0.45 | 92.83 ± 0.39 | 76.50 ± 1.51 | 66.00 ± 1.41 | 58.08 ± 1.16 | 98.67 ± 0.49 | 5.83 ± 0.94 | 77.42 ± 27.69 | 0.88 / 0.88 |
| | SVD 4 | 93.00 ± 0.00 | 96.64 ± 0.50 | 100.00 ± 0.00 | 100.00 ± 0.00 | 96.75 ± 0.87 | 88.83 ± 1.53 | 90.67 ± 1.23 | 72.17 ± 0.58 | 98.67 ± 0.49 | 16.67 ± 1.78 | 85.24 ± 24.39 | 0.75 / 0.75 |
| | SVD 8 | 98.89 ± 0.60 | 98.55 ± 0.52 | 100.00 ± 0.00 | 100.00 ± 0.00 | 99.42 ± 0.51 | 93.44 ± 0.73 | 97.22 ± 0.44 | 82.78 ± 1.64 | 99.00 ± 0.00 | 60.00 ± 0.87 | 93.70 ± 11.59 | 0.50 / 0.50 |
| | SVD 16 | 100.00 ± 0.00 | 100.00 ± 0.00 | 100.00 ± 0.00 | 100.00 ± 0.00 | 100.00 ± 0.00 | 99.67 ± 0.50 | 99.50 ± 0.55 | 99.67 ± 0.50 | 100.00 ± 0.00 | 98.11 ± 0.78 | 99.69 ± 0.68 | 0.00 / 0.00 |
| 10 diagonal (D) | 16 D | 83.33 ± 1.53 | 88.33 ± 0.58 | 100.00 ± 0.00 | 97.00 ± 2.00 | 88.33 ± 1.15 | 57.00 ± 1.00 | 48.67 ± 3.21 | 50.67 ± 4.93 | 97.67 ± 1.53 | 5.33 ± 1.15 | 71.63 ± 29.53 | 1.00 / 0.90 |
| | 32 D | 93.00 ± 1.00 | 95.33 ± 0.58 | 100.00 ± 0.00 | 98.00 ± 1.00 | 93.67 ± 1.53 | 80.67 ± 2.31 | 78.67 ± 1.15 | 68.00 ± 1.73 | 98.33 ± 0.58 | 14.67 ± 2.52 | 82.03 ± 24.99 | 1.00 / 0.80 |
| | 64 D | 99.00 ± 0.00 | 97.00 ± 1.00 | 100.00 ± 0.00 | 98.00 ± 0.00 | 98.00 ± 0.00 | 90.67 ± 1.53 | 95.33 ± 1.15 | 79.67 ± 1.53 | 99.00 ± 0.00 | 55.00 ± 4.36 | 91.37 ± 13.78 | 1.00 / 0.60 |
| | 128 D | 100.00 ± 0.00 | 99.33 ± 0.58 | 100.00 ± 0.00 | 100.00 ± 0.00 | 100.00 ± 0.00 | 96.67 ± 1.53 | 98.33 ± 1.15 | 95.67 ± 2.31 | 100.00 ± 0.00 | 91.67 ± 3.51 | 98.17 ± 2.94 | 1.00 / 0.20 |
| | 256 D | 100.00 ± 0.00 | 100.00 ± 0.00 | 100.00 ± 0.00 | 100.00 ± 0.00 | 100.00 ± 0.00 | 100.00 ± 0.00 | 100.00 ± 0.00 | 99.33 ± 1.15 | 100.00 ± 0.00 | 95.00 ± 1.00 | 99.43 ± 1.57 | 1.00 / -0.60 |
| 10 full (F) | 16 F | 83.00 ± 2.00 | 93.00 ± 1.00 | 100.00 ± 0.00 | 98.33 ± 0.58 | 91.67 ± 0.58 | 64.33 ± 3.21 | 59.33 ± 1.15 | 52.67 ± 1.53 | 98.33 ± 0.58 | 6.33 ± 1.15 | 74.70 ± 28.71 | 1.00/0.90 |
| | 32 F | 91.33 ± 0.58 | 96.00 ± 1.00 | 100.00 ± 0.00 | 98.33 ± 0.58 | 94.33 ± 0.58 | 84.00 ± 2.00 | 83.00 ± 1.73 | 70.33 ± 1.53 | 99.00 ± 1.00 | 22.00 ± 2.65 | 83.83 ± 22.82 | 0.99/0.79 |
| | 64 F | 99.00 ± 0.00 | 97.33 ± 0.58 | 100.00 ± 0.00 | 100.00 ± 0.00 | 99.33 ± 1.15 | 91.33 ± 1.53 | 96.33 ± 0.58 | 81.67 ± 2.31 | 99.00 ± 0.00 | 58.33 ± 1.53 | 92.23 ± 12.76 | 0.97/0.57 |
| | 128 F | 99.67 ± 0.58 | 99.33 ± 0.58 | 100.00 ± 0.00 | 100.00 ± 0.00 | 100.00 ± 0.00 | 97.67 ± 1.15 | 100.00 ± 0.00 | 95.67 ± 1.15 | 100.00 ± 0.00 | 91.00 ± 1.00 | 98.33 ± 2.89 | 0.88/0.07 |
| | 256 F | 100.00 ± 0.00 | 100.00 ± 0.00 | 100.00 ± 0.00 | 100.00 ± 0.00 | 100.00 ± 0.00 | 99.67 ± 0.58 | 99.67 ± 0.58 | 99.67 ± 0.58 | 100.00 ± 0.00 | 98.00 ± 1.00 | 99.70 ± 0.70 | 0.50/-1.10 |
| 50 diagonal (D) | 16 D | 52.67 ± 4.51 | 86.67 ± 3.06 | 100.00 ± 0.00 | 85.00 ± 3.46 | 65.33 ± 3.79 | 25.33 ± 5.03 | 10.00 ± 1.00 | 10.67 ± 10.26 | 81.00 ± 6.56 | 0.00 ± 0.00 | 51.67 ± 36.18 | 1.00 / 0.98 |
| | 32 D | 69.67 ± 3.21 | 88.67 ± 1.53 | 100.00 ± 0.00 | 95.00 ± 2.00 | 80.00 ± 3.00 | 36.67 ± 5.13 | 17.00 ± 2.65 | 26.33 ± 5.03 | 95.00 ± 2.00 | 0.00 ± 0.00 | 60.83 ± 36.02 | 1.00 / 0.96 |
| | 64 D | 79.67 ± 2.52 | 91.00 ± 1.00 | 100.00 ± 0.00 | 97.67 ± 0.58 | 88.00 ± 1.00 | 52.00 ± 1.00 | 36.67 ± 5.69 | 41.33 ± 1.15 | 96.00 ± 1.00 | 0.33 ± 0.58 | 68.27 ± 32.66 | 1.00 / 0.92 |
| | 128 D | 90.00 ± 1.00 | 91.33 ± 0.58 | 100.00 ± 0.00 | 98.33 ± 0.58 | 90.67 ± 2.08 | 73.67 ± 2.08 | 63.67 ± 1.53 | 56.33 ± 0.58 | 98.00 ± 0.00 | 7.33 ± 1.15 | 76.93 ± 27.79 | 1.00 / 0.84 |
| | 256 D | 94.67 ± 0.58 | 96.33 ± 0.58 | 100.00 ± 0.00 | 99.67 ± 0.58 | 96.33 ± 1.15 | 87.33 ± 0.58 | 87.00 ± 2.65 | 71.67 ± 1.53 | 99.67 ± 0.58 | 31.67 ± 1.15 | 86.43 ± 20.41 | 1.00 / 0.68 |
| 50 full (F) | 16 F | 61.67 ± 3.06 | 89.67 ± 1.15 | 99.67 ± 0.58 | 90.67 ± 2.52 | 78.33 ± 3.51 | 34.00 ± 1.00 | 7.00 ± 3.46 | 25.00 ± 6.24 | 90.00 ± 1.00 | 0.00 ± 0.00 | 57.60 ± 36.59 | 1.00/0.98 |
| | 32 F | 71.00 ± 1.00 | 89.00 ± 1.73 | 100.00 ± 0.00 | 98.00 ± 0.00 | 85.00 ± 1.00 | 47.00 ± 1.73 | 29.00 ± 3.00 | 35.00 ± 2.00 | 98.00 ± 0.00 | 0.00 ± 0.00 | 65.20 ± 34.00 | 0.99/0.95 |
| | 64 F | 81.67 ± 0.58 | 93.67 ± 1.15 | 100.00 ± 0.00 | 98.33 ± 0.58 | 90.67 ± 2.08 | 61.67 ± 1.53 | 54.33 ± 1.53 | 51.33 ± 1.15 | 98.33 ± 0.58 | 3.33 ± 0.58 | 73.33 ± 29.89 | 0.97/0.89 |
| | 128 F | 91.00 ± 1.00 | 94.33 ± 0.58 | 100.00 ± 0.00 | 99.00 ± 0.00 | 93.33 ± 1.53 | 75.00 ± 1.73 | 67.67 ± 2.08 | 67.67 ± 2.08 | 98.67 ± 0.58 | 16.67 ± 0.58 | 81.73 ± 24.46 | 0.88/0.72 |
| | 256 F | 97.00 ± 0.00 | 98.00 ± 0.00 | 100.00 ± 0.00 | 100.00 ± 0.00 | 99.67 ± 0.58 | 92.00 ± 0.00 | 94.33 ± 1.15 | 79.67 ± 1.53 | 99.00 ± 0.00 | 57.33 ± 2.52 | 91.70 ± 13.11 | 0.50/0.18 |
| 100 diagonal (D) | 16 D | 33.00 ± 8.19 | 79.33 ± 5.69 | 89.33 ± 5.51 | 80.00 ± 3.61 | 35.33 ± 6.03 | 4.00 ± 1.73 | 3.00 ± 1.00 | 0.00 ± 0.00 | 71.33 ± 4.51 | 0.00 ± 0.00 | 39.53 ± 36.15 | 1.00 / 0.99 |
| | 32 D | 51.00 ± 7.81 | 90.00 ± 1.00 | 100.00 ± 0.00 | 88.00 ± 7.00 | 58.67 ± 11.68 | 17.67 ± 10.26 | 7.67 ± 2.89 | 9.33 ± 12.86 | 86.33 ± 2.52 | 0.00 ± 0.00 | 50.87 ± 38.43 | 1.00 / 0.98 |
| | 64 D | 68.00 ± 2.65 | 87.33 ± 1.53 | 100.00 ± 0.00 | 94.33 ± 4.04 | 80.33 ± 2.08 | 38.00 ± 3.00 | 19.67 ± 4.51 | 28.33 ± 1.53 | 92.67 ± 1.15 | 0.33 ± 0.58 | 60.90 ± 34.91 | 1.00 / 0.96 |
| | 128 D | 82.00 ± 2.00 | 90.00 ± 2.00 | 100.00 ± 0.00 | 97.33 ± 0.58 | 85.33 ± 0.58 | 55.33 ± 2.08 | 34.33 ± 3.79 | 36.67 ± 2.52 | 94.67 ± 0.58 | 0.00 ± 0.00 | 67.57 ± 32.95 | 1.00 / 0.92 |
| | 256 D | 90.00 ± 1.00 | 90.00 ± 1.00 | 100.00 ± 0.00 | 97.67 ± 0.58 | 91.67 ± 0.58 | 71.67 ± 5.13 | 59.67 ± 1.15 | 36.67 ± 2.52 | 99.00 ± 0.00 | 4.00 ± 1.00 | 76.33 ± 28.88 | 1.00 / 0.84 |
| 100 full (F) | 16 F | 49.00 ± 2.00 | 89.67 ± 3.21 | 97.00 ± 3.00 | 84.33 ± 3.06 | 65.33 ± 2.52 | 20.67 ± 8.33 | 6.33 ± 2.08 | 8.33 ± 4.73 | 81.33 ± 2.08 | 0.00 ± 0.00 | 50.20 ± 37.06 | 1.00/0.99 |
| | 32 F | 65.00 ± 3.46 | 90.33 ± 1.53 | 100.00 ± 0.00 | 96.33 ± 1.53 | 80.00 ± 2.65 | 41.33 ± 3.21 | 16.00 ± 0.00 | 29.33 ± 2.08 | 92.00 ± 2.65 | 0.00 ± 0.00 | 61.03 ± 34.43 | 0.99/0.97 |
| | 64 F | 72.33 ± 0.58 | 89.67 ± 1.53 | 100.00 ± 0.00 | 97.67 ± 0.58 | 86.00 ± 1.00 | 53.00 ± 1.00 | 35.33 ± 1.53 | 38.00 ± 1.73 | 94.67 ± 0.58 | 0.00 ± 0.00 | 66.67 ± 32.54 | 0.97/0.93 |
| | 128 F | 84.33 ± 1.53 | 92.33 ± 1.53 | 100.00 ± 0.00 | 98.00 ± 0.00 | 91.33 ± 0.58 | 68.67 ± 0.58 | 56.00 ± 1.00 | 57.67 ± 1.15 | 99.00 ± 0.00 | 5.33 ± 0.58 | 75.27 ± 28.67 | 0.88/0.80 |
| | 256 F | 91.67 ± 1.15 | 96.67 ± 0.58 | 100.00 ± 0.00 | 100.00 ± 0.00 | 94.33 ± 0.58 | 84.67 ± 6.35 | 78.00 ± 0.00 | 69.67 ± 0.58 | 99.00 ± 0.00 | 22.00 ± 1.00 | 83.60 ± 23.08 | 0.50/0.34 |
| 100 w/clusters (C) | 16 C 5 | 74.00 | 92.00 | 100.00 | 95.00 | 86.00 | 54.00 | 41.00 | 47.00 | 99.00 | 0.00 | 68.80 | 1.00/0.95 |
| | 16 C 7 | 78.00 | 92.00 | 100.00 | 98.00 | 93.00 | 58.00 | 51.00 | 48.00 | 98.00 | 2.00 | 71.80 | 1.00/0.93 |
| 500 diagonal (D) | 16 D | 8.00 ± 3.61 | 51.50 ± 3.54 | 79.67 ± 4.93 | 28.00 ± 42.44 | 56.67 ± 2.89 | 0.67 ± 1.15 | 0.33 ± 0.58 | 0.00 ± 0.00 | 71.67 ± 6.03 | 0.00 ± 0.00 | 28.90 ± 33.54 | 1.00 / 1.00 |
| | 32 D | 14.33 ± 11.02 | 52.50 ± 9.19 | 80.67 ± 2.08 | 43.00 ± 31.48 | 60.67 ± 0.58 | 1.67 ± 1.15 | 0.00 ± 0.00 | 0.00 ± 0.00 | 74.33 ± 4.04 | 0.00 ± 0.00 | 32.10 ± 33.43 | 1.00 / 1.00 |
| | 64 D | 25.67 ± 1.15 | 62.50 ± 12.02 | 87.33 ± 4.04 | 78.33 ± 1.15 | 65.33 ± 3.06 | 5.33 ± 3.51 | 3.67 ± 1.15 | 1.00 ± 1.73 | 76.67 ± 2.31 | 0.00 ± 0.00 | 39.83 ± 35.80 | 1.00 / 0.99 |
| | 128 D | 38.33 ± 3.21 | 85.50 ± 2.12 | 96.00 ± 3.00 | 81.33 ± 2.31 | 65.67 ± 1.15 | 11.67 ± 4.73 | 5.33 ± 2.08 | 2.00 ± 1.00 | 80.00 ± 5.00 | 0.00 ± 0.00 | 45.24 ± 37.78 | 1.00 / 0.98 |
| | 256 D | 53.33 ± 0.58 | 91.00 ± 2.83 | 100.00 ± 0.00 | 89.00 ± 2.65 | 76.00 ± 2.00 | 20.67 ± 6.81 | 6.00 ± 1.73 | 12.00 ± 8.00 | 86.33 ± 2.31 | 0.00 ± 0.00 | 52.14 ± 38.64 | 1.00 / 0.97 |
| 500 full (F) | 16 F | 8.33 ± 2.08 | 41.00 ± 5.66 | 76.67 ± 0.58 | 78.00 ± 0.00 | 72.67 ± 0.58 | 6.00 ± 0.00 | 5.67 ± 0.58 | 0.00 ± 0.00 | 78.00 ± 1.00 | 0.00 ± 0.00 | 36.48 ± 35.46 | 1.00/1.00 |
| | 32 F | 33.67 ± 4.16 | 51.00 ± 1.41 | 92.67 ± 1.53 | 77.00 ± 1.73 | 75.00 ± 2.00 | 14.33 ± 1.53 | 8.00 ± 0.00 | 0.00 ± 0.00 | 80.67 ± 1.53 | 0.00 ± 0.00 | 42.97 ± 35.80 | 0.99/0.99 |
| | 64 F | 56.00 ± 2.65 | 88.50 ± 0.71 | 94.33 ± 0.58 | 89.33 ± 2.89 | 74.33 ± 1.15 | 36.33 ± 1.15 | 9.00 ± 1.00 | 2.67 ± 1.15 | 84.00 ± 1.00 | 0.00 ± 0.00 | 52.03 ± 36.92 | 0.97/0.96 |
| | 128 F | 69.33 ± 0.58 | 88.50 ± 0.71 | 99.00 ± 0.00 | 96.33 ± 1.53 | 80.33 ± 1.15 | 45.00 ± 2.00 | 16.33 ± 0.58 | 31.00 ± 1.73 | 92.00 ± 0.00 | 0.00 ± 0.00 | 60.86 ± 35.07 | 0.88/0.86 |
| | 256 F | 79.67 ± 0.58 | 89.50 ± 0.71 | 100.00 ± 0.00 | 97.67 ± 0.58 | 87.33 ± 0.58 | 57.00 ± 1.00 | 35.00 ± 1.00 | 42.00 ± 1.00 | 95.00 ± 1.00 | 0.00 ± 0.00 | 67.59 ± 32.67 | 0.50/0.47 |
| 500 w/clusters (C) | 16 C 7 | 63.00 | 90.00 | 99.00 | 96.00 | 78.00 | 31.00 | 9.00 | 15.00 | 89.00 | 1.00 | 57.10 | 1.00/0.98 |
| | 16 C 10 | 69.00 | 93.00 | 100.00 | 98.00 | 81.00 | 34.00 | 8.00 | 33.00 | 95.00 | 1.00 | 61.20 | 1.00/0.98 |
| | 16 C 25 | 79.00 | 90.00 | 100.00 | 97.00 | 88.00 | 53.00 | 38.00 | 48.00 | 98.00 | 0.00 | 69.10 | 1.00/0.95 |
| | 64 C 5 | 77.00 | 88.00 | 100.00 | 98.00 | 89.00 | 56.00 | 39.00 | 42.00 | 99.00 | 0.00 | 68.80 | 0.97/0.93 |
| | 64 C 7 | 76.00 | 90.00 | 100.00 | 97.00 | 89.00 | 60.00 | 48.00 | 49.00 | 99.00 | 3.00 | 71.10 | 0.97/0.91 |

Table 10: **Absolute** In-Distribution agreement for various tasks and methods

| Model Type | Method Type | Tasks | | | | | | | | | | Average |
|---|---|---|---|---|---|---|---|---|---|---|---|---|
| | | task039 | task190 | task280 | task290 | task391 | task442 | task620 | task1342 | task1391 | task1598 | |
| SVD | SVD 2 | 0.29 ± 0.00 | 0.43 ± 0.00 | 0.31 ± 0.00 | 0.40 ± 0.00 | 0.38 ± 0.00 | 0.31 ± 0.00 | 0.37 ± 0.00 | 0.31 ± 0.00 | 0.42 ± 0.00 | 0.30 ± 0.00 | 0.35 ± 0.05 |
| | SVD 4 | 0.16 ± 0.00 | 0.24 ± 0.00 | 0.16 ± 0.00 | 0.25 ± 0.00 | 0.23 ± 0.00 | 0.17 ± 0.00 | 0.22 ± 0.00 | 0.16 ± 0.00 | 0.25 ± 0.00 | 0.16 ± 0.00 | 0.20 ± 0.04 |
| | SVD 8 | 0.06 ± 0.00 | 0.09 ± 0.00 | 0.06 ± 0.00 | 0.11 ± 0.00 | 0.10 ± 0.00 | 0.07 ± 0.00 | 0.09 ± 0.00 | 0.06 ± 0.00 | 0.11 ± 0.00 | 0.06 ± 0.00 | 0.08 ± 0.02 |
| 10 diagonal (D) | 16 D | 0.37 ± 0.02 | 0.51 ± 0.02 | 0.36 ± 0.01 | 0.57 ± 0.02 | 0.55 ± 0.00 | 0.39 ± 0.02 | 0.49 ± 0.01 | 0.36 ± 0.02 | 0.53 ± 0.03 | 0.39 ± 0.01 | 0.45 ± 0.08 |
| | 32 D | 0.21 ± 0.01 | 0.28 ± 0.00 | 0.20 ± 0.01 | 0.35 ± 0.00 | 0.33 ± 0.01 | 0.22 ± 0.01 | 0.31 ± 0.01 | 0.20 ± 0.01 | 0.32 ± 0.01 | 0.22 ± 0.00 | 0.26 ± 0.06 |
| | 64 D | 0.10 ± 0.00 | 0.11 ± 0.01 | 0.09 ± 0.00 | 0.18 ± 0.01 | 0.18 ± 0.00 | 0.10 ± 0.00 | 0.15 ± 0.01 | 0.09 ± 0.00 | 0.14 ± 0.00 | 0.09 ± 0.00 | 0.12 ± 0.04 |
| | 128 D | 0.02 ± 0.00 | 0.01 ± 0.00 | 0.02 ± 0.00 | 0.03 ± 0.00 | 0.04 ± 0.00 | 0.02 ± 0.00 | 0.03 ± 0.00 | 0.02 ± 0.00 | 0.02 ± 0.00 | 0.02 ± 0.00 | 0.03 ± 0.01 |
| | 256 D | 0.00 ± 0.00 | 0.00 ± 0.00 | 0.00 ± 0.00 | 0.00 ± 0.00 | 0.00 ± 0.00 | 0.00 ± 0.00 | 0.00 ± 0.00 | 0.00 ± 0.00 | 0.00 ± 0.00 | 0.00 ± 0.00 | 0.00 ± 0.00 |
| 10 full (F) | 16 F | 0.35 ± 0.00 | 0.46 ± 0.00 | 0.34 ± 0.00 | 0.51 ± 0.00 | 0.47 ± 0.01 | 0.36 ± 0.00 | 0.45 ± 0.01 | 0.35 ± 0.01 | 0.49 ± 0.00 | 0.35 ± 0.01 | 0.41 ± 0.06 |
| | 32 F | 0.20 ± 0.00 | 0.24 ± 0.00 | 0.20 ± 0.00 | 0.30 ± 0.00 | 0.29 ± 0.00 | 0.22 ± 0.00 | 0.27 ± 0.00 | 0.20 ± 0.00 | 0.27 ± 0.00 | 0.21 ± 0.00 | 0.24 ± 0.04 |
| | 64 F | 0.10 ± 0.00 | 0.10 ± 0.00 | 0.09 ± 0.00 | 0.13 ± 0.00 | 0.13 ± 0.00 | 0.10 ± 0.00 | 0.12 ± 0.00 | 0.09 ± 0.00 | 0.12 ± 0.00 | 0.10 ± 0.00 | 0.11 ± 0.02 |
| | 128 F | 0.02 ± 0.00 | 0.02 ± 0.00 | 0.02 ± 0.00 | 0.01 ± 0.00 | 0.02 ± 0.00 | 0.02 ± 0.00 | 0.02 ± 0.00 | 0.02 ± 0.00 | 0.01 ± 0.00 | 0.02 ± 0.00 | 0.02 ± 0.00 |
| | 256 F | 0.00 ± 0.00 | 0.00 ± 0.00 | 0.00 ± 0.00 | 0.00 ± 0.00 | 0.00 ± 0.00 | 0.00 ± 0.00 | 0.00 ± 0.00 | 0.00 ± 0.00 | 0.00 ± 0.00 | 0.00 ± 0.00 | 0.00 ± 0.00 |
| 50 diagonal (D) | 16 D | 0.66 ± 0.01 | 0.69 ± 0.01 | 0.88 ± 0.01 | 0.76 ± 0.03 | 0.95 ± 0.02 | 0.91 ± 0.01 | 0.83 ± 0.02 | 0.88 ± 0.03 | 0.72 ± 0.02 | 0.88 ± 0.02 | 0.82 ± 0.10 |
| | 32 D | 0.50 ± 0.01 | 0.52 ± 0.02 | 0.73 ± 0.01 | 0.58 ± 0.03 | 0.88 ± 0.03 | 0.79 ± 0.03 | 0.72 ± 0.01 | 0.75 ± 0.01 | 0.57 ± 0.02 | 0.75 ± 0.01 | 0.68 ± 0.12 |
| | 64 D | 0.34 ± 0.01 | 0.37 ± 0.01 | 0.52 ± 0.00 | 0.38 ± 0.01 | 0.71 ± 0.02 | 0.58 ± 0.01 | 0.54 ± 0.00 | 0.56 ± 0.00 | 0.44 ± 0.01 | 0.58 ± 0.01 | 0.50 ± 0.11 |
| | 128 D | 0.21 ± 0.01 | 0.22 ± 0.01 | 0.31 ± 0.00 | 0.22 ± 0.00 | 0.51 ± 0.01 | 0.42 ± 0.01 | 0.38 ± 0.00 | 0.39 ± 0.00 | 0.27 ± 0.00 | 0.40 ± 0.00 | 0.33 ± 0.10 |
| | 256 D | 0.10 ± 0.00 | 0.12 ± 0.00 | 0.16 ± 0.00 | 0.10 ± 0.00 | 0.29 ± 0.01 | 0.21 ± 0.00 | 0.19 ± 0.00 | 0.23 ± 0.01 | 0.15 ± 0.00 | 0.20 ± 0.00 | 0.18 ± 0.06 |
| 50 full (F) | 16 F | 0.57 ± 0.01 | 0.60 ± 0.01 | 0.86 ± 0.01 | 0.71 ± 0.02 | 0.95 ± 0.01 | 0.88 ± 0.01 | 0.81 ± 0.00 | 0.83 ± 0.01 | 0.67 ± 0.01 | 0.86 ± 0.01 | 0.78 ± 0.12 |
| | 32 F | 0.47 ± 0.01 | 0.48 ± 0.01 | 0.71 ± 0.00 | 0.55 ± 0.01 | 0.78 ± 0.01 | 0.69 ± 0.01 | 0.69 ± 0.00 | 0.65 ± 0.01 | 0.53 ± 0.01 | 0.71 ± 0.00 | 0.63 ± 0.11 |
| | 64 F | 0.33 ± 0.00 | 0.35 ± 0.00 | 0.45 ± 0.00 | 0.36 ± 0.00 | 0.56 ± 0.00 | 0.50 ± 0.01 | 0.47 ± 0.00 | 0.49 ± 0.00 | 0.39 ± 0.01 | 0.49 ± 0.00 | 0.44 ± 0.08 |
| | 128 F | 0.19 ± 0.00 | 0.21 ± 0.00 | 0.25 ± 0.00 | 0.19 ± 0.00 | 0.35 ± 0.00 | 0.30 ± 0.00 | 0.28 ± 0.00 | 0.31 ± 0.00 | 0.24 ± 0.00 | 0.30 ± 0.00 | 0.26 ± 0.05 |
| | 256 F | 0.09 ± 0.00 | 0.10 ± 0.00 | 0.10 ± 0.00 | 0.08 ± 0.00 | 0.16 ± 0.00 | 0.13 ± 0.00 | 0.12 ± 0.00 | 0.15 ± 0.00 | 0.11 ± 0.00 | 0.13 ± 0.00 | 0.12 ± 0.02 |
| 100 diagonal (D) | 16 D | 0.90 ± 0.01 | 0.85 ± 0.01 | 0.87 ± 0.03 | 0.88 ± 0.02 | 0.68 ± 0.02 | 0.91 ± 0.01 | 0.97 ± 0.01 | 0.98 ± 0.01 | 0.96 ± 0.01 | 1.00 ± 0.00 | 0.90 ± 0.09 |
| | 32 D | 0.83 ± 0.02 | 0.77 ± 0.00 | 0.77 ± 0.01 | 0.78 ± 0.00 | 0.55 ± 0.02 | 0.79 ± 0.01 | 0.94 ± 0.02 | 0.94 ± 0.03 | 0.87 ± 0.00 | 0.98 ± 0.01 | 0.82 ± 0.12 |
| | 64 D | 0.67 ± 0.01 | 0.63 ± 0.00 | 0.59 ± 0.02 | 0.63 ± 0.01 | 0.40 ± 0.00 | 0.62 ± 0.01 | 0.86 ± 0.02 | 0.82 ± 0.02 | 0.71 ± 0.03 | 0.93 ± 0.00 | 0.68 ± 0.15 |
| | 128 D | 0.49 ± 0.01 | 0.47 ± 0.00 | 0.42 ± 0.01 | 0.45 ± 0.00 | 0.27 ± 0.02 | 0.44 ± 0.01 | 0.73 ± 0.01 | 0.69 ± 0.02 | 0.59 ± 0.02 | 0.80 ± 0.02 | 0.53 ± 0.16 |
| | 256 D | 0.32 ± 0.00 | 0.31 ± 0.00 | 0.26 ± 0.01 | 0.30 ± 0.00 | 0.15 ± 0.00 | 0.28 ± 0.00 | 0.51 ± 0.02 | 0.51 ± 0.02 | 0.40 ± 0.01 | 0.61 ± 0.01 | 0.36 ± 0.14 |
| 100 full (F) | 16 F | 0.88 ± 0.00 | 0.82 ± 0.00 | 0.84 ± 0.01 | 0.86 ± 0.00 | 0.67 ± 0.01 | 0.88 ± 0.01 | 0.99 ± 0.00 | 0.96 ± 0.01 | 0.91 ± 0.01 | 1.00 ± 0.00 | 0.88 ± 0.09 |
| | 32 F | 0.78 ± 0.00 | 0.72 ± 0.00 | 0.73 ± 0.00 | 0.74 ± 0.00 | 0.52 ± 0.00 | 0.74 ± 0.01 | 0.94 ± 0.01 | 0.89 ± 0.00 | 0.77 ± 0.02 | 0.99 ± 0.00 | 0.78 ± 0.13 |
| | 64 F | 0.60 ± 0.00 | 0.57 ± 0.00 | 0.57 ± 0.00 | 0.57 ± 0.00 | 0.39 ± 0.00 | 0.56 ± 0.00 | 0.76 ± 0.00 | 0.73 ± 0.00 | 0.60 ± 0.00 | 0.83 ± 0.01 | 0.62 ± 0.12 |
| | 128 F | 0.40 ± 0.00 | 0.38 ± 0.00 | 0.35 ± 0.00 | 0.37 ± 0.00 | 0.25 ± 0.00 | 0.37 ± 0.00 | 0.52 ± 0.00 | 0.54 ± 0.00 | 0.45 ± 0.00 | 0.60 ± 0.00 | 0.42 ± 0.10 |
| | 256 F | 0.21 ± 0.00 | 0.20 ± 0.00 | 0.18 ± 0.00 | 0.19 ± 0.00 | 0.13 ± 0.00 | 0.19 ± 0.00 | 0.30 ± 0.00 | 0.34 ± 0.00 | 0.26 ± 0.00 | 0.38 ± 0.00 | 0.24 ± 0.08 |
| 100 w/clusters (C) | 16 C 5 | 0.46 | 0.46 | 0.44 | 0.47 | 0.62 | 0.64 | 0.61 | 0.63 | 0.45 | 0.60 | 0.54 |
| | 16 C 7 | 0.42 | 0.43 | 0.40 | 0.42 | 0.50 | 0.55 | 0.52 | 0.55 | 0.42 | 0.55 | 0.48 |
| 500 diagonal (D) | 16 D | 0.97 ± 0.00 | 0.73 ± 0.00 | 0.96 ± 0.00 | 1.00 ± 0.00 | 0.99 ± 0.01 | 0.96 ± 0.01 | 0.90 ± 0.00 | 0.92 ± 0.00 | 1.00 ± 0.00 | 1.00 ± 0.00 | 0.94 ± 0.08 |
| | 32 D | 0.96 ± 0.00 | 0.70 ± 0.00 | 0.92 ± 0.01 | 0.98 ± 0.01 | 0.96 ± 0.01 | 0.93 ± 0.01 | 0.86 ± 0.00 | 0.89 ± 0.00 | 1.00 ± 0.00 | 1.00 ± 0.00 | 0.92 ± 0.09 |
| | 64 D | 0.90 ± 0.01 | 0.65 ± 0.00 | 0.86 ± 0.01 | 0.96 ± 0.02 | 0.90 ± 0.01 | 0.87 ± 0.01 | 0.81 ± 0.00 | 0.83 ± 0.01 | 0.99 ± 0.01 | 1.00 ± 0.00 | 0.88 ± 0.10 |
| | 128 D | 0.82 ± 0.01 | 0.60 ± 0.00 | 0.76 ± 0.00 | 0.90 ± 0.02 | 0.83 ± 0.01 | 0.78 ± 0.02 | 0.74 ± 0.00 | 0.74 ± 0.01 | 0.97 ± 0.01 | 1.00 ± 0.00 | 0.81 ± 0.12 |
| | 256 D | 0.59 ± 0.02 | 0.51 ± 0.00 | 0.56 ± 0.01 | 0.81 ± 0.02 | 0.70 ± 0.02 | 0.55 ± 0.01 | 0.57 ± 0.01 | 0.54 ± 0.01 | 0.91 ± 0.01 | 1.00 ± 0.01 | 0.67 ± 0.17 |
| 500 full (F) | 16 F | 0.94 ± 0.00 | 0.67 ± 0.00 | 0.88 ± 0.00 | 1.00 ± 0.00 | 0.98 ± 0.00 | 0.90 ± 0.00 | 0.82 ± 0.00 | 0.83 ± 0.00 | 1.00 ± 0.00 | 1.00 ± 0.00 | 0.90 ± 0.10 |
| | 32 F | 0.88 ± 0.00 | 0.61 ± 0.00 | 0.81 ± 0.00 | 0.97 ± 0.01 | 0.94 ± 0.00 | 0.84 ± 0.00 | 0.75 ± 0.00 | 0.77 ± 0.00 | 0.99 ± 0.00 | 0.99 ± 0.00 | 0.86 ± 0.12 |
| | 64 F | 0.80 ± 0.00 | 0.55 ± 0.00 | 0.72 ± 0.00 | 0.86 ± 0.00 | 0.82 ± 0.01 | 0.76 ± 0.00 | 0.67 ± 0.00 | 0.70 ± 0.00 | 0.94 ± 0.00 | 0.99 ± 0.00 | 0.78 ± 0.13 |
| | 128 F | 0.64 ± 0.00 | 0.46 ± 0.00 | 0.60 ± 0.00 | 0.74 ± 0.00 | 0.65 ± 0.00 | 0.63 ± 0.00 | 0.56 ± 0.00 | 0.58 ± 0.00 | 0.85 ± 0.00 | 0.96 ± 0.00 | 0.67 ± 0.14 |
| | 256 F | 0.43 ± 0.00 | 0.35 ± 0.00 | 0.44 ± 0.00 | 0.55 ± 0.00 | 0.49 ± 0.00 | 0.45 ± 0.00 | 0.40 ± 0.00 | 0.42 ± 0.00 | 0.67 ± 0.00 | 0.84 ± 0.00 | 0.50 ± 0.14 |
| 500 w/clusters (C) | 16 C 7 | 0.68 | 0.70 | 0.64 | 0.72 | 0.85 | 0.90 | 0.93 | 0.92 | 0.71 | 0.83 | 0.79 |
| | 16 C 10 | 0.61 | 0.65 | 0.61 | 0.66 | 0.84 | 0.86 | 0.88 | 0.84 | 0.62 | 0.76 | 0.73 |
| | 16 C 25 | 0.42 | 0.41 | 0.42 | 0.44 | 0.57 | 0.64 | 0.63 | 0.62 | 0.40 | 0.58 | 0.51 |
| | 64 C 5 | 0.49 | 0.49 | 0.45 | 0.51 | 0.64 | 0.66 | 0.62 | 0.67 | 0.50 | 0.65 | 0.57 |
| | 64 C 7 | 0.45 | 0.45 | 0.41 | 0.45 | 0.56 | 0.58 | 0.55 | 0.59 | 0.44 | 0.57 | 0.51 |

Table 11: **Reconstruction error** In-Distribution for various tasks and methods

| Model Type | Method Type | Tasks | | | | | | | | | | Average |
|---|---|---|---|---|---|---|---|---|---|---|---|---|
| | | task039 | task190 | task280 | task290 | task391 | task442 | task620 | task1342 | task1391 | task1598 | |
| 10 full (F) | 16 F | 0.46 ± 0.01 | 0.63 ± 0.00 | 0.50 ± 0.00 | 0.55 ± 0.01 | 0.50 ± 0.00 | 0.49 ± 0.01 | 0.50 ± 0.01 | 0.50 ± 0.01 | 0.61 ± 0.01 | 0.47 ± 0.01 | 0.52 ± 0.06 |
| | 32 F | 0.30 ± 0.01 | 0.37 ± 0.00 | 0.31 ± 0.00 | 0.35 ± 0.00 | 0.34 ± 0.00 | 0.31 ± 0.00 | 0.33 ± 0.00 | 0.31 ± 0.00 | 0.38 ± 0.00 | 0.30 ± 0.00 | 0.33 ± 0.03 |
| | 64 F | 0.15 ± 0.00 | 0.15 ± 0.00 | 0.16 ± 0.00 | 0.17 ± 0.00 | 0.17 ± 0.00 | 0.16 ± 0.00 | 0.16 ± 0.00 | 0.16 ± 0.00 | 0.17 ± 0.00 | 0.15 ± 0.00 | 0.16 ± 0.01 |
| 50 full (F) | 16 F | 0.80 ± 0.02 | 0.82 ± 0.01 | 0.85 ± 0.01 | 0.90 ± 0.02 | 0.78 ± 0.01 | 0.95 ± 0.01 | 0.76 ± 0.01 | 0.75 ± 0.00 | 0.79 ± 0.01 | 0.82 ± 0.01 | 0.82 ± 0.06 |
| | 32 F | 0.65 ± 0.01 | 0.67 ± 0.01 | 0.72 ± 0.01 | 0.76 ± 0.02 | 0.65 ± 0.01 | 0.82 ± 0.02 | 0.66 ± 0.01 | 0.65 ± 0.01 | 0.67 ± 0.02 | 0.69 ± 0.01 | 0.69 ± 0.06 |
| | 64 F | 0.50 ± 0.01 | 0.52 ± 0.00 | 0.52 ± 0.00 | 0.55 ± 0.01 | 0.52 ± 0.01 | 0.62 ± 0.00 | 0.54 ± 0.01 | 0.51 ± 0.00 | 0.54 ± 0.01 | 0.57 ± 0.00 | 0.54 ± 0.03 |
| 100 full (F) | 16 F | 0.93 ± 0.02 | 0.90 ± 0.02 | 0.93 ± 0.01 | 0.91 ± 0.02 | 0.88 ± 0.03 | 0.98 ± 0.01 | 0.96 ± 0.01 | 0.78 ± 0.00 | 0.82 ± 0.00 | 0.93 ± 0.02 | 0.90 ± 0.06 |
| | 32 F | 0.87 ± 0.01 | 0.81 ± 0.01 | 0.85 ± 0.02 | 0.80 ± 0.01 | 0.79 ± 0.02 | 0.91 ± 0.00 | 0.90 ± 0.01 | 0.74 ± 0.01 | 0.70 ± 0.02 | 0.85 ± 0.02 | 0.82 ± 0.07 |
| | 64 F | 0.65 ± 0.04 | 0.69 ± 0.01 | 0.71 ± 0.01 | 0.67 ± 0.01 | 0.64 ± 0.01 | 0.76 ± 0.01 | 0.77 ± 0.01 | 0.67 ± 0.00 | 0.61 ± 0.00 | 0.75 ± 0.06 | 0.69 ± 0.06 |
| 500 full (F) | 16 F | 0.98 ± 0.04 | 0.98 ± 0.01 | 0.99 ± 0.01 | 1.00 ± 0.00 | 0.99 ± 0.00 | 0.96 ± 0.05 | 0.93 ± 0.10 | 0.94 ± 0.09 | 1.00 ± 0.00 | 0.99 ± 0.00 | 0.98 ± 0.05 |
| | 32 F | 0.92 ± 0.07 | 0.84 ± 0.20 | 0.92 ± 0.10 | 0.98 ± 0.02 | 0.97 ± 0.02 | 0.89 ± 0.08 | 0.82 ± 0.13 | 0.84 ± 0.11 | 0.99 ± 0.00 | 0.99 ± 0.02 | 0.92 ± 0.10 |
| | 64 F | 0.80 ± 0.00 | 0.67 ± 0.21 | 0.78 ± 0.11 | 0.90 ± 0.07 | 0.86 ± 0.08 | 0.76 ± 0.00 | 0.67 ± 0.00 | 0.70 ± 0.00 | 0.96 ± 0.03 | 0.99 ± 0.00 | 0.81 ± 0.13 |

Table 12: **Reconstruction error on random LoRAs** The error is larger in comparison to reconstructing trained (i.e., non-random) LoRAs in Table 11 for the corresponding compression methods.

| Model Type | Method Type | Tasks | | | | | | | | | | Average |
|---|---|---|---|---|---|---|---|---|---|---|---|---|
| | | task039 | task190 | task280 | task290 | task391 | task442 | task620 | task1342 | task1391 | task1598 | |
| | base | 24.44 | 1.60 | 19.13 | 39.22 | 10.27 | 35.46 | 7.85 | 6.22 | 17.82 | 38.87 | 20.09 |
| | lora | 95.00 | 86.00 | 99.00 | 93.67 | 94.33 | 74.88 | 74.40 | 26.68 | 95.00 | 50.32 | 78.93 |
| 10 full (F) | 32 F | 97.00 | 90.00 | 99.00 | 93.33 | 94.67 | 74.09 | 72.13 | 27.83 | 94.00 | 50.71 | 79.28 |
| | 64 F | 95.00 | 89.00 | 99.00 | 93.67 | 94.67 | 74.29 | 74.80 | 26.63 | 96.00 | 51.04 | 79.41 |
| 50 full (F) | 32 F | 96.00 | 88.00 | 99.00 | 93.67 | 92.33 | 72.30 | 75.97 | 29.89 | 94.00 | 45.68 | 78.68 |
| | 64 F | 98.00 | 88.00 | 99.00 | 93.67 | 93.33 | 72.74 | 76.50 | 29.33 | 96.00 | 45.71 | 79.33 |
| 100 full (F) | 32 F | 92.10 | 83.00 | 99.00 | 93.67 | 92.00 | 71.09 | 63.29 | 27.87 | 88.00 | 42.36 | 75.24 |
| | 64 F | 97.00 | 87.00 | 99.00 | 93.67 | 92.33 | 72.23 | 74.69 | 29.98 | 95.00 | 44.71 | 78.56 |
| 500 full (F) | 32 F | 68.92 | 43.00 | 87.00 | 91.67 | 90.67 | 70.08 | 51.16 | 14.40 | 83.00 | 41.97 | 64.19 |
| | 64 F | 93.50 | 78.00 | 91.00 | 92.33 | 90.33 | 72.55 | 57.49 | 15.44 | 85.00 | 42.31 | 71.80 |

Table 13: **Performance with convergence** In-Distribution **Rouge-L**

Table 14: Agreement Comparison. 100 LoRAs

| Configuration | | Agreement (%) |
|---|---|---|
| Base Model | | 83.015 |
| Uncompressed LoRAs | | 100.000 |
| **Joint Compression** | | |
| Diagonal | Rank 8 | 87.032 |
| | Rank 16 | 88.908 |
| | Rank 32 | 91.545 |
| | Rank 64 | 94.659 |
| Full | Rank 8 | 87.686 |
| | Rank 16 | 90.163 |
| | Rank 32 | 94.018 |
| | Rank 64 | 96.918 |

Table 15: Performance Comparison. 100 LoRAs

| Configuration | | Average Performance |
|---|---|---|
| Base Model | | 32.28 |
| Uncompressed LoRAs | | 48.32 |
| **Join Compression** | | |
| Diagonal | Rank 8 | 41.90 |
| | Rank 16 | 45.44 |
| | Rank 32 | 46.89 |
| | Rank 64 | 47.43 |
| Full | Rank 8 | 43.88 |
| | Rank 16 | 45.79 |
| | Rank 32 | 46.83 |
| | Rank 64 | 47.66 |

Table 16: Task-Based Performance Evaluation Across Different Models and Ranks

| Task | Base Model | LoRA | Diagonal R8 | Diagonal R16 | Diagonal R32 | Diagonal R64 |
|---|---|---|---|---|---|---|
| Causal Judgement | 57.47 | 64.37 | 55.17 | 58.62 | 58.62 | 58.62 |
| Date Understanding | 15.33 | 23.33 | 20.67 | 22.00 | 21.33 | 22.67 |
| Formal Fallacies | 51.33 | 56.00 | 52.67 | 52.67 | 53.33 | 54.67 |
| Hyperbaton | 6.67 | 68.00 | 57.33 | 63.33 | 67.33 | 68.00 |
| Logical Deduction (5 Objects) | 21.33 | 37.33 | 32.00 | 36.67 | 37.33 | 37.33 |
| Logical Deduction (7 Objects) | 12.67 | 44.00 | 31.33 | 42.67 | 44.67 | 45.33 |
| Movie Recommendation | 62.67 | 67.33 | 62.00 | 64.67 | 66.67 | 67.33 |
| Object Counting | 34.67 | 38.00 | 35.33 | 36.67 | 36.67 | 38.00 |
| Snarks | 50.00 | 61.54 | 53.85 | 56.41 | 58.97 | 57.69 |
| Temporal Sequences | 16.67 | 23.33 | 18.67 | 20.67 | 24.00 | 24.67 |
| **Average** | 32.88 | 48.32 | 41.90 | 45.44 | 46.89 | 47.43 |

Table 17: Task-Based Performance Evaluation Across Different Models and Ranks

| Task | Base Model | LoRA | Full R8 | Full R16 | Full R32 | Full R64 |
|---|---|---|---|---|---|---|
| Causal Judgement | 57.47 | 64.37 | 56.32 | 57.47 | 58.62 | 60.92 |
| Date Understanding | 15.33 | 23.33 | 19.33 | 22.00 | 22.67 | 22.67 |
| Formal Fallacies | 51.33 | 56.00 | 51.33 | 52.67 | 53.33 | 56.00 |
| Hyperbaton | 6.67 | 68.00 | 63.33 | 66.00 | 69.33 | 68.00 |
| Logical Deduction (5 Objects) | 21.33 | 37.33 | 35.33 | 36.00 | 35.33 | 37.33 |
| Logical Deduction (7 Objects) | 12.67 | 44.00 | 40.00 | 44.67 | 44.67 | 44.67 |
| Movie Recommendation | 62.67 | 67.33 | 63.33 | 65.33 | 67.33 | 67.33 |
| Object Counting | 34.67 | 38.00 | 35.33 | 36.67 | 37.33 | 37.33 |
| Snarks | 50.00 | 61.54 | 53.85 | 55.13 | 57.69 | 58.97 |
| Temporal Sequences | 16.67 | 23.33 | 20.67 | 22.00 | 22.00 | 23.33 |
| **Average** | 32.88 | 48.32 | 43.88 | 45.79 | 46.83 | 47.66 |

