# OpenReview forum: "Compress then Serve: Serving Thousands of LoRA Adapters with Little Overhead"
_ICLR.cc/2025/Conference — ICLR 2025 Conference Desk Rejected Submission_

### Official Review · Reviewer_VvW1 · 2024-10-27

**Soundness:** 2
**Presentation:** 3
**Contribution:** 2
**Rating:** 5
**Confidence:** 4

**Summary:**

This work proposes a new approach to compress multiple LoRAs together to reduce the frequent CPU-GPU swapping when serving multiple LoRAs with the same base model (like S-LoRA). The essential technique is an SVD-like compression algorithm that decomposes multiple LoRAs simultaneously. To further support more LoRAs, a clustering-based approach is devised, which groups LoRAs into different clusters and apply the SVD-like compression within each group. The authors trained 500 LoRAs across different tasks to evaluate the effectiveness, including the task performance (in terms of validation metrics) and serving efficiency (in terms of serving throughputs).

**Strengths:**

This work studies a timely and important problem. How to serve multiple LoRAs more efficiently can help us move towards better personalization of foundation models. The authors made extensive efforts to train LoRAs over 500 tasks, which, from my humble opinion, may be valuable to the community.

**Weaknesses:**

Despite the significance in the studied topic and the self-trained LoRAs, the technical depth of this work seems rather shallow. There are many important experiments lacking, and the authors should go deeper in when is each proposed technique suitable and effective. Please see questions below.

**Questions:**

1. To motivate this work, I suggest the author conduct a detailed breakdown to show how much time (and ratio to end-to-end inference) is taken to load the LoRAs.

2. There lacks the time cost about the compression in the experiments. Besides, there are no discussions or experiments demonstrating how much the memory reduction is after applying compression. Please add experiments to show the compression time as well as the memory usage before and after compression, under different numbers of LoRAs.

3. After applying the compression, the computation also differs. It would be nice to see how this approach affects the computation cost of LoRAs (e.g., via a detailed breakdown of the inference time).

4. In addition to the tradeoff between task performance and reconstruction error (Figure 3), I believe showing the tradeoff between task performance and serving efficiency is more essential to practical uses. In other words, when compressing more LoRAs together, we could achieve better serving efficiency, while the task performance goes down. An in-depth study would be nice to have.

5. Based on the experimental results, it is hard to tell which approximation (i.e., JD-Full and JD-Diag) is better and how the number of clusters should be configured. It would be better to conclude a few takeaways to improve the practicality of the proposed techniques.

6. As an extension, I would like to see discussions about how this work could apply to larger models. In particular, when the base model is larger (e.g., over 100B), the portion of memory consumption of LoRAs should be smaller. Then, how would the effectiveness of this work becomes?

7. Minor comment: Punica (https://arxiv.org/abs/2310.18547), another work about multi-LoRA serving, is not mentioned.

---

> ### Author Response · Authors · 2024-11-20
> **Response**
>
> We appreciate the reviewer's constructive feedback.
>
> ### On the need for deeper technical exploration and additional experiments:
> We acknowledge the importance of comprehensive experimentation. We will conduct further analyses and include additional experiments to deepen the technical depth of our work before the end of the rebuttal period. These would include detailed breakdowns of compression time, memory usage, and the impact on computation costs.
>
> ### Time to load LoRAs and computation cost differences post-compression:
> We note that vLLM is a highly optimized system that performs a lot of operations in parallel, e.g., it uses non-blocking CPU-GPU communication to load LoRAs. However, we can conduct the requested analysis of loading time and computation time in a simpler setting, i.e., measuring the time it takes to load LoRA weights onto a GPU and timing the computations required for a single layer/token with original and compressed LoRAs. We are working on this experiment and will share the results before the rebuttal period ends. We expect that gains due to compressed LoRAs will be even higher in the absence of system design optimization due to vLLM.
>
> ### Regarding time cost for compression and memory usage experiments:
> We will add results detailing the time required for compression. We have added experiments showcasing memory usage before and after applying compression across varying numbers of LoRAs. These results are discussed in Section 6, and Figure 2 provides a detailed visual representation of memory reduction. Additional tables in Appendix G provide further details. Please let us know if further clarifications are needed regarding these experiments.
>
> ### Concerning the tradeoff between task performance and serving efficiency:
> We have added a new Figure 2b illustrating the tradeoff between task performance and serving efficiency. Note that Figures 1 and 4 already highlight configurations maintaining over 99% of performance (measured via ROUGE-L scores) while optimizing serving efficiency. The thick portions of the curves in Figure 4 correspond to configurations preserving performance.
>
> ### On determining the suitability of JD-Full and JD-Diag and configuring clusters:
> We have included a subsection titled "Recommendations" to provide concrete guidance on selecting hyperparameters, deciding between JD-Full and JD-Diag, and determining the number of clusters.
>
> ### Regarding applicability to larger models (e.g., >100B parameters):
> Size of LoRA adapters is typically around 1% of the model size (regardless of the model being 7B or >100B). Assuming fixed GPU memory, larger LLM will consume more of the GPU memory leaving less memory available to hold LoRAs in memory for efficient serving. In such a setting, compression becomes even more essential to maintain high throughput. Please let us know if this addresses your question, and we’ll add a comment regarding larger models to the discussion in the paper.

---

> > ### Author Response · Authors · 2024-11-20
> > **Response**
> >
> > ### Regarding related work Punica:
> > Thank you for highlighting this important point. **Punica** [Chen et al., 2023] is indeed highly relevant to our work, as its Segmented Gather Matrix-Vector Multiplication (SGMV) approach enhances multi-LoRA serving by efficiently parallelizing feature-weight multiplications in batches and grouping requests that use the same LoRA model. Our method offers an orthogonal approach focused on reducing parameters to efficiently serve multiple LoRAs, which can be integrated with Punica's methods to potentially further improve performance.
> >
> > In our vLLM experiments, we specifically utilized the Punica kernel for implementing multi-LoRA, applying our approach with Punica's capabilities. Our custom function, `add_lora_slice_with_sigma`, implements the following key steps:
> >
> > 1. **Initialize Buffers**: Creates temporary storage for intermediate calculations if not already provided.
> > 2. **Apply Matrix A**: Transforms `x` using matrix `A`, storing the result in `buffer`.
> > 3. **Apply Matrix Sigma**: Further transforms `buffer` using `Sigma`, storing the result in `buffer_sigma`.
> > 4. **Apply Matrix B and Update `y`**: Finally, transforms `buffer_sigma` using `B`, applies scaling, and updates a slice of `y` in place.
> >
> > Below is the pseudocode for `add_lora_slice_with_sigma`, illustrating the integration:
> >
> > ```python
> > Function add_lora_slice_with_sigma(y, x, wa_t_all, wb_t_all, wsigma_t_all, indices, layer_idx, scale, y_offset, y_slice_size, buffer=None):
> >     # Initialize buffers if not provided
> >     if buffer is None:
> >         buffer = create_tensor(shape=(x.size(0), R), dtype=float32)
> >         buffer_sigma = create_tensor(shape=(buffer.size(0), R), dtype=float32)
> >     # Step 1: Apply matrix A
> >     dispatch_bgmv_low_level(buffer, x, wa_t_all, indices, layer_idx, scale=1.0)
> >     # Step 2: Apply matrix Sigma
> >     dispatch_bgmv_low_level(buffer_sigma, buffer, wsigma_t_all, indices, layer_idx, scale=1.0)
> >     # Step 3: Apply matrix B and update y slice
> >     dispatch_bgmv_low_level(y, buffer_sigma, wb_t_all, indices, layer_idx, scale, y_offset, y_slice_size)
> > End Function
> > ```
> >
> > We have now incorporated a discussion of Punica into our revised manuscript to acknowledge this relevant work.

---

### Official Review · Reviewer_uG8u · 2024-10-28

**Soundness:** 4
**Presentation:** 4
**Contribution:** 3
**Rating:** 6
**Confidence:** 4

**Summary:**

This paper addresses the pressing challenge of efficiently serving a large number of low-rank adaptation (LoRA) adapters in real-time systems. The authors propose a novel Joint Diagonalization (JD) method that combines multiple LoRA adapters' weights into shared matrices, significantly reducing the storage and computational overhead required for inference. The experimental results demonstrate that this approach maintains model performance while enhancing throughput, particularly in scenarios involving thousands of adapters.

**Strengths:**

- The paper tackles a critical and timely issue in deploying large-scale machine learning models, particularly in real-time applications. The need to serve numerous LoRA adapters efficiently is a relevant concern in the field.
- The proposed JD method is a novel approach that leverages shared matrices to compress multiple LoRA adapters effectively. This innovation contributes significantly to the field of model compression and efficiency.
- The authors provide solid experimental evidence supporting their claims. The results illustrate that the proposed method effectively retains performance while significantly improving throughput, which is crucial for practical implementations.
- The introduction of a clustering strategy to handle LoRA adapters with varying ranks demonstrates a thoughtful extension of the core methodology, allowing for greater flexibility and scalability in real-world applications.

**Weaknesses:**

- The experimental section of the paper utilizes a uniform rank for the LoRA method across all fine-tuning tasks, demonstrating that this approach can achieve lower reconstruction loss. However, in current multi-LoRA deployment systems, the required rank values often vary significantly across different tasks. When there is a substantial disparity in the rank values of different LoRA modules, the method presented in this paper may be less effective. Although it is feasible to stratify LoRA parameters of varying ranks into different groups using clustering methods, the advantages of doing so may be limited.
- Several efficient fine-tuning methods with shared parameters have been proposed to reduce the parameter count in LoRA. For instance, VeRA [1] suggests fine-tuning a large language model by sharing global static parameters while setting local scaling variables. In a multi-LoRA deployment environment, VeRA naturally incorporates shared parameters and does not require additional computation to extract these shared parameters from different LoRAs. Using VeRA for fine-tuning eliminates the need for an additional computational process for shared parameter extraction, while also avoiding the introduction of errors. Furthermore, VeRA supports LoRAs with varying rank values without relying on clustering methods. In contrast, the JD method may introduce some reconstruction errors and necessitate additional computations, lacking a clear advantage.
- The experiments conducted in the paper utilize only the Mistral-7B-Instruct-v0 base model, which does not sufficiently demonstrate the applicability of the proposed method to other base models, such as those in the Llama series. Additionally, the experiments do not account for larger-scale models (e.g., 13B and 70B), which leaves unverified the effectiveness of the method in the context of large-scale models.
- The clustering methodology described lacks specific details. The paper mentions the use of clustering algorithms like k-means but does not elaborate on how different LoRA parameters are clustered or how the number of clusters is determined. Providing more detailed information on these aspects would significantly enhance the clarity and rigor of the methodology.

[1] VeRA: Vector-based Random Matrix Adaptation

**Questions:**

- The experimental section of the paper reports training LoRA on 500 distinct types of tasks. However, it is currently unclear what the primary categories of these tasks are and whether there is a high degree of task similarity. If there is a high similarity among the tasks, it is hypothesized that the resulting LoRA parameters may also be more similar, potentially leading to lower reconstruction loss. It would be beneficial to explore whether this is the case and to discuss how the reconstruction loss might compare when tasks are less related or entirely dissimilar.
- The paper introduces two distinct methods, JD-Full and JD-Diag. It is crucial to analyze under what scenarios each method is more suitable. Are there experimental comparisons that indicate which method achieves lower reconstruction loss in scenarios with higher ranks or with a larger number of clusters? Providing such analysis would significantly enhance the understanding of when and where each method is most effective.
- Regarding Figure 2, data is only presented for 0-100 and 500 LoRA instances, with other quantities left unfilled. It would be valuable to understand if there are any trends or patterns that emerge with different numbers of LoRA instances. Clarifying this would provide a more comprehensive understanding of the relationship between the number of LoRA instances and the associated outcomes.

---

> ### Author Response · Authors · 2024-11-20
> **Response**
>
> Thank you for your valuable comments.
>
> ### On effectiveness when LoRA modules have disparate rank values:
> This is an interesting question! One simple solution is to treat LoRAs with especially high ranks as singleton clusters (i.e., don’t compress them). We are currently training LoRAs with varying ranks and will include these results before the rebuttal period ends.
>
> ### Regarding VeRA: Vector-based Random Matrix Adaptation:
> VeRA represents a novel finetuning approach, whereas our work focuses on model compression rather than finetuning. Unlike VeRA, our method does not depend on specific training of PEFT, though it is designed with LoRA in mind. While LoRA may not remain the dominant PEFT paradigm indefinitely, the current challenges associated with serving multiple LoRAs remain significant. Notably, our compression method can be extended to the scaling vectors of VeRA, which opens an intriguing avenue for future research.
>
> ### Concerns about lack of experiments on larger-scale models (e.g., 13B and 70B):
> We respectfully note that while testing on larger models is valuable, it is often infeasible due to resource constraints and the lack of publicly available, similarly formatted fine-tuned LoRAs with necessary metadata for evaluation. Our method operates at the matrix (layer) level, making its effectiveness largely independent of the overall model size. In larger models, our compression becomes even more beneficial due to increased memory demands.
>
> We also would like to emphasize that we trained 500 LoRAs for the experiments in this paper, which is the largest collection of (reproducible) LoRAs to the best of our knowledge. We are also committed to releasing the LoRA weights, corresponding datasets, and code for reproducibility and future work on compression and merging.
>
> ### Clarification on clustering methodology:
> The clustering algorithm is presented in Appendix A.3. We have also added a subsection titled "Recommendations," where we detail how to choose the number of clusters.
>
> ### On the primary categories of the 500 tasks and task similarity:
> We have clarified in Section 5 that our approach does not depend on well-clustered LoRA adapters. We selected 10 diverse tasks and randomly sampled the remaining 490 tasks from the Natural Instructions dataset [1]. According to [1], "This large and diverse collection of tasks enables rigorous benchmarking of cross-task generalization." Thus, the tasks represent a realistic and varied set, not inherently clustered. We will include this list in the released code and have already added it to Appendix, Table 3. Importantly, even with high reconstruction loss, our method achieves significant compression, indicating that the LoRA adapters do not need to be well-clustered.
>
> [1] Wang et al., Natural Instructions: Benchmarking Generalization to New Tasks via Natural Language Instructions. arXiv:2204.07705.
>
> ### Comparing JD-Full and JD-Diag methods and cluster configurations:
> We have expanded our discussion on the scenarios where each method is most suitable. Detailed experimental results are provided in Appendix G. JD-Diag typically requires a higher rank to match the performance of JD-Full. For compressing fewer than 50 LoRAs, JD-Full is preferable. For 100 or more LoRAs, clustering significantly enhances performance due to the large ranks required for JD-Full alone.
>
> We have added a subsection called “Recommendations” as well as Figure 2b to clarify the performance differences.

---

### Official Review · Reviewer_rFdP · 2024-11-03

**Soundness:** 3
**Presentation:** 3
**Contribution:** 3
**Rating:** 5
**Confidence:** 3

**Summary:**

This paper presents a method for compressing low-rank adaptations (LoRAs), which is a common technique used in parameter-efficient fine-tuning. The main idea is to jointly diagonalize the set of LoRA matrices, which allows sharing parameters across multiple LoRAs and reducing the overall number of parameters. The authors also propose a clustering algorithm to improve reconstruction error by grouping similar LoRAs together. They evaluate their approach on a variety of natural instruction tasks and demonstrate that their compression techniques preserve the performance of the original LoRAs while maintaining high throughput when serving over 1000 LoRAs.

**Strengths:**

● The proposed joint diagonalization method is well-motivated and theoretically justified.
● The clustering algorithm helps improve the compression efficiency.
● Experimental results show that the proposed method can achieve high throughput while preserving performance.

**Weaknesses:**

1. The paper mentions that F-LoRA and S-LoRA are related methods, but it doesn't explicitly compare them with the proposed joint diagonalization method. Including a comparison with these and other state-of-the-art methods could highlight the advantages and disadvantages of the proposed method relative to existing solutions.

2. There are some doubts about the specific experimental results, as the model's performance improvement is not clearly addressed. The compression used here is lossy and cannot fully restore the original model's performance. Furthermore, there is a lack of comparative experiments to demonstrate whether the model's performance after compression maintains its efficacy or accuracy compared to the original. These aspects require more detailed data and experimental results for support.

**Questions:**

To fully understand the computational requirements of the proposed method, it's essential to evaluate several factors including the complexity of the algorithms used, the types and sizes of the datasets involved, and the specific hardware configurations required.

Regarding latency and resource usage:
- Latency: The method's impact on latency should be assessed by comparing the time it takes to process tasks with and without the use of the proposed method. This will reveal if there's an increase in processing time or if the method helps reduce delays typically associated with computation.
- Resource Usage: Analyzing resource usage involves looking at how much CPU and GPU time is required, the memory footprint, and whether the method optimizes or increases the demand on these resources. Additionally, it would be beneficial to compare the resource consumption with that of other existing methods to determine its efficiency.

For a comprehensive evaluation, you might consider setting up experiments that measure these metrics under controlled conditions, ensuring you can directly observe the impacts and effectively compare them with other techniques.

---

> ### Author Response · Authors · 2024-11-20
> **Response**
>
> We appreciate the reviewer's thoughtful feedback.
>
> ### Regarding comparison with related methods like F-LoRA and S-LoRA:
> Our method also avoids batched matrix multiplication (BMM), similar to F-LoRA, as discussed in the Appendix D. We have added a reference to this in the main text (Section 2). S-LoRA aligns with the approach used by vLLM [2] (i.e., our vLLM multi-LoRA baseline is S-LoRA adopted by vLLM; note that both S-LoRA and vLLM were published by the same research group), and our method is compatible with it. While vLLM does not currently support F-LoRA, implementing it would require custom CUDA kernels. We expect our method to have similar forward-pass speed to F-LoRA while offering additional compression benefits.
>
> We will include experiments timing matrix multiplications to compare LoRA computations for a single token/layer among F-LoRA, multi-LoRA with BMM, and our method, before the rebuttal period concludes.
>
> [2] vLLM: https://github.com/vllm-project/vllm
>
> ### Concerns about model performance after lossy compression:
> Please note that we did perform an extensive analysis of the performance of compressed LoRAs. Specifically, in Figure 2 we report results only for the compression settings where at least 99% of the LoRA performance is preserved (in terms of RougeL). We also present a detailed performance analysis in the Appendix for various metrics:
>
> - **Relative ROUGE-L Performance** (Appendix G.1)
> - **Absolute ROUGE-L Performance** (Appendix G.2)
> - **Relative ROUGE-1 Performance** (Appendix G.3)
> - **Absolute ROUGE-1 Performance** (Appendix G.4)
> - **Relative Exact-Match Performance** (Appendix G.5)
> - **Cross-Entropy Loss** (Appendix G.6)
> - **Agreement Metric** (Appendix G.7), a new metric measuring exact matches in task generations between the uncompressed and compressed models.
>
> These results indicate that our compression method preserves the model's efficacy and accuracy.
>
> In addition, we have added a new Figure 2(b) that provides detailed comparative analysis between compressed and original LoRAs, where each data point (represented as a bubble) corresponds to a different compression configuration, with bubble size indicating the number of compressed LoRAs. While non-clustering compression at higher rates does show some performance trade-offs, our extensive experiments demonstrate that our cluster-based compression method successfully maintains and even slightly improves performance while compressing up to 500 unique LoRAs.
>
> ### On latency and resource usage:
> We acknowledge the importance of analyzing resource consumption. Due to the complexity of systems like vLLM, which involve optimized overlapping computations, disentangling individual computational aspects is challenging.
>
> We will conduct time measurements in simpler environments to assess the computational time associated with different aspects of serving compressed LoRAs, before the rebuttal period ends.

---

### Official Review · Reviewer_GYVN · 2024-11-04

**Soundness:** 2
**Presentation:** 3
**Contribution:** 3
**Rating:** 5
**Confidence:** 3

**Summary:**

This paper proposes a method to accelerate multi-LoRA serving by jointly compressing multiple LoRAs together. They use a method that shares a basis and then rescales the basis to represent different LoRA adapters. They also extend their method to group clusters of LoRAs together which are more amenable to joint compression, and leverage an alternating assignment algorithm to optimize the cluster assignments and compressed LoRA representations. This method allows for storing a large number of LoRAs more compactly in memory, and also serving these adapters with significantly lower costs than existing LoRA serving systems.

**Strengths:**

- They present a novel joint compression approach for multi-LoRA serving that compresses together LoRAs using a joint diagonalization method (which projects groups of LoRAs onto a shared basis)
- They propose a clustering-based approach that groups together similar LoRAs (which are more amenable to shared basis compression) and then alternates cluster assignment with joint compression. This is necessary for compressing large numbers of LoRAs together without significant accuracy degradation.
- They provide theoretical error bounds on the reconstruction error from their method
- They also provide comprehensive evaluation for their method

**Weaknesses:**

- This approach relies on having well-clustered LoRA adapters for high attainable compression
- For multiple users who want to use LoRA-as-a-service, it seems unlikely that they would want the accuracy of their fine-tuned model to be dependent on the other LoRAs that are included in the batch (or to have the behavior of their model be influenced by the other LoRAs that their adapter is grouped with).
- For the throughput improvements, this work relies on the assumption that the LoRAs that are used together are consistent for throughput benefits (as if there are changes in terms of which LoRAs are being used heavily with a dynamic workload, then the throughput benefits would be more limited). Have the authors considered evaluating their method under dynamic workloads in order to assess whether the throughput benefits are comparable, or considered the implications of how their method could be applied in such a scenario?
  - For edge deployment (with one base model and many downstream task adapters), this approach would help significantly in terms of memory savings; however, for these applications there would be no throughput / latency benefits.

**Questions:**

- Are there any privacy concerns with this type of approach? It would be useful if the authors could provide an analysis of privacy implications of their joint compression approach (specifically, whether it is possible for joint compression to leak any information).
  - Do the authors have any ablation for the following: given the base model with adapter A for task T_A which has been jointly compressed with adapter B for task T_B, does the base model with adapter A now inadvertently perform better on task T_B, or does it maintain its "independence" in the sense that the compressed A adapter hasn't been leaked information from adapter B?
- I didn't understand the connection to LoRA merging at the end of Section 4, and how this supports the claims of the paper. Intuitively, I don't think that averaging adapters is actually a desired property (since it means that each adapter that you add in would bias the other adapters to be more similar to the added adapter). It would be helpful if the authors could clarify in the rebuttal what the intended implications are with the LoRA merging comparison.
- One minor point is that I found the evaluation hard to follow - this may be because there are multiple moving parts (number of LoRAs, rank, number of clusters, different "full" / "diagonal" variants of their method) which makes it hard to present results.

---

> ### Author Response · Authors · 2024-11-20
> **Response**
>
> We thank the reviewer for their insightful comments.
>
> ### Reliance on Well-Clustered LoRA Adapters for High Compression
> We have clarified in **Section 5** that our approach does not depend on well-clustered LoRA adapters. We selected 10 diverse tasks and randomly sampled the remaining 490 tasks from the Natural Instructions dataset [1]. According to [1], *"This large and diverse collection of tasks enables rigorous benchmarking of cross-task generalization."* Thus, the tasks represent a realistic and varied set, not inherently clustered. We will include this list in the released code and in **Appendix C, Table 3**. Importantly, even with high reconstruction loss, our method achieves significant compression, indicating that the LoRA adapters do not need to be well-clustered.
>
> [1] Wang et al., *Natural Instructions: Benchmarking Generalization to New Tasks via Natural Language Instructions*. arXiv:2204.07705.
>
> ### Dependence of a User's Model Accuracy on Other LoRAs in the Batch
> We acknowledge this concern but note that leveraging shared information across data points is common in compression methods to achieve efficiency. Our primary goal is to ensure maintained performance for each user's task. Understanding how grouped LoRAs influence individual performance is an interesting direction for future work.
>
> ### Throughput Improvements Under Dynamic Workloads
> Our compression enables all LoRAs to reside in GPU memory, making throughput largely independent of workload distribution. If the workload changes—for example, if most requests involve the same LoRA—alternative scheduling could be similarly effective. In our experiments, each request queries a single LoRA, but users could request multiple LoRAs per batch [1], potentially enhancing our method's throughput compared to the baseline. We will include an experiment using a Poisson distribution in the Appendix to address dynamic scenarios before the rebuttal period ends.
>
> [1] *Towards Modular LLMs by Building and Reusing a Library of LoRAs*, (2024), Oleksiy Ostapenko, et al.
>
> ### Privacy Concerns with Joint Compression
> We identify two privacy considerations:
>
> 1. **Privacy of training data**: Our method poses minimal risk here, as affected LoRAs can be trained with differentially private SGD, preserving privacy through compositionality. Any indirect influence on other LoRAs would be negligible.
> 2. **Privacy of task inclusion on the server**: While a user might infer the presence of another task, practical barriers such as the need for targeted knowledge and minimal performance gains make this unlikely. For sensitive applications, clients could opt for their LoRA to be in a singleton cluster, preventing potential leakage.
>
> Exploring these implications is an interesting avenue for future work. We will include experiments of the ablation study you described before the rebuttal period concludes.
>
> ### Clarification on the Connection to LoRA Merging in Section 4
> As discussed in Shah et al. (2023) and Huang et al. (2024), averaging or merging weights of two LLMs can be beneficial, often leading to improved performance. This is possible because the weight space is expansive, allowing the merged model to incorporate abilities from both without significant interference. In our context, we observe that a smaller rank \( r \) suffices for low reconstruction error. This suggests that tolerating larger weight-matrix errors without performance loss may stem from similar principles observed in model merging. Further exploration of this connection is beyond our current scope but warrants future investigation.
>
> ### Evaluation Clarity Due to Multiple Variables
> Thank you for highlighting this issue. We have added **Figure 2b** as well as a new **Section 6.4** to enhance clarity and make the results more accessible. Does this help?

---

### Comment · Senior_Area_Chairs · 2024-11-22
**URGENT: NON-ANONYMOUS REVISION**

The authors have (I assume mistakenly) posted a non-anonymous revised version of their submission. We'll need to figure out how to proceed, but in the meantime to keep as many options as possible open:

**Reviewers:** Please *do not download the revision* the authors have posted, as it includes their names and affiliations.
*If you have already seen the revised version with the author identities*, please say so in a comment as soon as possible.

**Authors:** Please *replace the revised version with an anonymized version* as soon as possible.

---

> ### Author Response · Authors · 2024-11-22
> **Reponse**
>
> We apologize for mistakenly uploading a non-anonymous version of our submission! This was unintentional, and we deeply regret any inconvenience or complications it may have caused. We have just promptly replaced the revised version with the correct anonymized document as per your instructions.
> Please let us know if there are any further steps we should take to address this issue.

---

### Note · Program_Chairs · 2024-11-23
**Submission Desk Rejected by Program Chairs**

The paper is desk rejected for breaking the anonymity policy. During the rebuttal period, a revised PDF was uploaded that contained the author names on the first page. Since the author names were seen by the reviewers, the review process is no longer double blind for this paper, preventing further review. This decision was discussed and confirmed by the program chair committee.